# CONFEX: Uncertainty-Aware Counterfactual Explanations with Conformal Guarantees

## Abstract

Counterfactual explanations (CFXs) provide human-understandable justifications for model predictions, enabling actionable recourse and enhancing interpretability. To be reliable, CFXs must avoid regions of high predictive uncertainty, where explanations may be misleading or inapplicable. However, existing methods often neglect uncertainty or lack principled mechanisms for incorporating it with formal guarantees. We propose CONFEX, a novel method for generating uncertainty-aware counterfactual explanations using Conformal Prediction (CP) and Mixed-Integer Linear Programming (MILP). CONFEX explanations are designed to provide local coverage guarantees, addressing the issue that CFX generation violates exchangeability. To do so, we develop a novel localised CP procedure that enjoys an efficient MILP encoding by leveraging an offline tree-based partitioning of the input space. This way, CONFEX generates CFXs with rigorous guarantees on both predictive uncertainty and optimality. We evaluate CONFEX against state-of-the-art methods across diverse benchmarks and metrics, demonstrating that in many cases, our approach more robust and plausible explanations compared to competing uncertainty-aware generators.

## 1 Introduction

Machine learning models are deployed in high-stakes decision-making scenarios like loan approvals, medical diagnoses, and employment screening. In these contexts, algorithmic recourse—providing actionable feedback to individuals influenced by these decisions—is not just a technical concern but also an ethical and legal imperative. Although the legal status of "right to explanations" under the EU's General Data Protection Regulation (GDPR) remains contested (Wachter et al., 2017; Selbst & Barocas, 2018), there is growing consensus that individuals should be offered meaningful information about algorithmic decisions that impact them (Edwards & Veale, 2017; Binns et al., 2018).

Counterfactual explanations (*CFX*) were formally introduced by Wachter et al. (2017) as a method for algorithmic recourse. CFXs answer questions like: "What minimal changes to my input features would have altered the model's decision desirably?", and Wachter's formalisation focuses on finding counterfactual explanations that are minimally close to the original point (*factual instance*) or have sparse feature changes. These criteria of closeness and sparseness have been extended in later methods to other desiderata such as diversity, causality, actionability, and plausibility, to generate explanations that work better as a recourse path and are distinguished from adversarial examples.

However, most existing CFX methods fail to account for the inherent uncertainty in both data and model predictions. This is problematic because explanations that ignore uncertainty may lead to false confidence in suggested changes, potentially resulting in ineffective recourse actions when deployed in practice. Uncertainty quantification in CFX is thus crucial for generating reliable and actionable insights.

We introduce CONFEX, an uncertainty-aware CFX generator that builds on *Conformal Prediction* (*CP*) (Vovk et al., 2022; Angelopoulos et al., 2023). CP is a popular uncertainty quantification framework that offers distribution-free and finite-sample coverage guarantees. It works by using calibration data to construct prediction regions that contain the true (unknown) outcome with a user-specified probability. CP does not require assumptions on the data distribution and the underlying model, except that the calibration data and the test point must be exchangeable. The core idea of our CONFEX method is to constrain the search space for CFXs only to those points leading to a

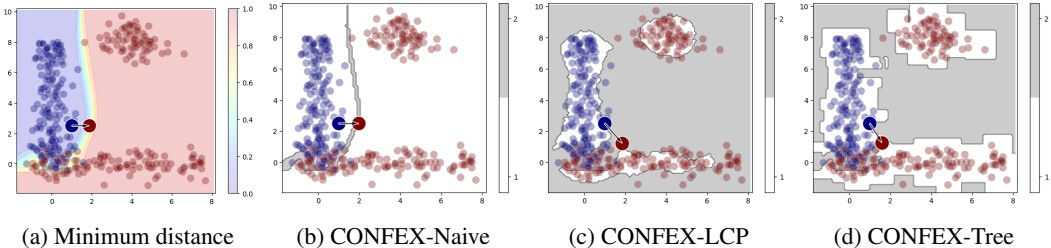

(a) Minimum distance     (b) CONFEX-Naive     (c) CONFEX-LCP     (d) CONFEX-Tree

Figure 1: Counterfactuals produced for the same factual instance (marked in blue) for a MLP classifier using approaches MILP-MinDist, CONFEX-Naive, CONFEX-LCP, CONFEX-Tree. CONFEX approaches use bandwidth as 35% of the median pairwise distance between calibration points, and alpha as 2%.

singleton prediction region $\{y^+\}$, i.e., points that yield the desired outcome $y^+$ with a high degree of certainty, since non-singleton CP regions represent uncertain predictions.

To illustrate our methods, Fig. 1a displays CFXs produced over a synthetic 2D dataset inspired from Poyiadzi et al. (2020). We can observe that counterfactuals produced by the minimal distance approach and by a naive application of CP to the CFX generation problem, called CONFEX-Naive (Section 3), fail to be plausible with respect to the data distribution.

These issues with naively applying CP to CFX generation stem from the fact that the generated (test-time) CFX may not be exchangeable with the calibration points, thereby affecting the validity of CP's guarantees. We solve this by imposing stricter coverage requirements for CP: we build prediction regions that approximately[1] attain *local* (aka *test-conditional*) guarantees, i.e., the target coverage probability is achieved for *any* test point. In contrast, normally, CP guarantees are marginal, i.e., the coverage probability is averaged over the joint calibration and test distribution.

Our CONFEX method relies on a *Mixed-Integer Linear Programming* (*MILP*) encoding of the optimisation problem, which not only guarantees optimality of solutions but also ensures satisfaction of the CP constraints. We present two methods for incorporating local coverage constraints. The first is *localised CP* (Guan, 2023), which frames conditional coverage as a covariate shift problem (Tibshirani et al., 2019). However, it requires encoding and solving calibration quantiles in MILP, which is computationally expensive and scales poorly with the dataset size. The second, more efficient, method is a KD-tree-based encoding of local calibration quantiles. For this method, we use regression trees, which can be efficiently encoded in MILP.

In summary, our main contributions are:

1. a mathematical formulation for distribution-free uncertainty-aware counterfactual explanations, the first to apply conformal prediction in a principled manner (i.e., by addressing the exchangeability problem via test-conditional coverage, retaining formal guarantees);

2. a novel localised CP procedure, with an efficient MILP encoding, for generation of CFXs, which can be used more generally to incorporate (test-conditional) CP uncertainty constraints in any search problem;

3. an extensive experimental evaluation demonstrating that our CONFEX method outperforms competing uncertainty-aware generators by providing more certain, plausible and stable explanations, as well as enjoying formal guarantees on uncertainty.

## 2   BACKGROUND AND PROBLEM FORMULATION

**Counterfactual Explanations**   Let $\hat{f} : \mathcal{X} \to \mathcal{Y}$ denote a trained classifier for which we seek to generate counterfactual explanations. Given an instance $x_0 \in \mathcal{X}$ such that $\hat{f}(x_0) \neq y^+$, the goal is to identify a counterfactual instance $x'$ such that $\hat{f}(x') = y^+$. Wachter et al. (2017) frame this as an

---

[1]Exact conditional guarantees for CP are known to be impossible unless the inputs are discrete (Vovk, 2012; Barber et al., 2020).

optimisation problem and solve it via gradient descent.

$$x_{\mathrm{cf}} \in \arg\min_{x'} \max_{\lambda} \left( \lambda \; \mathrm{yloss} \left( \hat{f}\left(x'\right), y^+ \right) + \mathrm{dist}\left(x_0, x'\right) \right). \tag{1}$$

The loss function aims to find an explanation that changes the predicted class to the target class (first term), while also ensuring that the explanation is close to the input instance (second term). Closeness is often defined as an $L_p$ norm, which can be weighted based on the observed data (e.g. the inverse median absolute deviation), or to reflect domain knowledge (Dandl et al., 2020). However, by optimising solely for closeness, this formulation often leads to counterfactual explanations that resemble adversarial examples and may not be actionable or robust.

Desirable properties of CFXs include *validity* (prediction flips to $y^+$), *proximity* (closeness to the factual instance), *sparsity* (few feature changes), *plausibility* (realistic and likely under the data distribution), *actionability* (only mutable features are altered), *causality* (identified counterfactual satisfies causal relationships) and *robustness* (stability under input perturbations); see (Verma et al., 2020; Karimi et al., 2021).

Uncertainty-aware CFX methods show promise for enhancing the robustness and plausibility of CFXs. In this line of work, Schut et al. (2021) propose minimising predictive entropy across an ensemble of models to consider the effect of uncertain regions. Bayesian approaches, such as CLUE (Antorán et al., 2020), leverage predictive uncertainty from Bayesian neural networks to generate epistemically informative counterfactuals.

**Conformal Prediction**   CP is a distribution-free inference framework that complements any predictive model with rigorous uncertainty quantification. CP outputs prediction sets guaranteed to contain the true (unknown) outcome with a user-specified probability $1 - \alpha$ without relying on asymptotic or parametric assumptions (Vovk et al., 2022; Angelopoulos et al., 2023). To construct these sets, CP performs the following steps:

1. **Calibration**: use a held-out calibration dataset $\mathcal{D}_{\mathrm{cal}} = \{(x_i, y_i)\}_{i=1}^{n}$ to find the critical value $q_{1-\alpha}$ (i.e., the $1 - \alpha$ quantile) of a chosen test statistic called the *(non-conformity) score* $s(x, y)$, which is normally chosen to quantify the deviation between the model prediction $\hat{f}(x)$ and the ground truth $y$. This step is performed only once, offline. Formally,

$$q_{1-\alpha} = Q_{1-\alpha} \left( \sum_{i=1}^{n} \frac{1}{n+1} \delta_{s(x_i, y_i)} + \frac{1}{n+1} \delta_{+\infty} \right), \tag{2}$$

   where $Q_{1-\alpha}$ is the $1 - \alpha$ quantile function and $\delta_v$ is the Dirac distribution centered at $v$.

2. **Inference**: for a test input $x^*$, construct a prediction region $C(x^*)$ by including all labels $y$ whose score is below the critical value (i.e., such that $s(x^*, y) \leq q_{1-\alpha}$).

The CP procedure provides the following marginal guarantee for an unseen test point $(x^*, y^*)$:

$$\mathbb{P}_{\mathcal{D}_{\mathrm{cal}}, (x^*, y^*)} \left( y^* \in C_{1-\alpha}(x^*) \right) \geq 1 - \alpha. \tag{3}$$

The above holds in finite sample regimes (as opposed to asymptotic) under the mild condition of exchangeability (a weaker assumption than IID), i.e., the joint distribution of calibration and test points is invariant under permutations. By marginal guarantees, we mean that the coverage probability of equation 3 is achieved on average over the joint calibration and test distribution.

**CP and CFXs**   To our knowledge, there exist only two methods which apply conformal prediction to CFX generation: ECCCo (Altmeyer et al., 2024) and CPICF (Adams et al., 2025).

CPICF (Adams et al., 2025) assumes an alternative "individualised" setting, where an institution holds a private black-box classifier and aims to provide CFXs to individuals without disclosing the classifier. The knowledge of each individual is modelled by their own classifier, and the organisation produces a CFX to reduce uncertainty in the global classifier via CP. This is a fundamentally different setting to ours, furthermore CPICF's formulation does not retain any formal CP guarantees.

In the standard setting, ECCCo extends Wachter's formulation (equation 1) with two additional terms: one that optimises the energy of the identified counterfactual to enhance plausibility, and

one that minimises uncertainty through the smooth conformal set size loss of Stutz et al. (2022). However, ECCCo has the following drawbacks: 1) it incorporates conformal prediction, but in a way that does not address exchangeability issues, which we detail in Section 3.1; 2) the procedure does not guarantee CP regions will have the required size (e.g., singletons); 3) it relies on energy-based training to obtain plausible CFXs. As we will show, our approach instead induces plausible CFXs solely by using CP constraints, formulating these constraints to enforce local validity (thereby solving the exchangeability issues), and thanks to the MILP formulation, it ensures satisfaction of the set size constraints whilst being optimally close.

Both ECCCo and CPICF fail to retain formal guarantees on generated counterfactuals, and mention that further analysis on the role of CP in CFXs is required. In the context of recouse recommendations, wthout uncertainty guarantees, the CFX method may suggest CFXs where the model is uncertain. This may mislead the recipient into making changes that do not actually alter the outcome. Instead we want to produce reliable explanations for every individual, backed up by formal guarantees.

**Mixed Integer Linear Programming (MILP) and CFXs**   MILP provides a framework for formulating and deriving CFXs as a constraint-solving problem. The problem is of finding a point $x'$ which minimises the distance to the original instance $x_0$ whilst being classified as $y^+$.

$$x_{\mathrm{cf}} \in \arg\min_{x'} \mathrm{dist}(x_0, x') \quad \text{s.t. } \hat{f}(x') = y^+ \tag{4}$$

We refer to this method as MILP-MinDist, and it serves as a baseline for our CONFEX method.

To allow the encoding, the model $f$ must be representable in MILP; this is the case for e.g. linear classifiers and multilayer perceptrons with ReLU activations, as well as non-differentiable models such as decision trees. Neural network layers like sigmoid or softmax are not linearly representable, but can be omitted from the MILP encoding if used at the last layer since we can identify if $f(x_{\mathrm{cf}}) = y^+$ based on the logits alone.

When presented to an MILP solver, this approach is guaranteed to yield a valid and optimal CFX, if such an explanation exists. Gradient-based methods, on the other hand, are incomplete, meaning that they may fail to find valid CFXs or may return suboptimal solutions.

We note that properties like causality and actionability can be incorporated in equation 4 through MILP constraints on the input variables; similarly, a set of diverse explanations (as opposed to an individual one) can be generated by repeatedly solving the problem and adding constraints or objective function terms to block or penalize explanations similar to those already identified (Kanamori et al., 2020). By adding such constraints, our method can accommodate these desiderata as well.

**Problem Formulation**   We aim to find CFXs that modify the factual input to the minimum extent necessary to yield the desired label $y^+$ with high probability. Below is the formal problem statement.
*Problem* 1 (Uncertainty-aware CFX). Given a factual input $x_0$ and an error level $\alpha \in (0, 1)$, an *uncertainty-aware counterfactual explanation* $x_{\mathrm{cf}}$ is a solution to the below optimisation problem:

$$x_{\mathrm{cf}} \in \arg\min_{x'} \mathrm{dist}(x_0, x') \quad \text{s.t. } \mathbb{P}_{Y|X=x'}(Y = y^+) \geq 1 - \alpha, \tag{5}$$

where $\mathbb{P}_{Y|X=x'}$ is the conditional distribution of labels $Y$ given $X = x'$. Hence, our base method targets just the minimal distance and low uncertainty desirable CFX properties.

## 3 CFXs with CP Constraints: A Naive Attempt

We first present a naive approach to apply conformal prediction to minimise the uncertainty in the generated CFX, which we call CONFEX-Naive. This approach extends MILP-MinDist (see equation 4) by restricting the search space to points yielding the singleton CP region $\{y^+\}$, i.e., points attaining the target class and with a high degree of certainty:

$$x_{\mathrm{cf}} \in \arg\min_{x'} \mathrm{dist}(x_0, x') \quad \text{s.t. } C_{1-\alpha}(x') = \{y^+\} \tag{6}$$

Note that the above constraint is equivalent to the constraints $s(x', y^+) \leq q_{1-\alpha}$ and $\bigwedge_{y \neq y^+} s(x', y) > q_{1-\alpha}$. The quantile $q_{1-\alpha}$ is pre-computed on the held-out calibration set.

For multi-layer perceptrons, we use the following log-likelihood ratio as the score function

$$s(x, y) = \log\left(\frac{\max_{y' \neq y} p(x)_{y'}}{p(x)_y}\right), \tag{7}$$

where $p(x)_y$ is the softmax probability of $y$ predicted by the model $f$ for input $x$. When the correct class is predicted, the ratio is below 1 and we obtain a negative score. When the model is wrong, the ratio is positive and the score grows bigger as the model confidence on $y$ decreases relative to that on the predicted class. Importantly, equation 7 can be equivalently expressed in a linear form as $s(x, y) = -l(x)_y + \max_{y' \neq y} l(x)_{y'}$, where $l(x)$ is the predicted vector of logits, making it efficiently representable in MILP.

**Relation with MILP-MinDist** We note that our score function is well-formed, i.e., $s(x, y)$ is lowest when $y$ is the label predicted by the model $f$ (and, in particular, $s(x, y)$ increases as the softmax probability of $y$ decreases). Thus, when a CP prediction region returns the singleton $\{y^+\}$, then $y^+$ is the class with the lowest score, i.e., the class predicted by $f$. That is, for any $\alpha \in (0, 1)$, $C_{1-\alpha}(x) = \{y^+\} \rightarrow f(x) = y^+$. This implies that the feasible set of CONFEX is a subset of that of MILP-MinDist, and so, CONFEX explanations can never attain smaller (better) distances than CFX-base. Importantly, since the above property holds for any $\alpha$, it also holds for any choice of quantile $q_{1-\alpha}$. This property also applies to the localised CP methods described later, which define a different quantile value.

### 3.1 NEED FOR CONDITIONAL GUARANTEES

A visual example of using CONFEX-Naive to generate a counterfactual explanation is shown in Figure 1 (plot b). We observe that when adding the singleton set size constraint, the obtained counterfactual explanation is further from the decision boundary compared to MILP-MinDist (plot a). This is is desirable since the identified CFX would resemble less an adversarial example. However, the counterfactual explanation the identified CFX is somewhat counterintuitive: it lies in an area without local datapoints, i.e., away from the data support (see plot d). Since the CP constraints enforce low-uncertainty predictions, we would expect to find the CFX in a region where datapoints unambiguously belong to the target class, and not in regions near the decision boundary, where multiple classes overlap, or with no or little data support.

The main issue is that CONFEX-Naive can return CFXs that are not exchangeable with the calibration points, violating CP's marginal guarantees. Hence, our prediction regions should be valid *for any* choice of test inputs (not just exchangeable ones), requiring the coverage requirements to be strengthened to enforce *conditional validity*, i.e., for *any* choice of $x = x'$, the following must hold:

$$\mathbb{P}_{\mathcal{D}_{\text{cal}},(x,y)}\left(y \in C_{1-\alpha}(x) \mid x = x'\right) \geq 1 - \alpha. \tag{8}$$

However, unless the inputs are discrete, the above exact conditional guarantees are known to be impossible if we require distribution-free and finite-sample guarantees (Vovk, 2012; Barber et al., 2020). To solve this issue, among the several methods recently proposed for CP with approximate conditional validity (Jung et al., 2022; Hore & Barber, 2023; Ding et al., 2023; Gibbs et al., 2025; Cabezas et al., 2025), we focus on the *localised CP (LCP)* method of Guan (2023), described in the next section. Below, we prove that conditional conformal prediction provides a solution to the uncertainty-aware CFX problem stated in Problem 1.

**Uncertainty-aware CFXs with Conditional Conformal Prediction** Consider the set of acceptable points $A = \{x \mid C_{1-\alpha}(x) = \{y^+\}\}$. If $C_{1-\alpha}$ satisfies the conditional guarantees of Equation 8, then for every $x \in A$, we have that

$$\mathbb{P}_{\mathcal{D}_{\text{cal}},Y|X=x}\left(Y = y^+\right) \geq 1 - \alpha, \tag{9}$$

which is equivalent to stating $\mathbb{P}_{\mathcal{D}_{\text{cal}},X,Y}(Y = y^+ \mid X = x) \geq 1 - \alpha$. This trivially follows from the fact that if $x \in A$, then $C_{1-\alpha}(x) = \{y^+\}$, and so $\mathbb{P}_{\mathcal{D}_{\text{cal}},X,Y}(Y = y^+ \mid X = x) = \mathbb{P}_{\mathcal{D}_{\text{cal}},X,Y}(Y \in C_{1-\alpha}(x) \mid X = x)$.

Thus, in our optimisation problem, we can use the constraint $C_{1-\alpha}(x) = \{y^+\}$ (provided $C_{1-\alpha}$ offers conditional guarantees) to ensure satisfaction of the uncertainty constraint (9). Note that (9) is equal to the chance constraint (5) in Problem 1 except that the probability is over $\mathcal{D}_{\text{cal}}$ too.

On the other hand, CONFEX-Naive uses standard CP, where $C_{1-\alpha}$ offers only marginal coverage, meaning that the constraint $C_{1-\alpha}(x) = \{y^+\}$ does not satisfy Eq. (9) but a weaker form of it:

$$\mathbb{P}_{\mathcal{D}_{\text{cal}},X,Y} \left( Y = y^+ \mid X \in A \right) \geq 1 - \alpha/P(A), \tag{10}$$

where $P(A) = \mathbb{P}_{\mathcal{D}_{\text{cal}},X,Y} (X \in A)$ is the probability that $X$ yields a the singleton region $\{y^+\}^2$. Therefore, with marginal CP, we cannot attain a principled uncertainty control in CFX generation.

## 4 THE CONFEX APPROACH

Our method CONFEX uses Localised Conformal Prediction (LCP) to generate CFXs with more principled, local coverage guarantees. We introduce two variants: CONFEX-LCP, which encodes LCP constraints via MILP, and CONFEX-Tree, which also provides local guarantees via MILP but is more computationally efficient thanks to an offline tree-based representation of the local quantiles.

### 4.1 LOCALISED CONFORMAL PREDICTION (LCP) AND CONFEX-LCP

Localised Conformal Prediction (LCP) (Guan, 2023) relaxes strict conditional coverage (see equation 8) by requiring coverage to hold only within a local neighbourhood around a test input $x^*$. To achieve this, LCP reweights the calibration points as if they were drawn under the localised distribution of $x^*$, thereby restoring exchangeability. The reweighted probabilities are computed by a *localiser kernel* $H : \mathcal{X} \times \mathcal{X} \to [0, 1]$, which measures how "close" $x'$ is to $x$, with $H(x, x) = 1$. In our method, we use the $L_1$-box kernel

$$H(x, x') = \mathbf{1}(\|x - x'\|_1 \leq h), \tag{11}$$

where $h$ is the kernel bandwidth controlling the degree of localisation. For numerical and ordinal features, the $L_1$ distance is computed after normalisation; for categorical features, we require exact matches over all or some categorical features, else $H(x, x') = 0$. Other kernels (e.g., based on infinity norm or Gaussian smoothing) are also possible.

For a test input $x^*$, the local quantile is

$$q_{1-\alpha}^{\text{LCP}}(x^*) = Q_{1-\alpha} \left( \sum_{i=1}^{n} w_i \delta_{s(x_i, y_i)} + w^* \delta_{+\infty} \right), \tag{12}$$

where $w_i = \frac{H(x^*, x_i)}{W}$ for $i = 1, \ldots, n$ and $w^* = \frac{H(x^*, x^*)}{W} = \frac{1}{W}$, with $W = 1 + \sum_{i=1}^{n} H(x^*, x_i)$ being a normalizing factor.

This reweighting step and the resulting prediction region $C_{1-\alpha}^{LCP}(x^*) = \{y : s(x^*, y) \leq q_{1-\alpha}^{\text{LCP}}(x^*)\}$ ensure, for any test point $x^*$, the following approximate conditional guarantee:

$$\mathbb{P}_{\mathcal{D}_{\text{cal}} \sim P_{X,Y}^n, (x,y) \sim P_{X,Y}^*} (y \in C_{LCP, 1-\alpha}(x)) \geq 1 - \alpha, \tag{13}$$

where $P_{X,Y}^n$ is the (product) distribution of the $n$ calibration points, and $P_{X,Y}^* = P_{Y|X} \times P_X^*$ is the localised test distribution, with $P_X^* = P_X \circ H(x^*, X)$ being the distribution of $X$ obtained by applying to $P_X$ the kernel $H$ centered at $x^*$.

**CONFEX-LCP**  We extend CONFEX-Naive by replacing CP regions with LCP regions, yielding more principled and adaptive counterfactual generation. Formally,

$$x_{\text{cf}} \in \arg\min_{x'} \text{dist}(x_0, x') \quad \text{s.t.} \quad C_{1-\alpha}^{LCP}(x') = \{y^+\}, \tag{14}$$

---

[2]The proof is based on rewriting $\mathbb{P} \left( Y \neq y^+ \mid X \in A \right) = 1 - \mathbb{P} \left( Y = y^+ \mid X \in A \right)$ as $\frac{\mathbb{P} \left( Y \neq y^+ \wedge X \in A \right)}{\mathbb{P} \left( X \in A \right)}$ and noticing that the numerator is bounded by $\alpha$.

which enforces $s(x', y^+) \leq q_{1-\alpha}^{LCP}(x')$ and $s(x', y) > q_{1-\alpha}^{LCP}(x')$ for all $y \neq y^+$. Unlike CONFEX-Naive, which uses a single global quantile $\hat{q}$, here the quantile depends on the candidate $x'$, requiring explicit encoding in the MILP formulation (see Algorithm 2 in the Appendix). This introduces additional variables and big-M constraints linear in the calibration set size. Fig. 1 (plot c) shows a CFX computed using CONFEX-LCP.

**Properties.** Thanks to the LCP method, CONFEX-LCP computes quantiles using only points local to the test input $x$, where locality is defined by the L1 kernel. This yields more adaptive and reliable uncertainty estimates than vanilla CP (and CONFEX-Naive), with larger prediction sets in sparse or ambiguous regions, whilst ensuring that counterfactual is grounded with the data, i.e., similar (local) individuals which are correctly predicted to be in the target class. We note that features in the kernel can be assigned different weights based on domain knowledge. The choice of the kernel bandwidth $h$ is application-specific and it allows us to balance between local and marginal coverage.

### 4.2 CONFEX-TREE: FAST VARIANT OF CONFEX-LCP

Due to the increased cost of resolving quantiles using MILP, LCP is infeasible for practical use with large calibration sets.

In this section, we introduce CONFEX-Tree, an efficient alternative formulation of Localised CP which retains formal guarantees. CONFEX-Tree leverages that decision trees are efficiently representable in MILP and uses precomputed local quantiles. While LCP operates at test-time by retaining only the calibration points within distance $h$ of the point, CONFEX-Tree works offline to determine locality constraints: it splits the feature space recursively to obtain local neighbourhoods of calibration points having kernel width of at most $h$.

The construction procedure is inspired by kd-trees (Skrodzki, 2019) and detailed in Algorithm 1. Each leaf specifies a precomputed local quantile using only calibration points within that leaf. From these points, we also compute the midpoint of the smallest enclosing hyper-rectangle. The tree construction ensures that no two points in a leaf can have a bigger $L_\infty$ distance than the kernel bandwidth $h$. Then, each new test point $x'$ is assigned to a leaf of the tree and is associated with the corresponding quantile if $x'$ is within $L_\infty$ distance of $h/2$ from the midpoint, which means that it is within distance of $h$ from any calibration point of that leaf. To handle categorical features, we stratify the dataset by each combination of (all or select) categorical values and generate a tree for each stratum (which is equivalent to first splitting on all categorical features).

The resulting tree is encoded in MILP and used to provide the quantile value for the test point, replacing the LCP regions from CONFEX-LCP . Formally, explanations are derived by solving

$$x_{\text{cf}} \in \arg\min_{x'} \text{dist}(x_0, x') \quad \text{s.t. } C_{1-\alpha}^{\text{Tree}}(x') = \{y^+\}, \tag{15}$$

where $C_{1-\alpha}^{\text{Tree}}$ is constructed using the local tree-based quantiles returned by Algorithm 1.

**Properties of CONFEX-Tree.** The tree constructed by the CONFEX-Tree defines a partitioning of the feature space into disjoint regions $\{\mathcal{X}_g\}_{g \in \mathcal{G}}$. Each $g$ has an associated quantile value $q_{1-\alpha,g}$ computed using only calibration points in $g$. This results in the following finite-sample group-conditional coverage guarantee

$$\mathbb{P}\big(y \in C_{1-\alpha}^{\text{Tree}}(x^*) \mid x^* \in \mathcal{X}_g\big) \geq 1 - \alpha \qquad \text{for all } g \in \mathcal{G}, \tag{16}$$

as per Vovk (2012). Note that our method overapproximates the group-conditional quantiles as it assigns a quantile of $\infty$ when $x^*$ has $L_\infty$ distance more than $h/2$ from the midpoint of $g$. For this reason, it still satisfies the above guarantee.

Moreover, by construction, the groups created by CONFEX-Tree are local regions of calibration points in the feature space. Hence, we obtain an approximate conditional guarantee, as the tree approximates the conditional quantile $Q_{1-\alpha}(s|x)$ with the granularity of the approximation being controlled by the bandwidth $h$.

Finally, CONFEX-Tree can be viewed as an instance of LCP using the following kernel

$$H(x, x') = \mathbf{1}(\|x - x'\|_\infty \leq h \land \exists g.x, x' \in \mathcal{X}_g), \tag{17}$$

---

**Algorithm 1:** CONFEX-Tree: Tree-based encoding of local quantiles

---

**Input** : Calibration set $\mathcal{D}_{\text{cal}}$, score function $s$, coverage level $1 - \alpha$, bandwidth $h$
**Output:** Tree-based quantile encoding
**Categorical Stratification**:

1. Stratify the calibration dataset by each distinct combination of (all or some) categorical feature values.

2. Generate a tree for each group using the Tree Construction procedure over the normalised numerical and ordinal values only.

**Tree Construction:**

1. If the maximum range along any feature dimension of all calibration points in the node is less than $h$, stop and create a leaf node. At each leaf, compute and store:
   - the $1 - \alpha$ quantile of the scores $s(x, y)$ of the calibration points assigned to the leaf;
   - the midpoint of the calibration features in the leaf.

2. Otherwise, split the current node along the feature with the maximum spread, using the midpoint of that feature's values as the split point. Recurse on the left and right subsets to build subtrees.

**Prediction for test point $x^*$:**

1. Select the correct tree based on the test point's categorical values.

2. Traverse the tree using $x'$ until reaching a leaf. Let $c$ and $q$ be its stored midpoint and quantile.

3. Reject point if assigned to the leaf but not local: if $\|x^* - c\|_\infty > h/2$, return $\infty$; o/w, return $q$.

---

i.e., both points need to belong to the same leaf and have $L_\infty$ distance bounded by $h$. Using this kernel, the guarantees of equation 13 also apply to CONFEX-Tree.

To summarise, explanations produced by CONFEX-Tree enjoy a distance optimality guarantee, validity guarantee (Relation with MILP-MinDist), and uncertainty guarantee Eq. (16). Additionally, our local (i.e., approx. conditional) guarantees imply that our CFXs are valid with high probability for any individual, even for out-of-distribution ones. This is preferable to a generator which, over a test set, empiricially produces good results over a particular metric, since the distribution may shift at test-time - our method is robust to this.

## 5 EVALUATION

In this section, we evaluate our method against competing CFX methods, assessing the cost (distance), plausibility and sensitivity of CFXs generated by CONFEX-Tree. We explore the impact of varying the kernel bandwidth and the user-specified coverage rate, and we verify the formal coverage guarantees of CONFEX methods. We find that CONFEX consistently produces more stable and plausible CFXs across the benchmarks, provided the kernel bandwidth is appropriately chosen.

**Experimental setup** For our experiments, two classes of models are considered: multi-layer perceptrons (MLPs) and random forests (RFs). We selected four tabular datasets commonly found in the CFX literature: AdultIncome (Becker & Kohavi, 1996), CaliforniaHousing (Pace & Barry, 1997), GiveMeSomeCredit and GermanCredit (Hofmann, 1994), using a training-calibration-test split of 60%-20%-20% for each.

To evaluate CONFEX, we compare our efficient tree-based approach CONFEX-Tree (CTree) against competing uncertainty-aware generators: ECCCo (Altmeyer et al., 2024), the only other CFX method which uses CP, and a modified version of Schut (Schut et al., 2021) (called 'Greedy' in our table) which uses a single MLP instead of an ensemble, as well as the Wachter et al. (2017) baseline. We also consider plausibility-targeting generators FACE Poyiadzi et al. (2020) and C-CHVAE Pawelczyk et al. (2020). For tree-based models, we compare against the popular methods FeatureTweak (FT) (Tolomei et al., 2017), which searches for possible paths which can change the classification, and FOCUS (Lucic et al., 2021), which optimises for distance over a differentiable relaxation of the tree models. As baselines, we include MILP-MinDist (MinDist) and CONFEX-

| | CaliforniaHousing | | | GermanCredit | | |
|---|---|---|---|---|---|---|
| | Distance | Plausibility | Sens $(10^{-1})$ | Distance | Plausibility | Sens $(10^{-1})$ |
| **Multi-Layer Perceptron** | | | | | | |
| MinDist | **0.03 ± 0.00** | 0.30 ± 0.07 | 42.75 ± 8.5 | 1.65 ± 0.18 | 0.54 ± 0.19 | 0.08 ± 0.02 |
| ECCCo | 0.37 ± 0.02 | -0.65 ± 0.05 | 0.26 ± 0.05 | 0.97 ± 0.06 | 0.16 ± 0.11 | 0.06 ± 0.02 |
| Greedy | 1.88 ± 0.27 | -0.99 ± 0.02 | 0.14 ± 0.02 | 0.99 ± 0.04 | -0.03 ± 0.09 | 0.08 ± 0.02 |
| Wachter | 0.09 ± 0.01 | 0.42 ± 0.08 | 1.66 ± 0.37 | **0.41 ± 0.02** | 0.73 ± 0.05 | 0.33 ± 0.09 |
| FACE | 0.21 ± 0.02 | **0.85 ± 0.04** | 0.34 ± 0.03 | 0.69 ± 0.05 | 0.92 ± 0.05 | 0.05 ± 0.01 |
| C-CHVAE | 1.27 ± 0.22 | -0.35 ± 0.12 | 0.06 ± 0.03 | 2.45 ± 0.13 | 0.80 ± 0.19 | 0.07 ± 0.01 |
| CNaive | 0.04 ± 0.01 | 0.24 ± 0.08 | 8.41 ± 2.26 | 2.00 ± 0.07 | 0.16 ± 0.28 | 0.05 ± 0.01 |
| CTree | 0.55 ± 0.04 | 0.75 ± 0.05 | **0.05 ± 0.05** | 2.31 ± 0.20 | **1.00 ± 0.00** | **0.01 ± 0.01** |
| **Random Forest** | | | | | | |
| MinDist | **0.01 ± 0.00** | 0.37 ± 0.07 | 89.15 ± 90.9 | 1.65 ± 0.06 | 0.52 ± 0.17 | 0.09 ± 0.01 |
| FT | 0.12 ± 0.03 | 0.29 ± 0.25 | 0.58 ± 0.16 | 0.50 ± 0.06 | 0.84 ± 0.05 | 0.09 ± 0.01 |
| FOCUS | 0.11 ± 0.01 | 0.34 ± 0.09 | 5.21 ± 2.31 | **0.45 ± 0.14** | 0.83 ± 0.02 | 0.58 ± 0.25 |
| FACE | 0.17 ± 0.01 | **0.81 ± 0.02** | 0.46 ± 0.08 | 0.59 ± 0.06 | 0.88 ± 0.07 | 0.07 ± 0.01 |
| CNaive | 0.03 ± 0.01 | 0.42 ± 0.07 | 12.34 ± 2.94 | 1.62 ± 0.08 | 0.63 ± 0.10 | 0.09 ± 0.01 |
| CTree | 0.19 ± 0.02 | 0.61 ± 0.10 | **0.40 ± 0.18** | 2.04 ± 0.16 | **1.00 ± 0.00** | **0.01 ± 0.01** |

Table 1: Results for CaliforniaHousing and GermanCredit datasets. The best result for each generator over its hyperparameters is reported. See Table 3 in Section A.2 for full results, further discussion, and $p$-values for significance of results.
Validity 58% for FT in CaliforniaHousing. For GermanCredit, validity is 50% for FT, 84% for Wachter, 82% for Schut, 84% for ECCCo, 74% for C-CHVAE. This explains why some methods seem to attain smaller distances than MinDist, which is always valid.

Naive (CNaive). As discussed previously, CONFEX-LCP is very expensive due to its "direct" (and inefficient) quantile encoding, hence, we did not conduct extensive experiments for it. Instead, we include a scalability analysis and comparison of CONFEX-LCP and CONFEX-Tree in Section C.

**Metrics** To evaluate the CFXs, we focus on two main dimensions: plausibility and sensitivity. *Plausibility* evaluates whether counterfactuals lie close to the data distribution, and is measured with the Local Outlier Factor (LOF) stratified per target class, with higher scores indicating more realistic examples. *Sensitivity* (Sens) captures robustness to small perturbations of the input instance $x$; counterfactuals with low sensitivity remain consistent under such perturbations.

We run our experiments over five repeats: in each repeat, we train a our models and for each model and generator, we compute metrics from 100 generated CFXs for factual points taken from the test set, plus an additional 100 for the sensitivity metric. The metrics obtained are then computed and averaged to ensure statistical reliability. We also record the distance, implausibility, stability, and validity of the method. Further details on the metrics and experimental setup can be found in the appendix.

**Evaluation of conformal guarantees** In the main setup, CFXs are generated for each test instance, but since their ground truth is unknown, coverage cannot be computed. We therefore run an additional simulated setup, identical to CONFEX in that it finds the *closest test point* whose CP region is a singleton comprising the target class. This way, true labels are known and we can compute the empirical coverage $\mathbb{E}(\mathbf{1}(y \in C_{1-\alpha}(x)))$ over this resampling of the test set. We measure the gap between the observed coverage and the target $1 - \alpha$. Note that this resampling considers only CFX-like points and hence breaks exchangeability. So, we expect CONFEX-Naive to miss the coverage target and the localised procedures to fare better.

**Results discussion** In Table 1, we observe that CONFEX-Tree consistently outperforms competing uncertainty-aware methods by producing, in many cases more plausible and less sensitive explanations. This is in contrast to CONFEX-Naive which shows substantially lower plausibility and higher sensitivity, validating the issues illustrated in Figure 1 and further motivating the use

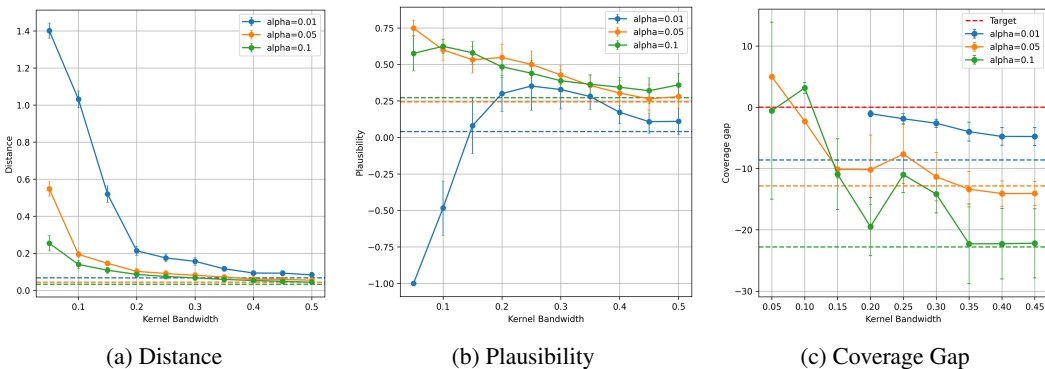

(a) Distance        (b) Plausibility        (c) Coverage Gap

Figure 2: Effect of coverage rate and kernel bandwidth on metrics for CONFEX-Tree on the CaliforniaHousing dataset. CONFEX-Naive is represented by dashed horizontal lines.

of localisation in CP. In terms of plausibility, we find that CONFEX-Tree performs comparably to generator FACE and outperforms C-CHVAE and ECCCo, which explicitly target this metric. In the appendix, we show that our method provides more certain explanations than competing generators, and include results for the AdultIncome and GiveMeSomeCredit datasets.

Fig. 16 illustrates the effect of varying the kernel bandwidth and coverage rate in the CONFEX-Tree method. Increasing the coverage rate $1 - \alpha$ leads to larger distances, since prediction sets become more conservative and singleton regions less frequent. Larger bandwidths yield shorter distances but at the cost of lower plausibility, as the notion of locality becomes weaker[3]. These observations are consistent with the fact that, as the kernel bandwidth grows, localised CP converges to standard marginal CP, as seen with CONFEX-Naive in the figures.

In the (simulated) CFX setting, the Coverage Gap results confirm that vanilla CP (used by CONFEX-Naive) fails to reach the target coverage, while localised CP with a suitably chosen kernel bandwidth succeeds. For small bandwidths (i.e., "strong" locality), all three choices of $\alpha$ attain or are close to the target coverage level, but the gap grows as the bandwidth increases and localisation diminishes. For $\alpha = 0.01$ and small bandwidths, no data is obtained since no test points produced a singleton prediction region (as required by our CONFEX constraints). These figures demonstrate that picking a correct bandwidth is crucial for obtaining good plausibility and coverage guarantees.

## 6 CONCLUSIONS

We introduced a novel MILP-based framework for generating uncertainty-aware counterfactual explanations with formal, distribution-free guarantees. By developing an efficient encoding of localised conformal prediction, we address the critical issue of exchangeability violation in the CFX search process. This allows us to enforce approximate test-conditional guarantees, ensuring the generation of provably reliable, plausible, and robust explanations.

**Limitations** Since our approach uses MILP to solve for CFXs, it will struggle scaling to very large models; gradient-based methods like Wachter and ECCCo are less prone to this problem, but they sacrifice guarantees on CFX validity. Additionally, unlike gradient-based methods, our method can be used on random forest and gradient-boosted trees, which remain competitive on tabular datasets.

Moreover, CP requires a held-out calibration dataset, which may be problematic when data is scarce. Fortunately, CP guarantees hold regardless of the calibration set size (but small sets will lead to more conservative prediction regions).

Picking an appropriate kernel bandwidth is an additional task which requires domain knowledge or evaluation on a validation set, for example, with the coverage gap simulation described in the previous section.

---

[3]For very small $\alpha$ (0.01) and small kernel bandwidths, we observe low plausibility: we conjecture this could be due to the CP method localising on outlier points.

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

# A  APPENDIX

**AI Use Declaration**   The authors acknowledge the use of Generative AI to minimally polish text.

## A.1  RELATED WORKS

Our work integrates three research areas: counterfactual explanations (CFXs), uncertainty quantification in explanations, and the application of conformal prediction (CP) to optimization problems. Counterfactual explanations, introduced by Wachter et al. (2017), provide recourse by identifying minimal feature changes to alter a model's prediction. While initial work focused on validity and distance, the field has expanded to include desiderata like plausibility and actionability (Verma et al., 2020; Karimi et al., 2021). Methodologies have also diversified from gradient-based optimization to tree-specific algorithms (Tolomei et al., 2017; Lucic et al., 2021) and constraint-based methods using Mixed-Integer Linear Programming (MILP) (Kanamori et al., 2020). However, a critical limitation of many approaches is their failure to account for model uncertainty, which can result in misleading or brittle explanations (Schut et al., 2021). To address this, prior works have employed Bayesian methods (Antorán et al., 2020) or model ensembles (Schut et al., 2021). CONFEX contributes a novel, principled alternative by using Conformal Prediction. More relevant is ECCCo (Altmeyer et al., 2024), which uses a loss term based on the conformal set size (Stutz et al., 2022) but crucially does not address the violation of the exchangeability assumption inherent in the CFX search process.

## A.2  FURTHER DISCUSSION OF TABLE 1

For GermanCredit, whilst Wachter obtained the closest counterfactuals, had a validity rate of 84%, demonstrating how gradient-based methods may fail to correctly change prediction to the target class. ECCCo (84%) and FeatureTweak (50%) also suffered validity issues. On the other hand, MILP-MinDistalways found a valid counterfactual, including satisfying correct categorical and ordinal encoding unlike some of the competing tree generators, and this is reflected with an increased distance. Note that in all figures, kernel bandwidth is measured as a multiple of the median pairwise distance between all points in the dataset.

Extended versions of Table 1 are included as the following two tables.

In each cell, the first line is the mean value of the metric over 5 repeats, the second line is the standard deviation over those repeats, and the third line is the the $p$-value for a two-tailed (paired per repeat) t-test to check whether the mean value of the generator's metric is significantly different from the mean value of the best performing generator. Note that the values in the table, including $p$-values, are over valid results only: see the list below.

- For CaliforniaHousing, validity is 58% for FeatureTweak.
- For GermanCredit, validity is 50% for FT, 84% for Wachter, 82% for Schut, 84% for ECCCo, 74% for C-CHVAE.
- For GiveMeSomeCredit, validity is 71% for Wachter, 80% for Schut, 50% for FeatureTweak.
- For AdultIncome, Validity 80% for Wachter, 85% for ECCCo, 54% for C-CHVAE, 61% for FeatureTweak

Full results, including details on the Certainty metric "Cert" and for full results.

| | CaliforniaHousing | | | | GermanCredit | | | |
|---|---|---|---|---|---|---|---|---|
| | Dist | Plaus | Sens | Cert | Dist | Plaus | Sens | Cert |
| **Multi-Layer Perceptron** | | | | | | | | |
| MinDist | **0.03** ±0.00 (-) | 0.30 ±0.07 (2.81e-5) | 42.75 ±8.45 (5.47e-4) | 2.44e-4 ±2.62e-4 (4.83e-3) | 1.65 ±0.18 (1.19e-4) | 0.54 ±0.19 (8.54e-3) | 0.08 ±0.02 (5.03e-3) | 0.140 ±1.23e-2 (1.08e-4) |
| ECCCo | 0.37 ±0.02 (9.44e-7) | -0.65 ±0.05 (1.14e-6) | 0.26 ±0.05 (4.39e-3) | 0.00e+00 ±0.00e+00 (4.78e-3) | 0.97 ±0.06 (9.72e-6) | 0.16 ±0.11 (1.16e-4) | 0.06 ±0.02 (7.32e-3) | 0.173 ±4.51e-2 (2.77e-3) |
| Greedy | 1.88 ±0.27 (1.60e-4) | -0.99 ±0.02 (1.52e-7) | 0.14 ±0.02 (0.037) | 0.00e+00 ±0.00e+00 (4.78e-3) | 0.99 ±0.04 (5.00e-5) | -0.03 ±0.09 (2.44e-5) | 0.08 ±0.02 (2.91e-3) | 0.101 ±3.58e-2 (3.44e-4) |
| Wachter | 0.09 ±0.01 (7.98e-5) | 0.42 ±0.08 (1.29e-4) | 1.66 ±0.37 (8.20e-4) | 3.58e-2 ±1.24e-2 (4.78e-3) | **0.41** ±0.02 (-) | 0.73 ±0.05 (3.76e-4) | 0.33 ±0.09 (2.35e-3) | -8.07e-3 ±1.25e-2 (3.19e-5) |
| FACE | 0.21 ±0.02 (4.23e-6) | **0.85** ±0.04 (-) | 0.34 ±0.03 (1.12e-3) | 5.40e-3 ±4.46e-3 (5.32e-3) | 0.69 ±0.05 (3.74e-4) | 0.92 ±0.05 (4.07-2) | 0.05 ±0.01 (1.26e-3) | 0.313 ±3.71e-2 (1.26e-2) |
| C-CHVAE | 1.27 ±0.22 (3.49e-4) | -0.35 ±0.12 (3.37e-5) | 0.06 ±0.03 (0.79) | 0.00e+00 ±0.00e+00 (4.78e-3) | 2.45 ±0.13 (5.28e-6) | 0.80 ±0.19 (0.10) | 0.07 ±0.01 (2.66e-4) | 8.17e-2 ±5.95e-2 (2.41e-4) |
| CNaive | 0.04 ±0.01 (7.10e-4) | 0.24 ±0.08 (2.68e-5) | 8.41 ±2.26 (1.73e-3) | 3.26e-2 ±9.71e-3 (6.67e-3) | 2.00 ±0.07 (1.26e-6) | 0.16 ±0.28 (3.79e-3) | 0.05 ±0.01 (5.47e-3) | 0.213 ±3.03e-2 (2.76e-4) |
| CTree | 0.55 ±0.04 (7.50e-6) | 0.75 ±0.05 (3.00e-2) | **0.05** ±0.05 (-) | **0.101** ±3.57e-2 (-) | 2.31 ±0.20 (3.66e-5) | **1.00** ±0.00 (-) | **0.01** ±0.01 (-) | **0.483** ±4.42e-2 (-) |
| **Random Forest** | | | | | | | | |
| MinDist | **0.01** ±0.00 (-) | 0.37 ±0.07 (2.05e-4) | 89.15 ±90.9 (0.12) | 1.78e-2 ±4.18e-3 (2.45e-4) | 1.65 ±0.06 (4.52e-5) | 0.52 ±0.17 (4.88e-3) | 0.09 ±0.01 (2.02e-4) | 4.84e-2 ±2.14e-2 (2.37e-2) |
| FT | 0.12 ±0.03 (1.27e-3) | 0.29 ±0.25 (1.12e-2) | 0.58 ±0.16 (0.29) | 4.68e-3 ±6.00e-3 (1.32e-4) | 0.50 ±0.06 (0.52) | 0.84 ±0.05 (2.17e-3) | 0.09 ±0.01 (6.76e-4) | 4.27e-2 ±4.41e-2 (1.09e-2) |
| FOCUS | 0.11 ±0.01 (1.21e-4) | 0.34 ±0.09 (6.20e-4) | 5.21 ±2.31 (1.51e-2) | 1.96e-2 ± 7.52e-3 (2.07e-4) | **0.45** ±0.14 (-) | 0.83 ±0.02 (7.24e-5) | 0.58 ±0.25 (1.11e-2) | 0.134 ± 7.82e-3 (6.00e-2) |
| FACE | 0.17 ±0.01 (8.05e-6) | **0.81** ±0.02 (-) | 0.46 ±0.08 (0.60) | 3.79e-2 ±8.69e-3 (4.21e-4) | 0.59 ±0.06 (4.59e-2) | 0.88 ±0.07 (3.70e-2) | 0.07 ±0.01 (3.81e-4) | 0.04 ±0.01 (7.10e-1) |
| CNaive | 0.03 ±0.01 (3.42e-3) | 0.42 ±0.07 (5.67e-4) | 12.34 ±2.94 (1.34e-3) | 8.34e-2 ±1.54e-2 (5.17e-3) | 1.62 ±0.08 (1.05e-5) | 0.63 ±0.10 (1.54e-3) | 0.09 ±0.01 (4.94e-4) | 0.175 ±6.52e-2 (1.95e-1) |
| CTree | 0.19 ±0.02 (2.17e-5) | 0.61 ±0.10 (1.71e-2) | **0.40** ±0.18 (-) | **0.143** ±1.83e-2 (-) | 2.04 ±0.16 (1.35e-4) | **1.00** ±0.00 (-) | **0.01** ±0.01 (-) | **0.335** ± 0.148 (-) |

Table 2: Extended version of Table 1. CFX generation results for CaliforniaHousing and German-Credit, including mean and standard deviation of metric value over 5 runs, and $p$-value of a t-test to check whether the mean value of the generator's metric is significantly different from the mean value of the best performing generator. See Section A.2 for more detail, including on validities.

| | GiveMeSomeCredit | | | | AdultIncome | | | |
|---|---|---|---|---|---|---|---|---|
| | Dist | Plaus | Sens | Cert | Dist | Plaus | Sens | Cert |
| **Multi-Layer Perceptron** | | | | | | | | |
| MinDist | **0.03** ±0.00 (-) | 0.93 ±0.04 (3.12e-2) | 1.39 ±0.57 (9.67e-3) | 6.00e-3 ±5.64e-3 (6.42e-2) | 1.16 ±0.07 (6.20e-5) | -0.13 ±0.05 (1.04e-3) | 0.11 ±0.11 (0.35) | 2.11e-2 ±1.33e-2 (1.78e-3) |
| ECCCo | 0.69 ±0.28 (8.87e-3) | -0.97 ±0.03 (2.69e-8) | 0.21 ±0.06 (7.09e-3) | 0.00e+00 ±0.00e+00 (5.80e-2) | 0.73 ±0.11 (2.69e-2) | -0.05 ±0.07 (1.07e-3) | 0.05 ±0.00 (0.719) | 9.42e-3 ±9.68e-3 (1.92e-3) |
| Greedy | 0.13 ±0.07 (4.53e-2) | -0.02 ±0.38 (6.37e-3) | 1.05 ±0.81 (8.13e-2) | 6.90e-3 ±4.06e-3 (7.53e-2) | 0.95 ±0.08 (2.64e-3) | 0.02 ±0.10 (4.18e-3) | 24190 ±48381 (0.374) | -5.43e-3 ±2.44e-2 (1.20e-3) |
| Wachter | 0.09 ±0.01 (3.42e-4) | 0.93 ±0.02 (4.64e-3) | 0.95 ±0.07 (1.51e-5) | 0.00e+00 ±0.00e+00 (5.80e-2) | **0.43** ±0.09 (-) | 0.28 ±0.05 (0.424) | 0.21 ±0.08 (2.23e-2) | -2.91e-3 ±1.03e-2 (7.85e-4) |
| FACE | 0.12 ±0.01 (7.66e-6) | 0.95 ±0.02 (3.49e-2) | 0.35 ±0.04 (2.76e-4) | 1.92e-2 ±1.52e-2 (8.98e-2) | 1.36 ±0.16 (1.99e-4) | **0.34** ±0.12 (-) | 0.06 ±0.02 (0.428) | 0.144 ±1.71e-2 (4.32e-2) |
| C-CHVAE | 1.32 ±0.08 (5.38e-6) | -0.92 ±0.06 (3.69e-7) | **0.09** ±0.02 (-) | 6.67e-4 ±1.33e-3 (5.86e-2) | 6.39 ±0.55 (2.05e-5) | 0.33 ±0.15 (0.847) | 0.04 ±0.01 (0.448) | 2.26e-2 ±1.22e-1 (6.90e-2) |
| CNaive | 0.04 ±0.00 (7.87e-4) | 0.76 ±0.05 (2.08e-3) | 0.78 ±0.11 (6.41e-4) | 4.86e-3 ±4.56e-3 (6.28e-2) | 1.24 ±0.07 (2.10e-4) | -0.13 ±0.04 (1.90e-3) | 0.05 ±0.01 (0.876) | 6.28e-2 ±2.71e-2 (2.62e-3) |
| CTree | 0.24 ±0.06 (1.76e-3) | **0.98** ±0.01 (-) | 0.14 ±0.03 (1.41e-2) | **7.75e-2** ± 5.88e-2 (-) | 1.78 ±0.19 (2.98e-4) | -0.02 ±0.14 (2.91e-2) | **0.05** ±0.02 (-) | **0.210** ±5.06e-2 (-) |
| **Random Forest** | | | | | | | | |
| MinDist | **0.01** ±0.00 (-) | 0.96 ±0.01 (4.64e-3) | 96.79 ±35.89 (5.81e-3) | 1.10e-2 ±6.28e-3 (5.49e-4) | 0.96 ±0.03 (7.56e-2) | 0.03 ±0.07 (1.44e-3) | 0.14 ±0.03 (5.97e-2) | 5.03e-2 ±1.28e-2 (7.97e-4) |
| FT | 0.03 ±0.01 (9.17e-4) | 0.96 ±0.02 (1.61e-2) | 1.40 ±0.16 (6.45e-4) | 1.09e-3 ±9.31e-3 (3.55e-4) | 0.24 ±0.10 (0.152) | 0.30 ±0.11 (2.69e-2) | 0.06 ±0.01 (0.862) | 4.68e-2 ± 7.16e-2 (6.77e-3) |
| FOCUS | 0.05 ±0.00 (4.38e-5) | 0.91 ±0.05 (3.76e-2) | 2.27 ±0.79 (1.29e-2) | 9.82e-3 ±7.81e-3 (1.46e-3) | **0.58** ±0.34 (-) | 0.40 ±0.06 (0.795) | 0.29 ±0.14 (0.0196) | 0.153 ±1.44e-2 (8.34e-3) |
| FACE | 0.10 ±0.01 (9.25e-6) | 0.97 ±0.01 (3.41e-2) | **0.46** ±0.08 (-) | 5.45e-2 ±9.73e-3 (1.02e-2) | 1.50 ±0.07 (6.27e-3) | **0.39** ±0.07 (-) | 0.06 ±0.02 (0.989) | 2.21e-1 ±2.63e-2 (7.01e-2) |
| CNaive | 0.01 ±0.00 (1.50e-3) | 0.96 ±0.01 (3.88e-3) | 28.68 ±25.01 (8.76e-2) | 3.20e-2 ±1.38e-2 (3.05e-3) | 0.97 ±0.09 (1.62e-3) | 0.00 ±0.10 (9.69e-4) | 0.12 ±0.02 (0.108) | 2.43e-1 ±5.17e-2 (0.169) |
| CTree | 0.07 ±0.01 (6.85e-5) | **0.99** ±0.01 (-) | 0.55 ±0.14 (8.43e-2) | **8.56e-2** ±1.21e-2 (-) | 1.56 ±0.21 (2.85e-3) | 0.08 ±0.15 (1.17e-2) | **0.06** ±0.03 (-) | **0.311** ±6.12e-2 (-) |

Table 3: Extended version of Table 1. CFX generation results for GiveMeSomeCredit and Adult-Income, including mean and standard deviation of metric value over 5 runs, and $p$-value of a t-test to check whether the mean value of the generator's metric is significantly different from the mean value of the best performing generator. See Section A.2 for more detail, including on validities.

# B MILP FORMULATION DETAILS

The full CONFEX model optimises the following problem:

$$\text{argmin}_{x'} \qquad \|x_0 - x'\|_1$$

subject to      encoding validity constraints (C1-C5)

classifier constraints (C6)

conformal quantile constraints (Alg 1-2)

conformal singleton set constraints (C7-14)

optional further constraints (C15)

where $x_0$ is the factual instance and $x'$ is the counterfactual explanation returned by the optimisation.

**Encoding validity constraints** The obtained counterfactual explanation must follow correct numeric/categorical/ordinal encoding of the dataset.

Let the indices $i = 0, 1, 2, N$ of the $N$-length vector $x'$ be partitioned into $I_{num}$, indices of numeric variables, $I_{ord}$, indices of ordinally encoded variables, and $I_{cat}^1, I_{cat}^2, \ldots, I_{cat}^C$, which are index groups for each of $C$ one-hot categorically encoded variables. It is possible that some of these sets are empty.

For each numeric variable, we require that the variable is within bounds. Let $l[i]$ and $u[i]$ represent the lower and upper bound of feature $i$.

$$x'[i] \geq l[i] \qquad \text{for all } i \in I_{num} \text{ where } l[i] \neq -\inf \qquad \text{(C1)}$$

$$x'[i] \leq u[i] \qquad \text{for all } i \in I_{num} \text{ where } u[i] \neq +\inf \qquad \text{(C2)}$$

For ordinal features, we must encode that $x'[i] \in v[i]$, where $v[i]$ is the set of possible ordinal values that $x'[i]$ can take.

Add $|v[i]|$ binary indicator variables $V_{i,1}, V_{i,2}, \ldots, V_{i,|v[i]|}$ for each $i \in I_{ord}$

corresponding to possible ordinal values $v_{i,1}, v_{i,2}, \ldots, v_{i,|v[i]|}$

$$\sum_{j=1}^{|v[i]|} V_{i,j} = 1 \text{ for all } i \in I_{ord} \qquad \text{(C3)}$$

$$x'[i] = \sum_{j=1}^{|v[i]|} V_{i,j} v_{i,j} \text{ for all } i \in I_{ord} \qquad \text{(C4)}$$

For each group of one-hot encoded categorical features $I_{cat}^1, I_{cat}^2, \ldots I_{cat}^C$, we must encode that $x'[i] = 1$ for one $i$ in $I_{cat}^c$ and $x'[i] = 0$ for all $j \in I_{cat}^c, j \neq i$.

Add $|I_{cat}^c|$ binary indicator variables $C_{c,1}, C_{c,2}, \ldots, C_{c,|I_{cat}^c|}$ for each $c \in \{1, \ldots, C\}$

corresponding to each entry in the one-hot feature $i_1^c, i_2^c, \ldots, i_{|I_{cat}^c|}^c \in I_{cat}^c$

$$\sum_{j=1}^{|I_{cat}^c|} i_j^c = 1 \text{ for all } c \in \{1, \ldots, C\} \qquad \text{(C5)}$$

**Classifier constraints** To encode the classifier prediction $f(x')$ of the factual $x'$ we repurpose the core components of the `gurobi-machinelearning` library (originally designed for encoding regressors) to instead produce an MILP encoder of Neural Network and Random Forest classifers.

In constraint Eq. (C6), the model prediction is constrained to the variable $y'$.

$$y' = f(x') \tag{C6}$$

Note that in the MILP-MinDistmethod, we constrain $y'$ to be the target class $y^+$. This is done by checking if the output logit for the correct class is larger than all other classes.

$$y'_{y^+} > y'_i \text{ for all } i \neq y^+ \tag{C6b}$$

However, in the CONFEX method, this explicit constraint Eq. (C6b) is not required as explained in Section 3, we only require Eq. (C6).

### B.1 MILP ENCODING OF LOCALISED CP

**MILP encoding of CONFEX-LCP**   The following algorithm Algorithm 2 computes the LCP quantile value in MILP. To do this, all calibration scores and calibration points must be accessible to the optimiser. Variables are constrained as distances from the test point to each calibration point, and another set of variables compute the corresponding weight according to the L1 kernel. These weights are used alongside calibration scores to identify the desired weighted quantile. This encoding is linear in the size of the calibration set.

---

**Algorithm 2:** Localised CP constraints in MILP

---

**Input** : Calibration dataset $\{(x_i, y_i)\}_{i=1}^n$, corresponding scores $\{s_i\}_{i=1}^n$, test input $x^*$, L1
          localisation kernel with bandwidth $h$, level $\alpha \in (0, 1)$
**Output:** Local quantile $q_{1-\alpha}^{LCP}$

1 Sort $\{(x_i, y_i)\}_{i=1}^n$ in ascending order w.r.t. scores.
2 Add $n$ real variables $d_1, \ldots, d_n$.
3 For $i = 1, \ldots, n$, add the L1 distance constraint $d_i = \|x_i - x^*\|_1$.
4 Add $n$ binary variables $w_1, \ldots, w_n$ as the weights induced by the L1 kernel.
5 For $i = 1, \ldots, n$, add the constraint $w_i = \mathbf{1}(d_i \leq h)$, implemented for arbitrarily large $M > 0$
   as
$$d_i \leq h + M(1 - w_i) \wedge d_i \geq h - Mw_i$$
6 Add $n$ binary variables $in_1, \ldots, in_n$; each $in_i$ keeps track if the score $s_i$ is below the quantile.
7 Add integer variables $W$ and $W_{1-\alpha}$ denoting, respectively, the sum of all weights and of those
   weights whose score is below the quantile.
8 Add constraints $W = \sum_{i=1}^n w_i$, $W_{1-\alpha} = \sum_{i=1}^n in_i \cdot w_i$ and $W_{1-\alpha} \geq \lceil (1-\alpha)W \rceil$. The latter
   expresses that the scores below the quantile have probability at least $1 - \alpha$.
9 Define $W'_{1-\alpha} = \sum_{i=1}^n (1 - in_i) \cdot w_i$ and add constraint $W'_{1-\alpha} \geq \lfloor \alpha W \rfloor$
10 Solve constraints and return $s_k$.
11 $q_{1-\alpha}^{LCP}$ will be the largest calibration score $s_i$ for which $in_i = 1$. To identify it, add an integer
    variable $k \in \{1, \ldots, n\}$.
12 For $i = 1, \ldots, n$, add the constraint $in_i = \mathbf{1}(i \leq k)$ using a big-M encoding as done in line 5.

---

**MILP encoding of CONFEX-Tree**   Following the Tree Construction procedure in Algorithm 1, we obtain a family of decision trees. Each tree contains with leaf nodes holding a centre midpoint $m$ and quantile $q$, these are concatenated as a single vector $[c_1, c_2, \ldots, c_D, q]$.

The MILP encoding procedure consists of encoding the decision tree in the MILP problem using the `gurobi-machinelearning` library, selecting the correct tree T to use based on the categorical values of $x'$, identifying the leaf of T corresponding to $x'$, to obtain or reject the quantile based on the midpoint and distance.

Let there be $N_T$ trees, each corresponding to a particular categorical combination. Note that certain categorical features can be ignored, or different categorical values can be considered the same, in order to reduce the number of trees required to be encoded without sacrificing any formal guarantees. This corresponds to a different notion of similarity in the LCP kernel.

---

**Algorithm 3:** CONFEX-Tree constraints in MILP

---

**Input** : Trees $T_1, \ldots, T_{N_T}$ with corresponding categorical indicators $\theta_1, \theta_2, \ldots, \theta_{N_T}$, produced by Algorithm 1, bandwidth $h$, test input $x'$

**Output:** CONFEX-Tree local quantile $q_{1-\alpha}^{\text{Tree}}$

**1 Encoding of trees**

**2** Add $N_T$ vector variables $t_1, t_2, \ldots, t_{N_T}$ of shape (1 + length of $x'$), for the output of each tree.

**3** Let $x'_{\text{noncat}}$ be the vector $x'$ excluding all categorically encoded entries.

**4** Constrain $t_i = T_1(x'_{\text{noncat}})$ for all $i \in 1, \ldots, N_T$

**5 Selection of tree**

**6** Add $N_T$ binary indicators $\tau_1, \tau_2, \ldots, \tau_{N_T}$ to determine the active tree.

**7** Constrain $\sum_{i=1}^{N_T} \tau_i = 1$

▷ *The following constraints determine if tree $T_i$ is active by considering values of $\theta_i$, which are the indices of one-hot entries in $x'$ which should be 1 if the tree $T_i$ is selected.*

**8** For each $i$, constrain $\tau_i \leq x'[j]$ for all $j \in \theta_i$

▷ *Ensure $\theta_i$ is 0 if any one-hot entries corresponding to the tree is 1. $|\theta_i|$ is the number of categorical features.*

**9** For each $i$, constrain $\tau_i \geq \sum_{j \in \theta_i} x'[j] - |\theta_i| + 1$ ▷ *Ensure $\theta_i$ is 1 if all one-hot entries corresponding to the tree is 0*

**10** Obtain the tree output $t$ as $t = \sum_{i=1}^{N_T} t_i \tau_i$

**11 Obtaining of quantile**

▷ *Note that in the case of no categorical values (only numeric/ordinal), we have only one tree $T$ and the algorithm can at this line after constraining $t = T(x'_{\text{noncat}})$*

**12** Index $t$ as $t = [c_1, c_2, \ldots, c_D, q]$

**13** Let $c = [c_1, c_2, \ldots, c_D]$

**14** Constrain $d$ to distance of $x^*$ to centre: $d = \|x^* - c\|_\infty$

**15** Constrain $d \leq h/2$, since otherwise the point would be rejected and the quantile would be $\infty$.

**16** Obtain $q$ as the local quantile.

---

**MILP encoding of singleton conformal prediction set** The local quantile $q$ is constrained using Algorithm 2 or Algorithm 3. This is used to constrain the conformal prediction set $C_{1-\alpha}(x')$ to a singleton set.

Add $|\mathcal{Y}|$ real variables $s_1, s_2, \ldots, s_{|\mathcal{Y}|}$ to represent the score $s(x, y)$ for each $y \in \mathcal{Y}$.

For random forest classifiers, we use the score function $s(x, y) = 1 - f(x)_y$.

$$s_i = 1 - y_i' \text{ for } i = 1, \ldots, |\mathcal{Y}| \tag{C7}$$

For MLP classifiers that output unnormalised logits, we use the score function Eq. (7), which is $s(x, y) = -l(x)_y + \max_{y' \neq y} l(x)_{y'}$, where $l(x)$ is the predicted vector of logits.

$$\text{Add } m \text{ real variables } m_1, \ldots, m_{|\mathcal{Y}|} \tag{C8}$$

$$m_i = \max_{j \neq i} y_j, \qquad i = 1, \ldots, |\mathcal{Y}| \tag{C9}$$

$$s_i = -y_i' + m_i, \qquad i = 1, \ldots, |\mathcal{Y}| \tag{C10}$$

In the case of binary classification, which any multiclass CFX problem can be reduced to, our score function is the difference between the two logits and we can remove the maximum constraint as follows.

$$s_1 = y_2' - y_1' \text{ for } i = 1 \tag{C11}$$

$$s_2 = -s_1 \text{ for } i = 2 \tag{C12}$$

Finally, we constrain $C_{1-\alpha}(x')$ to a singleton set containing the target class, $\{y^+\}$.

$$s_i \leq q \text{ for all } i \neq y^+ \tag{C13}$$

$$s_i > q \text{ for } i = y^+ \tag{C14}$$

**Further constraints** As mentioned in Section 2, further properties such as causality and actionability can be incorporated into the model by introducing further constraints on $x'$. $\qquad$ (C15)

**Notes on MILP** Strict inequalities such as those present in Eq. (C6b), Eq. (C14) and some model encodings Eq. (C6) can not directly be modelled in MILP, this is resolved by adding a small epsilon to one side of the equation.

In Eq. ((C4)), Algorithm 2, Algorithm 3, we observe products of two variables which would usually indicate a quadratic constraint. However, in all of these cases, at least one variable is a binary variable. This allows the solver to linearise it, see Klotz (2021) for further details.

## C   SCALABILITY ANALYSIS

In this section, we empirically analyse the scalabiluty of the CONFEX-LCP and CONFEX-Tree methods. All experiments were conducted on a MacBook Pro, M3 Pro chipset, 18 GB memory.

### C.1   DATASET DIMENSION

The dimensionality of the dataset affects the number of variables involved in the distances and weights constraints for CONFEX-LCP and the complexity of the tree for CONFEX-Tree. For this experiment, we analyse the effect of changing the dimensionality of the dataset.

We create synthetic datasets of varying dimensionalities by using sklearn's `make_classification` method with all features being informative, and we plot the number of counterfactuals generated per second (observed over a 5-minute period) in Fig. 3. Tabular results are available in Table 4. Note that we fix the kernel bandwidth and alpha value, use a MLP model with 50 hidden units, and use a calibration set size of 150.

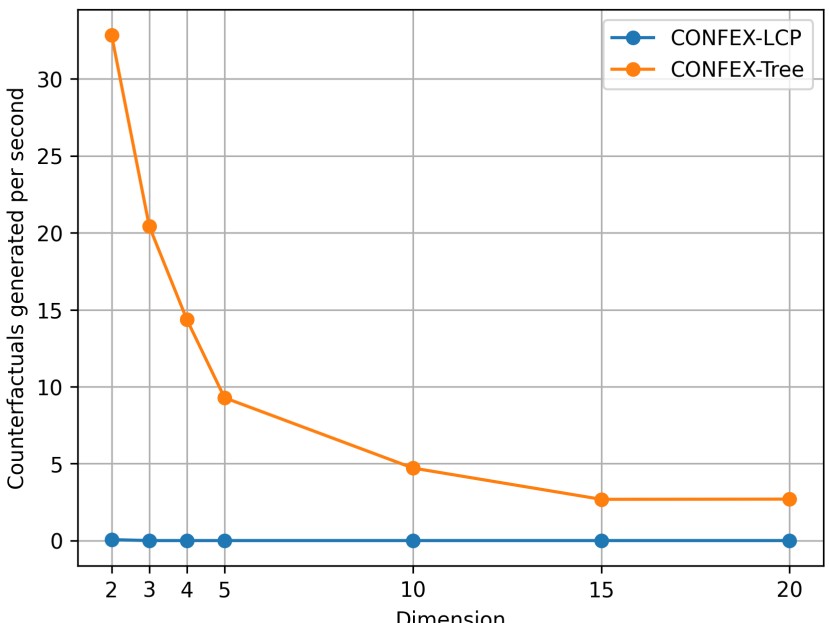

Figure 3: Counterfactuals generated per second for CONFEX-LCP and CONFEX-Tree, against dimensionality of the dataset.

We find that the beyond a dimensionality of 2, CONFEX-LCP is infeasible for use, whilst CONFEX-Tree's generation rate eventually flattens.

| Dataset dimension | CONFEX-LCP | CONFEX-Tree |
|---|---|---|
| 2 | 0.060 | 32.853 |
| 3 | 0.000 | 20.427 |
| 4 | 0.000 | 14.353 |
| 5 | 0.000 | 9.280 |
| 10 | 0.000 | 4.713 |
| 15 | 0.000 | 2.680 |
| 20 | 0.000 | 2.693 |

Table 4: Counterfactuals generated per second for CONFEX-LCP and CONFEX-Tree, against dimensionality of the dataset.

## C.2 SIZE OF CALIBRATION SET

Varying the size of the calibration set also affects the number of variables involved in constraining distances and weights in CONFEX-LCP as well as affecting the complexity of the tree in CONFEX-Tree procedure.

For this experiment, we use the CaliforniaHousing dataset (which has 8 dimensions) with an MLP model containing 50 hidden units, and fix the kernel bandwidth and alpha value. We vary the size of the calibration set from between 10 and 2000 points. The effect on the rate of counterfactual generation is found in Fig. 4.

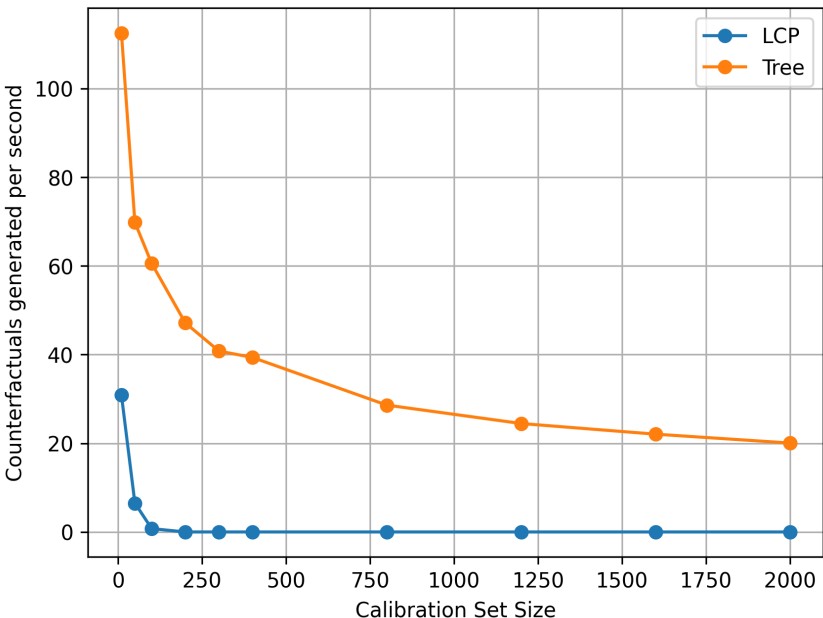

Figure 4: Number of counterfactuals generated per second for CONFEX-LCP and CONFEX-Tree on the CaliforniaHousing dataset, varying the calibration set size.

Fig. 8 shows the effect that an increased calibration set size has on distance and plausibility: we get reduced distances with improved plausibility.

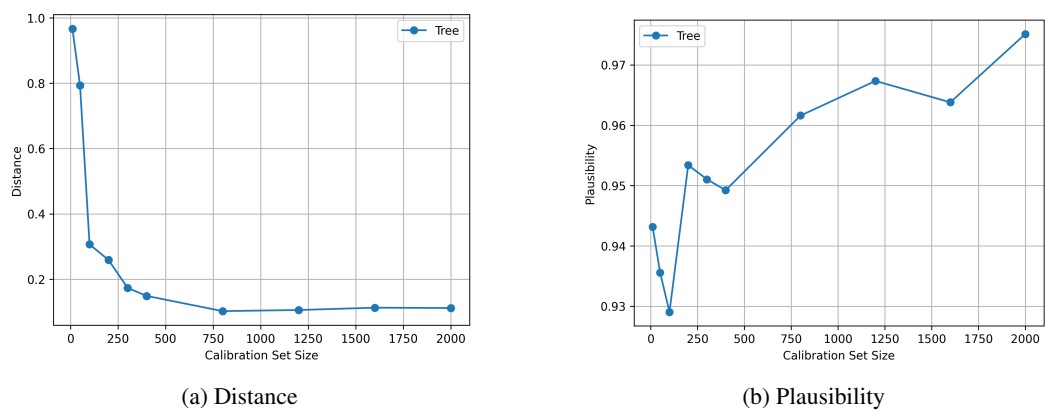

(a) Distance                                              (b) Plausibility

Figure 5: Effect of varying the calibration set size on distance and plausibility on the California-Housing dataset, CONFEX-Tree.

## C.3 MODEL COMPLEXITY

Model outputs are obtained within the MILP formulation through a series of constraints involving the input variables and the details of the trained model. As explained in Section B, (C6), we modify implementations from the `gurobi-machinelearning` library.

In this section, we see how CONFEX-Tree performs as the model complexity changes. We consider three classes of models: multi-layer perceptron, random forests and gradient-boosted trees, varying their hyperparameters. Although we use CONFEX-Tree to generate counterfactuals, this analysis focusses on the MILP encoding of the classifiers and would apply to other MILP methods as well, e.g. MinDist. We use the CaliforniaHousing dataset and fix alpha to 0.1, and bandwidth scale to 1. The following figures show how the number of CFXs generated per second varies as we change hyperparmeters. The accuracy of the models over a test set is also shown.

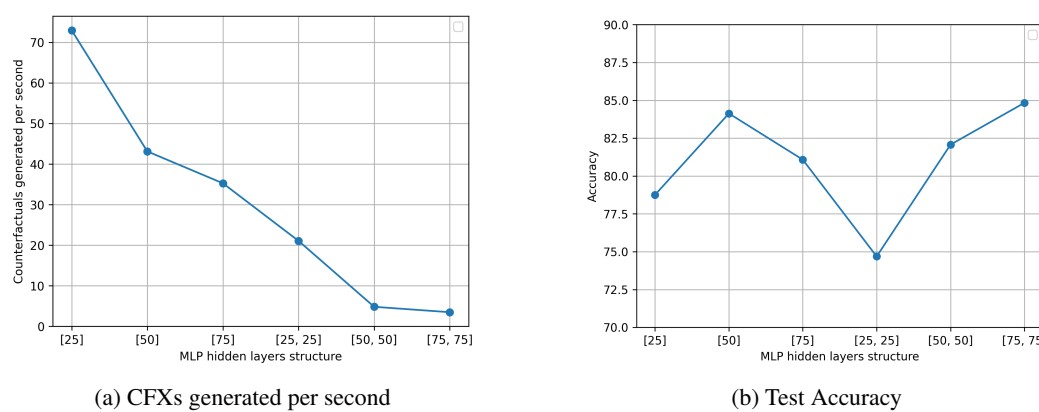

(a) CFXs generated per second                    (b) Test Accuracy

Figure 6: Effect of varying MLP model hyperparameters on number of CFXs generated per second, and accuracy for CONFEX-Tree.

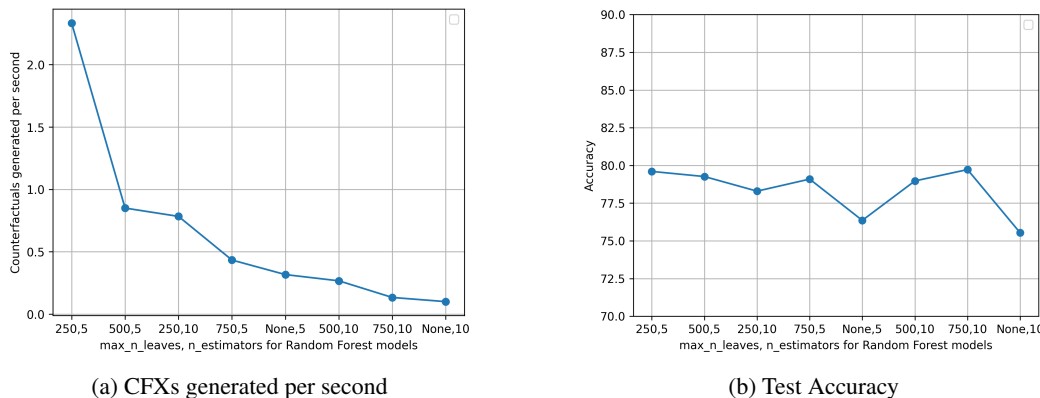

(a) CFXs generated per second                    (b) Test Accuracy

Figure 7: Effect of varying Random Forest model hyperparameters on number of CFXs generated per second, and accuracy for CONFEX-Tree.

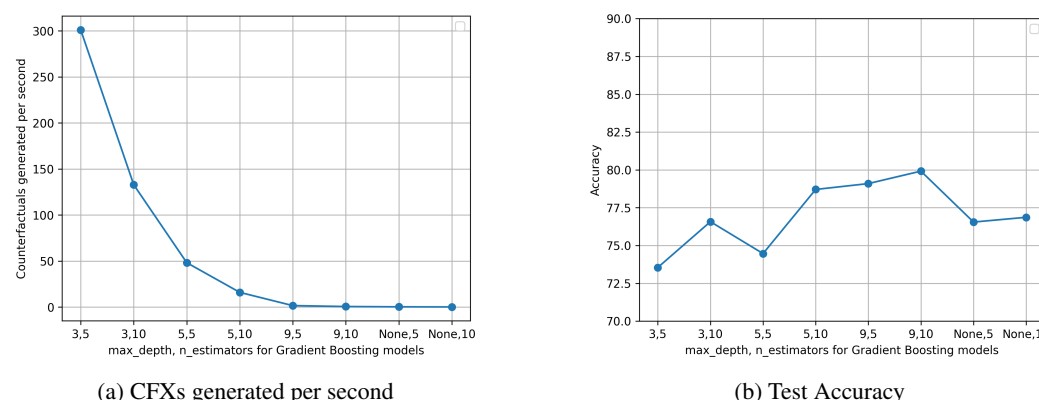

(a) CFXs generated per second · (b) Test Accuracy

Figure 8: Effect of varying Gradient Boosted Trees hyperparameters on number of CFXs generated per second, and accuracy for CONFEX-Tree.

We find that as the complexity of the model increases, the number of CFXs generated per second decreases. In the worst case tested, Gradient Boosted Trees with no depth limit and 10 estimators, we obtain a reasonable 4 CFXs per second. From the accuracy plots, we can see that in many cases, a less complex model can provide similar accuracy and more complex models exhibit overfitting.

# D  FURTHER EVALUATION

## D.1  EXPERIMENTAL SETUP

**Generators.**  For solving MILP instances, we utilise the Gurobi solver, and utilise the Gurobi Machine Learning Gurobi (2022) library to formulate the trained classifiers as constraints. All generators, except FOCUS (using the CFXplorer package Morita (2023)) and FeatureTweak/FACE/C-CHVAE (implementations ported from CARLA Pawelczyk et al. (2021), and FeatureTweakPy[4]), were implemented as part of a Python library to generate CFXs. The details of this library are removed for anonymous submission.

**Model Configuration.**  For all datasets, we used a multilayer perceptron (MLP) with 50 hidden units. The batch size was set to 64 for California Housing and German Credit, trained for 100 epochs, and 256 for GiveMeSomeCredit and Adult Income, trained for 50 epochs. For the random forest model, we also evaluated a Random Forest classifier with 5 estimators and number of leaves limited to 500 for the GiveMeSomeCredit and AdultIncome models.

## D.2  METRICS

In order to evaluate the quality of the generated counterfactual explanations, we adopt a set of quantitative metrics that measure different aspects of their usefulness and reliability. Specifically, we focus on three core dimensions: *plausibility*, *sensitivity*, and *stability*. In addition, we report auxiliary metrics such as the distance of counterfactuals to the original instance, the proportion of failures, and the validity rate of generated explanations. Together, these metrics provide a comprehensive view of both the fidelity and robustness of counterfactual explanations.

**Plausibility.**  A counterfactual explanation should lie close to the underlying data distribution so that it represents a realistic and interpretable alternative. To assess this, we measure plausibility using the Local Outlier Factor (LOF) (Breunig et al., 2000), which quantifies how isolated a sample is with respect to its nearest neighbours. A LOF score of $+1$ indicates that the counterfactual is consistent with observed data, whereas $-1$ suggest that the counterfactual is implausible. We use the `scikit-learn` implementation of LOF with `novelty=True` and $n\_\text{neighbors} = 20$, stratified by the target class. In practice, we average over 100 test points.

**Sensitivity.**  Beyond plausibility, we also want to assess whether counterfactuals are *robust* to small changes in the input instance. Sensitivity measures how much a counterfactual explanation changes when the original instance $x$ is perturbed within a small neighbourhood. Formally, given an input $x$ and its counterfactual $x_c$, we uniformly sample a perturbed instance $x' \sim U_b(x)$ from the $\ell_2$ ball centred around the factual, compute a new counterfactual $x'_c$. Sensitivity is then defined as the relative deviation between the two counterfactuals, normalised by the cost of the initial counterfactual:

$$\text{CFX Sensitivity} = \mathbb{E}_{x' \sim U_b(x)} \left[ \frac{\|x'_c - x_c\|_2}{\|x_c - x\|_2} \right].$$

In practice, we sample 4 neighbours from 25 test points to inform our sensitivity metric. Intuitively, low sensitivity indicates that the explanation remains stable when the factual input undergoes small variations, thereby suggesting robustness and consistency.

In our experiments, we choose the budget $b$ of the uniform sampling to correspond to a ball with 0.1% of the volume of the feature space.

$$V_{\text{ball}} = \frac{\pi^{d/2}}{\Gamma\left(\frac{d}{2} + 1\right)} r^d = b V_{\text{total}}$$

where $d$ is the number of non-categorical features in the space. Solving for $r$,

$$r = \left( \frac{b V_{\text{total}}}{\pi^{d/2} / \Gamma\left(\frac{d}{2} + 1\right)} \right)^{1/d}$$

---

[4]https://github.com/upura/featureTweakPy/blob/master/featureTweakPy.py

This allows the same budget to be used across datasets with differing numbers of features. When sampling neighbours, we do not change categorical values and we fix ordinal values to their closest valid value.

**Stability.** Complementary to sensitivity, stability measures how consistent the counterfactual is under perturbations applied directly to the counterfactual itself. That is, we perturb $x_c$ within a budgeted neighbourhood and evaluate the variance in the model predictions across these perturbed samples. Following an adaptation of (Dutta et al., 2022), stability is computed as:

$$\text{CFX Stability} = \frac{1}{K} \sum_{x' \in N_x} \hat{f}(x')_{y^+} - \sqrt{\frac{1}{K} \sum_{x' \in N_x} \left( \hat{f}(x')_{y^+} - \frac{1}{K} \sum_{x' \in N_x} \hat{f}(x')_{y^+} \right)^2},$$

where $N_x$ is a set of $K$ points sampled as $x' \sim U_b(x_c)$.

where $\hat{f}(x')_{y^+}$ refers to the predicted probability of the target class. The metric neighbours a large mean value for the predicted probability of sampled neighbours, whilst penalising variations in these values by subtracting the standard deviation to ensure that that mean is not a combination of very high and very low values. Similarly to the Sensivity metric, $U_b(x_c)$ denotes sampling from the $\ell_2$ ball centred around the counterfactual, computing the radius in the same way, taking the budget to represent 0.1% of the total feature volume.

Stability is high when the predictions across perturbed counterfactuals remain close to each other, which indicates that the explanation is not overly sensitive to minor fluctuations in its actualisation.

**Certainty.** CONFEX minimises the uncertainty of the counterfactual by constraining the conformal prediction set to be a singleton containing the target class only. Certainty in the counterfactual relates to the property $\mathbb{P}(y = y^+ | x = x')$.

To quantify the certainty of the counterfactuals in a principled way, we use local conformal $p$-values. In a conformal prediction procedure, the conformal $p$-value of a point (x,y) is the proportion of the calibration points with score above s(x,y). It is used to determine which labels are included in the prediction set: labels with a $p$-value over $\alpha$ are included and the rest excluded. This is equivalent to checking if s(x,y) is above the $1 - \alpha$ quantile of the calibration score (as explained in Section 2).

A high $p$-value for (x,y) provides strong evidence that y is is the true label for x. Hence, for our certainty metric, we compute the average difference between the conformal $p$-value for the target class and the max of conformal $p$-values for all other classes. If the $p$-value of the target class is high and the max $p$-value of the other classes are low - indicating a certain prediction with strong evidence in favour of the target class and against others - then our metric will be high. If the prediction is uncertain then the $p$-values of all classes will be similar, leading to a lower value of our metric.

Note that we use the LCP procedure to compute $p$-values for this metric because of its local guarantees. We don't use vanilla CP, because its resulting $p$-values would be affected by calibrations points well-away (not local) to the counterfactual point of interest. We do not use the CONFEX-Tree procedure to compute local $p$-values since the metric may be seen as tailored to our generator.

$$\text{cert}(x') = p_{y^+}(x') - \max_{y \neq y^+} p_y(x') \tag{18}$$

Certainty results are reported in tables found in Section A.2.

**Auxiliary metrics.** In addition to the three core dimensions, we report the following supplementary measures:

- *Distance:* the average L1 distance between the original instance and the counterfactual,

$$\text{Distance} = \mathbb{E}\big(\|x' - x_0\|_1\big),$$

which quantifies the minimality of the intervention required.

- *Validity:* the proportion of counterfactuals that successfully change the prediction to the desired class,

$$\text{Validity} = \mathbb{E}(1\{\hat{f}(x') = y^+\}).$$

For example, invalidity could be due to numerical artefacts in encoding the models in MILP, or failure for SGD procedures to converge to a flipped class. We report whenever a method a method produces less than 90% validity, and exclude invalid CFXs from the computation of other metrics.

- *Failure rate:* the proportion of runs where the generator fails to produce a counterfactual, for example due to infeasible constraints in optimisation-based methods such as MILP.

- *Implausibility:* The average distance from the counterfactual to the closest 10% of points of the target class, similar to Altmeyer et al. (2024).

### D.2.1 CONDITIONAL COVERAGE RESULTS

In the additional results, we furthermore evaluate the performance of different conformal CFX generators under four evaluation settings: marginal coverage, class-conditional coverage, random binning, and counterfactual similarity. In the paper we discussed the counterfactual simulation, however we also evaluate the marginal coverage over a test set, average class-conditional coverage, average coverage over a random paritioning of the test set into 3 bins. We report the coverage gap (Barber et al. (2023)): the difference between the empirical coverage and target coverage, in percentage points.

$$\text{CovGap} = 100 \times (\mathbb{P}\{y \in C(x)\} - (1 - \alpha)) \tag{19}$$

## D.3 CALIFORNIA HOUSING

We use the California Housing dataset Pace & Barry (1997) from the StatLib repository through scikit-learn's `sklearn.datasets.fetch_california_housing` function[5]. The original regression problem was changed into a binary classification task by categorizing houses based on whether the median income exceeds $20,000 (42% above, 58% below). The dataset contains 8 numeric features, which we scaled to the range $(0, 1)$ using MinMax scaling.

Our results demonstrate a nice pattern showing that distance decreases and plausibility decreases as the kernel bandwidth increases. CONFEX methods outperform all other methods (except FACE, where it comes second) on plausibility and sensitivity.

### D.3.1 PLOTS

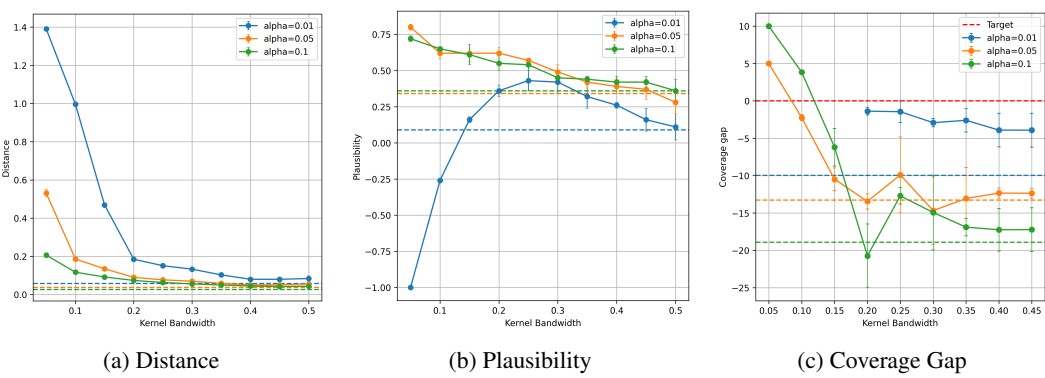

| (a) Distance | (b) Plausibility | (c) Coverage Gap |

Figure 9: Effect of coverage rate and kernel bandwidth on metrics for CONFEX-Tree on the CaliforniaHousing dataset, MLP. CONFEX-Naive is represented by dashed horizontal lines.

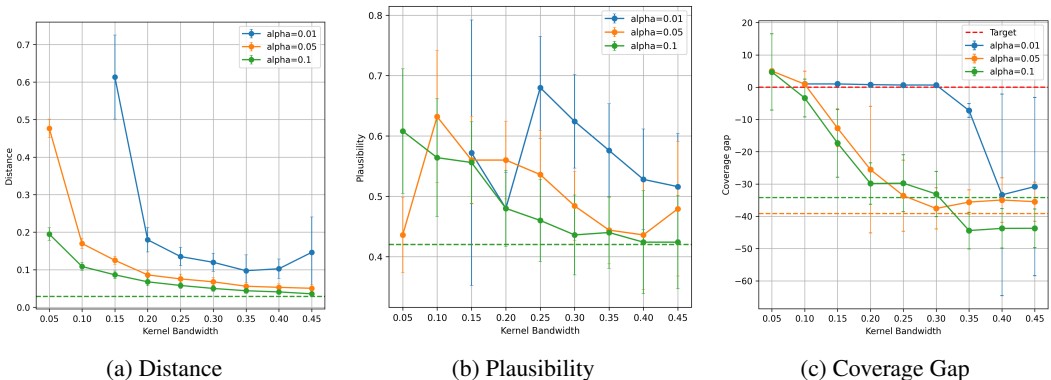

| (a) Distance | (b) Plausibility | (c) Coverage Gap |

Figure 10: Effect of coverage rate and kernel bandwidth on metrics for CONFEX-Tree on the CaliforniaHousing dataset, RandomForest. CONFEX-Naive is represented by dashed horizontal lines.

---

[5] https://www.dcc.fc.up.pt/~ltorgo/Regression/cal_housing.html

### D.3.2   MODEL EVALUATION RESULTS

| Repeat | Accuracy (%) | Precision (%) | F1 Score (%) |
|---|---|---|---|
| repeat0,MLP | 83.58 | 83.61 | 83.59 |
| repeat1,MLP | 82.95 | 83.59 | 82.95 |
| repeat2,MLP | 78.20 | 80.17 | 78.06 |
| repeat3,MLP | 81.59 | 82.37 | 81.59 |
| repeat4,MLP | 79.31 | 80.71 | 79.25 |
| repeat0,RF | 78.05 | 80.60 | 77.85 |
| repeat1,RF | 78.10 | 80.60 | 77.90 |
| repeat2,RF | 77.59 | 81.06 | 77.26 |
| repeat3,RF | 76.02 | 79.26 | 75.68 |
| repeat4,RF | 76.36 | 79.09 | 76.10 |

Table 5: Model evaluation results, CaliforniaHousing.

### D.3.3 CFX GENERATION RESULTS

| Generator | Distance | Plausibility | Implausibility | Sensitivity ($10^{-1}$) | Stability |
|---|---|---|---|---|---|
| **MLP** | | | | | |
| MinDist | 0.03 ± 0.00 | 0.30 ± 0.07 | 0.21 ± 0.01 | 42.75 ± 8.45 | 0.02 ± 0.04 |
| Wachter | 0.09 ± 0.01 | 0.42 ± 0.08 | 0.20 ± 0.00 | 1.66 ± 0.37 | 0.02 ± 0.05 |
| Greedy | 1.88 ± 0.27 | -0.99 ± 0.02 | 0.89 ± 0.10 | 0.14 ± 0.02 | 0.42 ± 0.10 |
| ConfexNaive | | | | | |
| $\alpha = 0.01$ | 0.07 ± 0.01 | 0.04 ± 0.06 | 0.22 ± 0.01 | 3.45 ± 1.25 | 0.02 ± 0.05 |
| $\alpha = 0.05$ | 0.04 ± 0.01 | 0.24 ± 0.08 | 0.22 ± 0.01 | 8.41 ± 2.26 | 0.02 ± 0.05 |
| $\alpha = 0.1$ | 0.03 ± 0.01 | 0.27 ± 0.08 | 0.21 ± 0.01 | 14.51 ± 3.09 | 0.02 ± 0.04 |
| ECCCo | | | | | |
| $\alpha = 0.01$ | 0.39 ± 0.02 | -0.69 ± 0.04 | 0.21 ± 0.01 | 0.24 ± 0.05 | 0.26 ± 0.08 |
| $\alpha = 0.05$ | 0.37 ± 0.02 | -0.65 ± 0.05 | 0.21 ± 0.01 | 0.26 ± 0.05 | 0.25 ± 0.08 |
| $\alpha = 0.1$ | 0.37 ± 0.02 | -0.63 ± 0.04 | 0.21 ± 0.01 | 0.26 ± 0.05 | 0.24 ± 0.08 |
| ConfexTree, $\alpha = 0.01$ | | | | | |
| bw = 0.05 | 1.40 ± 0.04 | -1.00 ± 0.00 | 0.39 ± 0.00 | 0.00 ± 0.00 | 1.00 ± 0.00 |
| bw = 0.1 | 1.03 ± 0.04 | -0.48 ± 0.19 | 0.15 ± 0.01 | 0.02 ± 0.01 | 0.03 ± 0.04 |
| bw = 0.15 | 0.52 ± 0.04 | 0.08 ± 0.19 | 0.16 ± 0.00 | 0.07 ± 0.01 | 0.05 ± 0.04 |
| bw = 0.2 | 0.21 ± 0.03 | 0.30 ± 0.12 | 0.16 ± 0.00 | 0.40 ± 0.27 | 0.05 ± 0.04 |
| bw = 0.25 | 0.17 ± 0.02 | 0.35 ± 0.17 | 0.16 ± 0.00 | 0.60 ± 0.32 | 0.05 ± 0.05 |
| bw = 0.3 | 0.16 ± 0.02 | 0.33 ± 0.13 | 0.17 ± 0.00 | 1.02 ± 0.37 | 0.05 ± 0.05 |
| bw = 0.35 | 0.12 ± 0.01 | 0.28 ± 0.09 | 0.18 ± 0.00 | 2.07 ± 0.62 | 0.04 ± 0.05 |
| bw = 0.4 | 0.09 ± 0.01 | 0.17 ± 0.08 | 0.20 ± 0.00 | 2.91 ± 0.99 | 0.03 ± 0.05 |
| bw = 0.45 | 0.09 ± 0.01 | 0.11 ± 0.08 | 0.20 ± 0.00 | 2.88 ± 0.76 | 0.03 ± 0.05 |
| bw = 0.5 | 0.08 ± 0.01 | 0.11 ± 0.09 | 0.20 ± 0.00 | 2.14 ± 0.31 | 0.08 ± 0.03 |
| ConfexTree, $\alpha = 0.05$ | | | | | |
| bw = 0.05 | 0.55 ± 0.04 | 0.75 ± 0.05 | 0.15 ± 0.00 | 0.05 ± 0.05 | 0.07 ± 0.03 |
| bw = 0.1 | 0.19 ± 0.02 | 0.60 ± 0.07 | 0.17 ± 0.01 | 0.43 ± 0.12 | 0.05 ± 0.05 |
| bw = 0.15 | 0.15 ± 0.01 | 0.53 ± 0.09 | 0.17 ± 0.01 | 0.83 ± 0.48 | 0.05 ± 0.05 |
| bw = 0.2 | 0.10 ± 0.02 | 0.55 ± 0.09 | 0.18 ± 0.01 | 2.15 ± 0.73 | 0.03 ± 0.05 |
| bw = 0.25 | 0.09 ± 0.02 | 0.50 ± 0.09 | 0.18 ± 0.01 | 3.05 ± 0.79 | 0.03 ± 0.04 |
| bw = 0.3 | 0.08 ± 0.01 | 0.43 ± 0.07 | 0.19 ± 0.01 | 4.73 ± 2.37 | 0.03 ± 0.04 |
| bw = 0.35 | 0.07 ± 0.01 | 0.36 ± 0.07 | 0.19 ± 0.00 | 6.03 ± 0.74 | 0.03 ± 0.04 |
| bw = 0.4 | 0.06 ± 0.01 | 0.30 ± 0.08 | 0.20 ± 0.00 | 6.69 ± 1.37 | 0.03 ± 0.04 |
| bw = 0.45 | 0.06 ± 0.01 | 0.26 ± 0.10 | 0.20 ± 0.01 | 6.93 ± 1.44 | 0.03 ± 0.04 |
| bw = 0.5 | 0.05 ± 0.01 | 0.28 ± 0.08 | 0.20 ± 0.00 | 7.16 ± 0.70 | 0.07 ± 0.02 |
| ConfexTree, $\alpha = 0.1$ | | | | | |
| bw = 0.05 | 0.25 ± 0.04 | 0.58 ± 0.12 | 0.17 ± 0.01 | 0.20 ± 0.09 | 0.06 ± 0.04 |
| bw = 0.1 | 0.14 ± 0.02 | 0.62 ± 0.05 | 0.17 ± 0.01 | 0.96 ± 0.75 | 0.04 ± 0.04 |
| bw = 0.15 | 0.11 ± 0.02 | 0.58 ± 0.08 | 0.18 ± 0.00 | 2.76 ± 2.77 | 0.03 ± 0.05 |
| bw = 0.2 | 0.09 ± 0.01 | 0.48 ± 0.07 | 0.19 ± 0.00 | 4.88 ± 2.61 | 0.03 ± 0.04 |
| bw = 0.25 | 0.07 ± 0.01 | 0.44 ± 0.09 | 0.19 ± 0.00 | 4.34 ± 1.72 | 0.03 ± 0.04 |
| bw = 0.3 | 0.07 ± 0.01 | 0.39 ± 0.05 | 0.19 ± 0.00 | 6.42 ± 2.59 | 0.02 ± 0.04 |
| bw = 0.35 | 0.06 ± 0.01 | 0.36 ± 0.07 | 0.19 ± 0.00 | 9.80 ± 0.97 | 0.02 ± 0.04 |
| bw = 0.4 | 0.05 ± 0.01 | 0.34 ± 0.07 | 0.20 ± 0.00 | 12.93 ± 3.75 | 0.02 ± 0.04 |
| bw = 0.45 | 0.05 ± 0.01 | 0.32 ± 0.09 | 0.20 ± 0.01 | 14.53 ± 3.82 | 0.02 ± 0.04 |
| bw = 0.5 | 0.04 ± 0.01 | 0.36 ± 0.08 | 0.20 ± 0.00 | 13.87 ± 0.66 | 0.07 ± 0.02 |
| FACE | 0.21 ± 0.02 | 0.85 ± 0.04 | 0.16 ± 0.00 | 0.34 ± 0.03 | 0.06 ± 0.05 |
| C-CHVAE | 1.27 ± 0.22 | -0.35 ± 0.12 | 0.46 ± 0.15 | 0.06 ± 0.03 | 0.16 ± 0.03 |

Table 6: CFX generation results, CaliforniaHousing, MLP. All methods attained full validity.

| Generator | Distance | Plausibility | Implausibility | Sensitivity ($10^{-1}$) | Stability |
|---|---|---|---|---|---|
| MinDist | 0.01 ± 0.00 | 0.37 ± 0.07 | 0.21 ± 0.00 | 89.15 ± 90.88 | 0.22 ± 0.02 |
| ConfexNaive | | | | | |
| $\alpha = 0.01$ | 0.03 ± 0.01 | 0.42 ± 0.07 | 0.20 ± 0.00 | 12.34 ± 2.94 | 0.23 ± 0.02 |
| $\alpha = 0.05$ | 0.03 ± 0.01 | 0.42 ± 0.07 | 0.20 ± 0.00 | 12.34 ± 2.94 | 0.23 ± 0.02 |
| $\alpha = 0.1$ | 0.03 ± 0.01 | 0.42 ± 0.07 | 0.20 ± 0.00 | 12.34 ± 2.94 | 0.23 ± 0.02 |
| ConfexTree, $\alpha = 0.01$ | | | | | |
| bw = 0.05 | nan ± nan | nan ± nan | nan ± nan | nan ± nan | nan ± nan |
| bw = 0.1 | nan ± nan | nan ± nan | nan ± nan | nan ± nan | nan ± nan |
| bw = 0.15 | 0.61 ± 0.11 | 0.57 ± 0.22 | 0.16 ± 0.01 | 0.08 ± 0.03 | 0.27 ± 0.03 |
| bw = 0.2 | 0.18 ± 0.03 | 0.48 ± 0.06 | 0.16 ± 0.00 | 0.59 ± 0.23 | 0.23 ± 0.02 |
| bw = 0.25 | 0.14 ± 0.02 | 0.68 ± 0.08 | 0.16 ± 0.00 | 1.55 ± 0.76 | 0.23 ± 0.02 |
| bw = 0.3 | 0.12 ± 0.02 | 0.62 ± 0.08 | 0.17 ± 0.00 | 2.13 ± 0.77 | 0.23 ± 0.02 |
| bw = 0.35 | 0.10 ± 0.04 | 0.58 ± 0.08 | 0.17 ± 0.01 | 3.77 ± 1.59 | 0.23 ± 0.02 |
| bw = 0.4 | 0.10 ± 0.03 | 0.53 ± 0.08 | 0.18 ± 0.00 | 5.53 ± 1.39 | 0.22 ± 0.02 |
| bw = 0.45 | 0.15 ± 0.09 | 0.52 ± 0.09 | 0.18 ± 0.01 | 4.78 ± 1.98 | 0.23 ± 0.02 |
| ConfexTree, $\alpha = 0.05$ | | | | | |
| bw = 0.05 | 0.48 ± 0.02 | 0.44 ± 0.06 | 0.16 ± 0.00 | 0.06 ± 0.05 | 0.26 ± 0.03 |
| bw = 0.1 | 0.17 ± 0.01 | 0.63 ± 0.11 | 0.17 ± 0.00 | 0.47 ± 0.10 | 0.23 ± 0.01 |
| bw = 0.15 | 0.12 ± 0.01 | 0.56 ± 0.07 | 0.17 ± 0.00 | 0.91 ± 0.41 | 0.22 ± 0.01 |
| bw = 0.2 | 0.09 ± 0.01 | 0.56 ± 0.06 | 0.18 ± 0.00 | 2.58 ± 1.75 | 0.22 ± 0.02 |
| bw = 0.25 | 0.08 ± 0.01 | 0.54 ± 0.07 | 0.18 ± 0.00 | 3.57 ± 1.14 | 0.23 ± 0.02 |
| bw = 0.3 | 0.07 ± 0.01 | 0.48 ± 0.06 | 0.19 ± 0.00 | 4.52 ± 1.61 | 0.22 ± 0.02 |
| bw = 0.35 | 0.06 ± 0.01 | 0.44 ± 0.06 | 0.19 ± 0.00 | 8.32 ± 2.21 | 0.22 ± 0.02 |
| bw = 0.4 | 0.05 ± 0.01 | 0.44 ± 0.09 | 0.19 ± 0.00 | 7.12 ± 2.07 | 0.22 ± 0.02 |
| bw = 0.45 | 0.05 ± 0.01 | 0.48 ± 0.11 | 0.19 ± 0.01 | 8.99 ± 3.49 | 0.22 ± 0.02 |
| ConfexTree, $\alpha = 0.1$ | | | | | |
| bw = 0.05 | 0.19 ± 0.02 | 0.61 ± 0.10 | 0.17 ± 0.00 | 0.40 ± 0.18 | 0.23 ± 0.02 |
| bw = 0.1 | 0.11 ± 0.01 | 0.56 ± 0.10 | 0.18 ± 0.00 | 1.64 ± 1.18 | 0.22 ± 0.02 |
| bw = 0.15 | 0.09 ± 0.01 | 0.56 ± 0.07 | 0.18 ± 0.00 | 1.56 ± 0.89 | 0.22 ± 0.02 |
| bw = 0.2 | 0.07 ± 0.01 | 0.48 ± 0.06 | 0.19 ± 0.00 | 4.61 ± 2.05 | 0.22 ± 0.02 |
| bw = 0.25 | 0.06 ± 0.01 | 0.46 ± 0.07 | 0.19 ± 0.00 | 5.80 ± 1.61 | 0.22 ± 0.02 |
| bw = 0.3 | 0.05 ± 0.01 | 0.44 ± 0.07 | 0.19 ± 0.00 | 7.30 ± 1.29 | 0.22 ± 0.02 |
| bw = 0.35 | 0.04 ± 0.01 | 0.44 ± 0.06 | 0.19 ± 0.00 | 11.46 ± 1.56 | 0.22 ± 0.02 |
| bw = 0.4 | 0.04 ± 0.01 | 0.42 ± 0.09 | 0.19 ± 0.00 | 13.16 ± 1.83 | 0.22 ± 0.02 |
| bw = 0.45 | 0.04 ± 0.01 | 0.42 ± 0.08 | 0.20 ± 0.00 | 15.65 ± 2.91 | 0.22 ± 0.02 |
| FeatureTweak | 0.12 ± 0.03 | 0.29 ± 0.25 | 0.21 ± 0.02 | 0.58 ± 0.16 | 0.24 ± 0.03 |
| FOCUS | 0.11 ± 0.01 | 0.34 ± 0.09 | 0.20 ± 0.00 | 5.21 ± 2.31 | 0.24 ± 0.02 |
| FACE | 0.17 ± 0.01 | 0.81 ± 0.02 | 0.17 ± 0.00 | 0.46 ± 0.08 | 0.24 ± 0.02 |

Table 7: CFX generation results, CaliforniaHousing, RandomForest. Methods with nan values had 100% failures. Validity 58% for FeatureTweak.

### D.3.4 CONFORMAL EVALUATION RESULTS

| Generator | Marginal CovGap | Binning CovGap | Class Cond CovGap | Simulated CovGap |
|---|---|---|---|---|
| **MLP** | | | | |
| ConfexNaive | | | | |
| $\alpha = 0.01$ | 0.99 ± 0.00 | -0.35 ± 0.66 | -0.38 ± 0.63 | -8.61 ± 1.51 |
| $\alpha = 0.05$ | 0.96 ± 0.02 | 0.22 ± 1.84 | 0.17 ± 1.78 | -12.83 ± 4.89 |
| $\alpha = 0.1$ | 0.92 ± 0.02 | -0.19 ± 1.76 | -0.34 ± 1.71 | -22.79 ± 7.76 |
| ConfexTree, $\alpha = 0.01$ | | | | |
| bw = 0.05 | 1.00 ± 0.00 | 1.00 ± 0.00 | 1.00 ± 0.00 | nan ± nan |
| bw = 0.1 | 1.00 ± 0.00 | 1.00 ± 0.00 | 1.00 ± 0.00 | nan ± nan |
| bw = 0.15 | 1.00 ± 0.00 | 1.00 ± 0.00 | 1.00 ± 0.00 | nan ± nan |
| bw = 0.2 | 1.00 ± 0.00 | 0.91 ± 0.02 | 0.90 ± 0.02 | -1.04 ± 0.49 |
| bw = 0.25 | 1.00 ± 0.00 | 0.81 ± 0.04 | 0.80 ± 0.04 | -1.87 ± 0.84 |
| bw = 0.3 | 1.00 ± 0.00 | 0.75 ± 0.03 | 0.74 ± 0.04 | -2.60 ± 0.66 |
| bw = 0.35 | 1.00 ± 0.00 | 0.60 ± 0.08 | 0.57 ± 0.08 | -3.98 ± 1.54 |
| bw = 0.4 | 1.00 ± 0.00 | 0.34 ± 0.05 | 0.30 ± 0.06 | -4.76 ± 1.45 |
| bw = 0.45 | 1.00 ± 0.00 | 0.32 ± 0.05 | 0.28 ± 0.06 | -4.77 ± 1.45 |
| ConfexTree, $\alpha = 0.05$ | | | | |
| bw = 0.05 | 1.00 ± 0.00 | 5.00 ± 0.00 | 5.00 ± 0.00 | 5.00 ± 0.00 |
| bw = 0.1 | 1.00 ± 0.00 | 4.91 ± 0.00 | 4.90 ± 0.00 | -2.29 ± 0.31 |
| bw = 0.15 | 1.00 ± 0.00 | 4.82 ± 0.03 | 4.81 ± 0.03 | -10.12 ± 1.42 |
| bw = 0.2 | 1.00 ± 0.00 | 4.67 ± 0.04 | 4.65 ± 0.04 | -10.19 ± 5.66 |
| bw = 0.25 | 0.99 ± 0.00 | 4.35 ± 0.08 | 4.32 ± 0.07 | -7.64 ± 4.82 |
| bw = 0.3 | 0.99 ± 0.00 | 3.83 ± 0.12 | 3.76 ± 0.11 | -11.33 ± 3.95 |
| bw = 0.35 | 0.97 ± 0.01 | 1.55 ± 0.27 | 1.35 ± 0.29 | -13.37 ± 2.89 |
| bw = 0.4 | 0.95 ± 0.00 | 0.09 ± 0.47 | -0.19 ± 0.53 | -14.08 ± 2.06 |
| bw = 0.45 | 0.95 ± 0.00 | 0.07 ± 0.47 | -0.22 ± 0.53 | -14.06 ± 1.97 |
| ConfexTree, $\alpha = 0.1$ | | | | |
| bw = 0.05 | 1.00 ± 0.00 | 9.99 ± 0.01 | 9.99 ± 0.01 | -0.56 ± 14.46 |
| bw = 0.1 | 1.00 ± 0.00 | 9.76 ± 0.08 | 9.74 ± 0.08 | 3.16 ± 0.88 |
| bw = 0.15 | 1.00 ± 0.00 | 9.54 ± 0.21 | 9.52 ± 0.22 | -10.92 ± 5.77 |
| bw = 0.2 | 0.99 ± 0.00 | 8.98 ± 0.14 | 8.93 ± 0.15 | -19.47 ± 4.74 |
| bw = 0.25 | 0.99 ± 0.00 | 8.49 ± 0.25 | 8.42 ± 0.24 | -11.00 ± 2.93 |
| bw = 0.3 | 0.98 ± 0.00 | 7.55 ± 0.38 | 7.44 ± 0.35 | -14.15 ± 3.08 |
| bw = 0.35 | 0.91 ± 0.01 | 1.87 ± 0.80 | 1.41 ± 0.90 | -22.28 ± 6.52 |
| bw = 0.4 | 0.91 ± 0.01 | 1.01 ± 0.82 | 0.51 ± 0.92 | -22.27 ± 5.75 |
| bw = 0.45 | 0.91 ± 0.01 | 0.97 ± 0.83 | 0.47 ± 0.93 | -22.19 ± 5.64 |

Table 8: Conformal evaluation results, CaliforniaHousing, MLP

| Generator | Marginal CovGap | Binning CovGap | Class Cond CovGap | Simulated CovGap |
|---|---|---|---|---|
| **RandomForest** | | | | |
| ConfexNaive | | | | |
| $\alpha = 0.01$ | 1.00 ± 0.00 | 1.00 ± 0.00 | 1.00 ± 0.00 | nan ± nan |
| $\alpha = 0.05$ | 0.95 ± 0.01 | 0.16 ± 0.53 | -0.10 ± 0.56 | -39.17 ± 8.15 |
| $\alpha = 0.1$ | 0.95 ± 0.01 | 5.16 ± 0.53 | 4.90 ± 0.56 | -34.17 ± 8.15 |
| ConfexTree, $\alpha = 0.01$ | | | | |
| bw = 0.05 | 1.00 ± 0.00 | 1.00 ± 0.00 | 1.00 ± 0.00 | nan ± nan |
| bw = 0.1 | 1.00 ± 0.00 | 1.00 ± 0.00 | 1.00 ± 0.00 | 1.00 ± nan |
| bw = 0.15 | 1.00 ± 0.00 | 1.00 ± 0.00 | 1.00 ± 0.00 | 1.00 ± nan |
| bw = 0.2 | 1.00 ± 0.00 | 0.97 ± 0.02 | 0.97 ± 0.03 | 0.79 ± 0.41 |
| bw = 0.25 | 1.00 ± 0.00 | 0.97 ± 0.03 | 0.97 ± 0.03 | 0.66 ± 0.65 |
| bw = 0.3 | 1.00 ± 0.00 | 0.97 ± 0.03 | 0.97 ± 0.03 | 0.66 ± 0.65 |
| bw = 0.35 | 1.00 ± 0.00 | 0.96 ± 0.07 | 0.96 ± 0.07 | -7.24 ± 2.12 |
| bw = 0.4 | 1.00 ± 0.00 | 0.83 ± 0.23 | 0.83 ± 0.23 | -33.35 ± 31.24 |
| bw = 0.45 | 1.00 ± 0.00 | 0.83 ± 0.23 | 0.83 ± 0.23 | -30.80 ± 27.63 |
| ConfexTree, $\alpha = 0.05$ | | | | |
| bw = 0.05 | 1.00 ± 0.00 | 5.00 ± 0.00 | 5.00 ± 0.00 | 5.00 ± 0.00 |
| bw = 0.1 | 1.00 ± 0.00 | 4.93 ± 0.05 | 4.93 ± 0.05 | 0.93 ± 3.99 |
| bw = 0.15 | 1.00 ± 0.00 | 4.88 ± 0.05 | 4.87 ± 0.06 | -12.72 ± 5.73 |
| bw = 0.2 | 1.00 ± 0.00 | 4.76 ± 0.10 | 4.75 ± 0.10 | -25.53 ± 19.62 |
| bw = 0.25 | 1.00 ± 0.00 | 4.48 ± 0.10 | 4.46 ± 0.10 | -33.60 ± 11.02 |
| bw = 0.3 | 0.99 ± 0.00 | 4.00 ± 0.25 | 3.98 ± 0.26 | -37.57 ± 6.37 |
| bw = 0.35 | 0.97 ± 0.01 | 2.67 ± 0.31 | 2.56 ± 0.32 | -35.59 ± 3.75 |
| bw = 0.4 | 0.97 ± 0.01 | 2.54 ± 0.46 | 2.43 ± 0.47 | -34.96 ± 6.88 |
| bw = 0.45 | 0.97 ± 0.01 | 2.51 ± 0.45 | 2.40 ± 0.46 | -35.47 ± 6.05 |
| ConfexTree, $\alpha = 0.1$ | | | | |
| bw = 0.05 | 1.00 ± 0.00 | 10.00 ± 0.01 | 10.00 ± 0.01 | 4.72 ± 11.81 |
| bw = 0.1 | 1.00 ± 0.00 | 9.81 ± 0.05 | 9.80 ± 0.05 | -3.35 ± 5.87 |
| bw = 0.15 | 1.00 ± 0.00 | 9.60 ± 0.23 | 9.57 ± 0.24 | -17.34 ± 10.55 |
| bw = 0.2 | 1.00 ± 0.00 | 9.33 ± 0.04 | 9.31 ± 0.04 | -29.85 ± 6.39 |
| bw = 0.25 | 0.99 ± 0.01 | 8.51 ± 0.40 | 8.47 ± 0.42 | -29.75 ± 8.83 |
| bw = 0.3 | 0.98 ± 0.01 | 7.57 ± 0.56 | 7.53 ± 0.58 | -33.10 ± 7.00 |
| bw = 0.35 | 0.93 ± 0.01 | 3.39 ± 0.99 | 3.16 ± 1.02 | -44.42 ± 5.68 |
| bw = 0.4 | 0.93 ± 0.01 | 2.95 ± 1.07 | 2.69 ± 1.10 | -43.75 ± 6.11 |
| bw = 0.45 | 0.93 ± 0.01 | 2.84 ± 1.02 | 2.58 ± 1.05 | -43.73 ± 5.96 |

Table 9: Conformal evaluation results, CaliforniaHousing, RandomForest

## D.4 GERMAN CREDIT

We use the German Credit dataset from the UCI Machine Learning Repository Hofmann (1994), with a cleaned version obtained through Kaggle[6]. The preprocessing included: (i) scaling numeric features (Age, Credit amount, Duration) to $(0, 1)$ using MinMax scaling, (ii) ordinal encoding of categorical features (job, savings account, checking account), then normalised. The Purpose feature was dropped.

Our results show that distance decreases and plausibility decreases as the kernel bandwidth increases. When the bandwidth is properly tuned, CONFEX methods outperform all other methods on plausibility and sensitivity.

### D.4.1 PLOTS

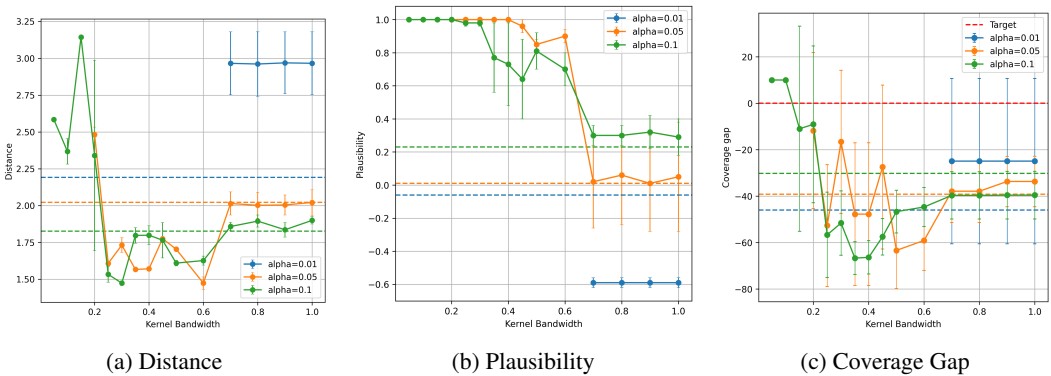

| (a) Distance | (b) Plausibility | (c) Coverage Gap |

Figure 11: Effect of coverage rate and kernel bandwidth on metrics for CONFEX-Tree on the GermanCredit dataset, MLP. CONFEX-Naive is represented by dashed horizontal lines.

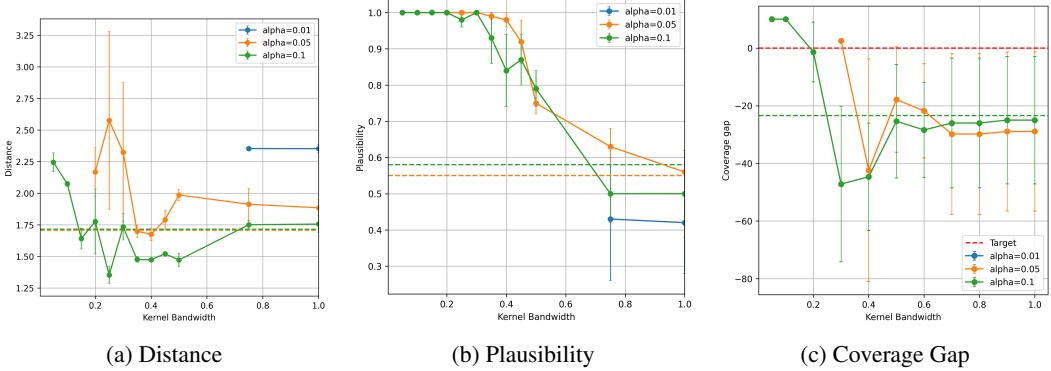

| (a) Distance | (b) Plausibility | (c) Coverage Gap |

Figure 12: Effect of coverage rate and kernel bandwidth on metrics for CONFEX-Tree on the GermanCredit dataset, RandomForest. CONFEX-Naive is represented by dashed horizontal lines.

[6]https://www.kaggle.com/datasets/uciml/german-credit/data

### D.4.2 MODEL EVALUATION RESULTS

| Repeat | Accuracy (%) | Precision (%) | F1 Score (%) |
|---|---|---|---|
| repeat0,MLP | 72.00 | 72.00 | 72.00 |
| repeat1,MLP | 71.00 | 70.01 | 70.39 |
| repeat2,MLP | 72.00 | 72.00 | 72.00 |
| repeat3,MLP | 73.00 | 75.40 | 73.74 |
| repeat4,MLP | 71.50 | 71.12 | 71.29 |
| repeat0,RF | 70.00 | 68.27 | 68.77 |
| repeat1,RF | 69.50 | 68.31 | 68.76 |
| repeat2,RF | 70.00 | 68.27 | 68.77 |
| repeat3,RF | 68.50 | 67.26 | 67.74 |
| repeat4,RF | 72.50 | 70.28 | 69.81 |

Table 10: Model evaluation results, GermanCredit.

### D.4.3 CFX GENERATION RESULTS

| Generator | Distance | Plausibility | Implausibility | Sensitivity $(10^{-1})$ | Stability |
|---|---|---|---|---|---|
| **MLP** | | | | | |
| MinDist | 1.65 ± 0.18 | 0.54 ± 0.19 | 0.71 ± 0.06 | 0.08 ± 0.02 | 0.58 ± 0.04 |
| Wachter | 0.41 ± 0.02 | 0.73 ± 0.05 | 0.59 ± 0.02 | 0.33 ± 0.09 | 0.24 ± 0.02 |
| Greedy | 0.99 ± 0.04 | -0.03 ± 0.09 | 0.80 ± 0.01 | 0.08 ± 0.02 | 0.68 ± 0.03 |
| ConfexNaive | | | | | |
| $\alpha = 0.01$ | 2.26 ± 0.14 | -0.16 ± 0.38 | 0.88 ± 0.06 | 0.03 ± 0.01 | 0.97 ± 0.02 |
| $\alpha = 0.05$ | 2.00 ± 0.07 | 0.16 ± 0.28 | 0.80 ± 0.08 | 0.05 ± 0.01 | 0.83 ± 0.07 |
| $\alpha = 0.1$ | 1.80 ± 0.04 | 0.40 ± 0.22 | 0.72 ± 0.05 | 0.06 ± 0.02 | 0.72 ± 0.09 |
| ECCCo | | | | | |
| $\alpha = 0.01$ | 1.01 ± 0.07 | 0.12 ± 0.11 | 0.77 ± 0.03 | 0.06 ± 0.02 | 0.73 ± 0.01 |
| $\alpha = 0.05$ | 1.00 ± 0.06 | 0.08 ± 0.14 | 0.77 ± 0.02 | 0.05 ± 0.02 | 0.73 ± 0.01 |
| $\alpha = 0.1$ | 0.97 ± 0.06 | 0.16 ± 0.11 | 0.75 ± 0.02 | 0.06 ± 0.02 | 0.72 ± 0.01 |
| ConfexTree, $\alpha = 0.01$ | | | | | |
| bw = 0.05 | nan ± nan | nan ± nan | nan ± nan | nan ± nan | nan ± nan |
| bw = 0.6 | nan ± nan | nan ± nan | nan ± nan | nan ± nan | nan ± nan |
| bw = 0.8 | 2.97 ± 0.35 | -0.58 ± 0.27 | 0.87 ± 0.10 | 0.02 ± 0.01 | 0.99 ± 0.01 |
| bw = 1 | 2.97 ± 0.35 | -0.57 ± 0.27 | 0.87 ± 0.10 | 0.02 ± 0.01 | 0.99 ± 0.01 |
| bw = 1.2 | 2.97 ± 0.21 | -0.60 ± 0.04 | 0.84 ± 0.07 | 0.03 ± 0.01 | 0.98 ± 0.00 |
| bw = 1.4 | 2.97 ± 0.21 | -0.60 ± 0.04 | 0.84 ± 0.07 | 0.03 ± 0.01 | 0.98 ± 0.00 |
| bw = 1.6 | 2.97 ± 0.21 | -0.60 ± 0.04 | 0.84 ± 0.07 | 0.03 ± 0.01 | 0.98 ± 0.00 |
| ConfexTree, $\alpha = 0.05$ | | | | | |
| bw = 0.1 | nan ± nan | nan ± nan | nan ± nan | nan ± nan | nan ± nan |
| bw = 0.2 | 2.31 ± 0.20 | 1.00 ± 0.00 | 0.28 ± 0.02 | 0.01 ± 0.01 | 0.88 ± 0.03 |
| bw = 0.4 | 1.60 ± 0.16 | 1.00 ± 0.00 | 0.44 ± 0.02 | 0.03 ± 0.01 | 0.69 ± 0.04 |
| bw = 0.6 | 1.48 ± 0.10 | 0.88 ± 0.04 | 0.59 ± 0.02 | 0.06 ± 0.02 | 0.65 ± 0.07 |
| bw = 0.8 | 1.91 ± 0.10 | 0.26 ± 0.27 | 0.75 ± 0.05 | 0.05 ± 0.01 | 0.79 ± 0.07 |
| bw = 1 | 1.94 ± 0.09 | 0.23 ± 0.29 | 0.76 ± 0.06 | 0.05 ± 0.01 | 0.80 ± 0.07 |
| bw = 1.2 | 2.00 ± 0.08 | 0.04 ± 0.32 | 0.79 ± 0.05 | 0.05 ± 0.00 | 0.81 ± 0.07 |
| bw = 1.4 | 2.01 ± 0.08 | 0.05 ± 0.31 | 0.80 ± 0.05 | 0.05 ± 0.00 | 0.81 ± 0.06 |
| bw = 1.6 | 2.01 ± 0.08 | 0.05 ± 0.31 | 0.80 ± 0.05 | 0.05 ± 0.00 | 0.81 ± 0.06 |
| ConfexTree, $\alpha = 0.1$ | | | | | |
| bw = 0.1 | 2.18 ± 0.21 | 1.00 ± 0.00 | 0.25 ± 0.03 | 0.01 ± 0.00 | 0.76 ± 0.11 |
| bw = 0.2 | 2.21 ± 0.71 | 1.00 ± 0.00 | 0.37 ± 0.01 | 0.01 ± 0.00 | 0.66 ± 0.08 |
| bw = 0.4 | 1.67 ± 0.15 | 0.83 ± 0.18 | 0.62 ± 0.05 | 0.07 ± 0.01 | 0.57 ± 0.08 |
| bw = 0.6 | 1.61 ± 0.08 | 0.84 ± 0.13 | 0.62 ± 0.04 | 0.07 ± 0.01 | 0.64 ± 0.07 |
| bw = 0.8 | 1.79 ± 0.12 | 0.45 ± 0.14 | 0.71 ± 0.05 | 0.07 ± 0.01 | 0.73 ± 0.06 |
| bw = 1 | 1.80 ± 0.10 | 0.47 ± 0.18 | 0.71 ± 0.05 | 0.07 ± 0.01 | 0.74 ± 0.06 |
| bw = 1.2 | 1.86 ± 0.03 | 0.31 ± 0.09 | 0.74 ± 0.01 | 0.07 ± 0.01 | 0.73 ± 0.04 |
| bw = 1.4 | 1.86 ± 0.03 | 0.28 ± 0.10 | 0.75 ± 0.02 | 0.07 ± 0.02 | 0.73 ± 0.05 |
| bw = 1.5 | 1.86 ± 0.03 | 0.28 ± 0.10 | 0.75 ± 0.02 | 0.07 ± 0.02 | 0.73 ± 0.05 |
| bw = 1.6 | 1.86 ± 0.03 | 0.28 ± 0.10 | 0.75 ± 0.02 | 0.07 ± 0.02 | 0.73 ± 0.05 |
| FACE | 0.69 ± 0.05 | 0.92 ± 0.05 | 0.45 ± 0.02 | 0.05 ± 0.01 | 0.43 ± 0.01 |
| C-CHVAE | 2.45 ± 0.13 | 0.80 ± 0.19 | 0.52 ± 0.05 | 0.07 ± 0.01 | 0.38 ± 0.06 |

Table 11: CFX generation results, GermanCredit, MLP. Methods with nan values had 100% failures. Validity 84% for all ECCCo methods, 82% for Greedy, 84% for Wachter, 74% for C-CHVAE.

| Generator | Distance | Plausibility | Implausibility | Sensitivity $(10^{-1})$ | Stability |
|---|---|---|---|---|---|
| **RandomForest** | | | | | |
| MinDist | 1.65 ± 0.06 | 0.52 ± 0.17 | 0.71 ± 0.05 | 0.09 ± 0.01 | 0.36 ± 0.01 |
| ConfexNaive | | | | | |
| $\alpha = 0.01$ | 1.62 ± 0.08 | 0.63 ± 0.10 | 0.66 ± 0.04 | 0.09 ± 0.01 | 0.43 ± 0.02 |
| $\alpha = 0.05$ | 1.62 ± 0.08 | 0.63 ± 0.10 | 0.66 ± 0.04 | 0.09 ± 0.01 | 0.43 ± 0.02 |
| $\alpha = 0.1$ | 1.62 ± 0.09 | 0.65 ± 0.09 | 0.66 ± 0.04 | 0.09 ± 0.01 | 0.43 ± 0.02 |
| ConfexTree, $\alpha = 0.01$ | | | | | |
| bw = 0.05 | nan ± nan | nan ± nan | nan ± nan | nan ± nan | nan ± nan |
| bw = 0.7 | 2.23 ± 0.11 | 0.45 ± 0.09 | 0.61 ± 0.03 | 0.08 ± 0.01 | 0.42 ± 0.02 |
| bw = 0.8 | 2.23 ± 0.11 | 0.46 ± 0.12 | 0.61 ± 0.03 | 0.07 ± 0.01 | 0.42 ± 0.02 |
| bw = 0.9 | 2.22 ± 0.11 | 0.44 ± 0.10 | 0.61 ± 0.03 | 0.07 ± 0.01 | 0.42 ± 0.03 |
| bw = 1 | 2.22 ± 0.11 | 0.44 ± 0.10 | 0.61 ± 0.03 | 0.07 ± 0.01 | 0.42 ± 0.03 |
| bw = 1.1 | 2.39 ± 0.00 | 0.28 ± 0.00 | 0.67 ± 0.00 | 0.09 ± 0.00 | 0.43 ± 0.00 |
| bw = 1.2 | 2.39 ± 0.00 | 0.28 ± 0.00 | 0.67 ± 0.00 | 0.09 ± 0.00 | 0.43 ± 0.00 |
| ConfexTree, $\alpha = 0.05$ | | | | | |
| bw = 0.1 | nan ± nan | nan ± nan | nan ± nan | nan ± nan | nan ± nan |
| bw = 0.2 | 2.04 ± 0.16 | 1.00 ± 0.00 | 0.35 ± 0.06 | 0.01 ± 0.01 | 0.44 ± 0.26 |
| bw = 0.3 | 2.12 ± 0.43 | 1.00 ± 0.00 | 0.39 ± 0.06 | 0.07 ± 0.02 | 0.38 ± 0.05 |
| bw = 0.4 | 1.73 ± 0.07 | 0.99 ± 0.02 | 0.45 ± 0.07 | 0.07 ± 0.05 | 0.35 ± 0.04 |
| bw = 0.5 | 1.88 ± 0.13 | 0.80 ± 0.08 | 0.54 ± 0.04 | 0.05 ± 0.01 | 0.39 ± 0.03 |
| bw = 0.6 | 1.40 ± 0.13 | 0.82 ± 0.13 | 0.57 ± 0.05 | 0.07 ± 0.01 | 0.39 ± 0.03 |
| bw = 0.7 | 1.63 ± 0.10 | 0.72 ± 0.14 | 0.61 ± 0.07 | 0.11 ± 0.03 | 0.41 ± 0.02 |
| bw = 0.8 | 1.71 ± 0.15 | 0.70 ± 0.16 | 0.63 ± 0.06 | 0.11 ± 0.03 | 0.43 ± 0.03 |
| bw = 0.9 | 1.65 ± 0.08 | 0.74 ± 0.09 | 0.62 ± 0.05 | 0.11 ± 0.02 | 0.43 ± 0.04 |
| bw = 1 | 2.10 ± 0.64 | 0.59 ± 0.15 | 0.71 ± 0.07 | 0.09 ± 0.02 | 0.45 ± 0.03 |
| ConfexTree, $\alpha = 0.1$ | | | | | |
| bw = 0.1 | 2.01 ± 0.07 | 1.00 ± 0.00 | 0.30 ± 0.05 | 0.01 ± 0.01 | 0.45 ± 0.24 |
| bw = 0.2 | 1.76 ± 0.17 | 1.00 ± 0.00 | 0.38 ± 0.02 | 0.04 ± 0.02 | 0.25 ± 0.06 |
| bw = 0.3 | 1.48 ± 0.23 | 0.99 ± 0.01 | 0.49 ± 0.04 | 0.10 ± 0.05 | 0.35 ± 0.03 |
| bw = 0.4 | 1.39 ± 0.08 | 0.87 ± 0.09 | 0.55 ± 0.02 | 0.08 ± 0.01 | 0.36 ± 0.02 |
| bw = 0.5 | 1.46 ± 0.10 | 0.76 ± 0.07 | 0.60 ± 0.03 | 0.09 ± 0.01 | 0.37 ± 0.03 |
| bw = 0.6 | 1.49 ± 0.12 | 0.77 ± 0.08 | 0.61 ± 0.01 | 0.13 ± 0.10 | 0.38 ± 0.02 |
| bw = 0.7 | 1.57 ± 0.10 | 0.67 ± 0.14 | 0.64 ± 0.05 | 0.10 ± 0.01 | 0.41 ± 0.01 |
| bw = 0.8 | 1.58 ± 0.09 | 0.66 ± 0.14 | 0.65 ± 0.04 | 0.09 ± 0.01 | 0.42 ± 0.01 |
| bw = 0.9 | 1.61 ± 0.12 | 0.64 ± 0.15 | 0.65 ± 0.05 | 0.09 ± 0.01 | 0.42 ± 0.01 |
| bw = 1 | 1.63 ± 0.11 | 0.66 ± 0.16 | 0.66 ± 0.05 | 0.09 ± 0.01 | 0.42 ± 0.02 |
| FeatureTweak | 0.50 ± 0.06 | 0.84 ± 0.05 | 0.55 ± 0.03 | 0.09 ± 0.01 | 0.20 ± 0.02 |
| FOCUS | 0.45 ± 0.14 | 0.83 ± 0.02 | 0.56 ± 0.02 | 0.58 ± 0.25 | 0.26 ± 0.02 |
| FACE | 0.59 ± 0.06 | 0.88 ± 0.07 | 0.48 ± 0.01 | 0.07 ± 0.01 | 0.22 ± 0.02 |

Table 12: CFX generation results, GermanCredit, RandomForest. Methods with nan values had 100% failures. Validity 50% for FeatureTweak.

### D.4.4 CONFORMAL EVALUATION RESULTS

| Generator | Marginal CovGap | Binning CovGap | Class Cond CovGap | Simulated CovGap |
|---|---|---|---|---|
| **MLP** | | | | |
| ConfexNaive | | | | |
| $\alpha = 0.01$ | 1.00 ± 0.00 | 1.00 ± 0.00 | 1.00 ± 0.00 | nan ± nan |
| $\alpha = 0.05$ | 1.00 ± 0.00 | 5.00 ± 0.00 | 5.00 ± 0.00 | nan ± nan |
| $\alpha = 0.1$ | 0.93 ± 0.03 | -0.81 ± 0.66 | 2.90 ± 0.42 | -23.40 ± 21.12 |
| ConfexTree, $\alpha = 0.01$ | | | | |
| bw = 0.05 | 1.00 ± 0.00 | 1.00 ± 0.00 | 1.00 ± 0.00 | nan ± nan |
| bw = 0.1 | 1.00 ± 0.00 | 1.00 ± 0.00 | 1.00 ± 0.00 | nan ± nan |
| bw = 0.2 | 1.00 ± 0.00 | 1.00 ± 0.00 | 1.00 ± 0.00 | nan ± nan |
| bw = 0.3 | 1.00 ± 0.00 | 1.00 ± 0.00 | 1.00 ± 0.00 | nan ± nan |
| bw = 0.4 | 1.00 ± 0.00 | 1.00 ± 0.00 | 1.00 ± 0.00 | nan ± nan |
| bw = 0.5 | 1.00 ± 0.00 | 1.00 ± 0.00 | 1.00 ± 0.00 | nan ± nan |
| bw = 0.6 | 1.00 ± 0.00 | 1.00 ± 0.00 | 1.00 ± 0.00 | nan ± nan |
| bw = 0.7 | 1.00 ± 0.00 | 1.00 ± 0.00 | 1.00 ± 0.00 | nan ± nan |
| bw = 0.8 | 1.00 ± 0.00 | 1.00 ± 0.00 | 1.00 ± 0.00 | nan ± nan |
| bw = 0.9 | 1.00 ± 0.00 | 1.00 ± 0.00 | 1.00 ± 0.00 | nan ± nan |
| bw = 1 | 1.00 ± 0.00 | 1.00 ± 0.00 | 1.00 ± 0.00 | nan ± nan |
| ConfexTree, $\alpha = 0.05$ | | | | |
| bw = 0.05 | 1.00 ± 0.00 | 5.00 ± 0.00 | 5.00 ± 0.00 | nan ± nan |
| bw = 0.1 | 1.00 ± 0.00 | 5.00 ± 0.00 | 5.00 ± 0.00 | nan ± nan |
| bw = 0.2 | 1.00 ± 0.00 | 5.00 ± 0.00 | 5.00 ± 0.00 | nan ± nan |
| bw = 0.3 | 1.00 ± 0.00 | 4.93 ± 0.16 | 4.90 ± 0.22 | 2.50 ± nan |
| bw = 0.4 | 0.99 ± 0.02 | 3.31 ± 1.53 | 3.80 ± 1.04 | -42.39 ± 38.61 |
| bw = 0.5 | 0.95 ± 0.05 | 2.58 ± 1.90 | 3.30 ± 1.44 | -17.85 ± 18.25 |
| bw = 0.6 | 0.95 ± 0.05 | 2.42 ± 1.75 | 3.20 ± 1.35 | -21.79 ± 16.32 |
| bw = 0.7 | 0.97 ± 0.03 | 2.23 ± 3.11 | 3.20 ± 2.14 | -29.83 ± 27.83 |
| bw = 0.8 | 0.97 ± 0.03 | 2.23 ± 3.11 | 3.20 ± 2.14 | -29.83 ± 27.83 |
| bw = 0.9 | 0.97 ± 0.03 | 2.23 ± 3.11 | 3.20 ± 2.14 | -28.95 ± 27.52 |
| bw = 1 | 0.97 ± 0.03 | 2.23 ± 3.11 | 3.20 ± 2.14 | -28.95 ± 27.52 |
| ConfexTree, $\alpha = 0.1$ | | | | |
| bw = 0.05 | 1.00 ± 0.00 | 10.00 ± 0.00 | 10.00 ± 0.00 | 10.00 ± 0.00 |
| bw = 0.1 | 1.00 ± 0.00 | 10.00 ± 0.00 | 10.00 ± 0.00 | 10.00 ± 0.00 |
| bw = 0.2 | 0.99 ± 0.02 | 8.51 ± 0.82 | 8.70 ± 0.76 | -1.36 ± 10.37 |
| bw = 0.3 | 0.95 ± 0.05 | 4.33 ± 0.60 | 5.70 ± 0.67 | -47.17 ± 26.96 |
| bw = 0.4 | 0.95 ± 0.06 | 1.56 ± 1.82 | 3.90 ± 1.64 | -44.65 ± 18.65 |
| bw = 0.5 | 0.91 ± 0.04 | -0.34 ± 0.58 | 2.40 ± 0.55 | -25.40 ± 19.69 |
| bw = 0.6 | 0.92 ± 0.04 | 0.76 ± 1.33 | 3.30 ± 1.15 | -28.40 ± 16.51 |
| bw = 0.7 | 0.93 ± 0.03 | -0.08 ± 1.43 | 3.40 ± 0.82 | -26.00 ± 22.49 |
| bw = 0.8 | 0.93 ± 0.03 | -0.08 ± 1.43 | 3.40 ± 0.82 | -26.00 ± 22.49 |
| bw = 0.9 | 0.93 ± 0.03 | -0.08 ± 1.43 | 3.40 ± 0.82 | -25.00 ± 22.10 |
| bw = 1 | 0.93 ± 0.03 | -0.08 ± 1.43 | 3.40 ± 0.82 | -25.00 ± 22.10 |

Table 13: Conformal evaluation results, GermanCredit, MLP

| Generator | Marginal CovGap | Binning CovGap | Class Cond CovGap | Simulated CovGap |
|---|---|---|---|---|
| **RandomForest** | | | | |
| ConfexNaive | | | | |
| $\alpha = 0.01$ | 1.00 ± 0.00 | 1.00 ± 0.00 | 1.00 ± 0.00 | nan ± nan |
| $\alpha = 0.05$ | 1.00 ± 0.00 | 5.00 ± 0.00 | 5.00 ± 0.00 | nan ± nan |
| $\alpha = 0.1$ | 0.92 ± 0.04 | -0.97 ± 1.23 | 2.75 ± 0.35 | -33.75 ± 12.37 |
| ConfexTree, $\alpha = 0.01$ | | | | |
| bw = 0.1 | 1.00 ± 0.00 | 1.00 ± 0.00 | 1.00 ± 0.00 | nan ± nan |
| bw = 0.2 | 1.00 ± 0.00 | 1.00 ± 0.00 | 1.00 ± 0.00 | nan ± nan |
| bw = 0.3 | 1.00 ± 0.00 | 1.00 ± 0.00 | 1.00 ± 0.00 | nan ± nan |
| bw = 0.4 | 1.00 ± 0.00 | 1.00 ± 0.00 | 1.00 ± 0.00 | nan ± nan |
| bw = 0.5 | 1.00 ± 0.00 | 1.00 ± 0.00 | 1.00 ± 0.00 | nan ± nan |
| bw = 0.6 | 1.00 ± 0.00 | 1.00 ± 0.00 | 1.00 ± 0.00 | nan ± nan |
| bw = 0.7 | 1.00 ± 0.00 | 1.00 ± 0.00 | 1.00 ± 0.00 | nan ± nan |
| bw = 0.8 | 1.00 ± 0.00 | 1.00 ± 0.00 | 1.00 ± 0.00 | nan ± nan |
| bw = 0.9 | 1.00 ± 0.00 | 1.00 ± 0.00 | 1.00 ± 0.00 | nan ± nan |
| bw = 1 | 1.00 ± 0.00 | 1.00 ± 0.00 | 1.00 ± 0.00 | nan ± nan |
| ConfexTree, $\alpha = 0.05$ | | | | |
| bw = 0.1 | 1.00 ± 0.00 | 5.00 ± 0.00 | 5.00 ± 0.00 | nan ± nan |
| bw = 0.2 | 1.00 ± 0.00 | 5.00 ± 0.00 | 5.00 ± 0.00 | nan ± nan |
| bw = 0.3 | 1.00 ± 0.00 | 4.82 ± 0.25 | 4.75 ± 0.35 | 2.50 ± nan |
| bw = 0.4 | 1.00 ± 0.00 | 4.82 ± 0.25 | 4.75 ± 0.35 | 2.50 ± nan |
| bw = 0.5 | 0.98 ± 0.04 | 2.13 ± 1.74 | 3.25 ± 1.06 | -16.31 ± 11.59 |
| bw = 0.6 | 0.98 ± 0.04 | 1.72 ± 1.16 | 3.00 ± 0.71 | -26.15 ± 2.32 |
| bw = 0.7 | 0.95 ± 0.00 | 0.13 ± 4.57 | 1.75 ± 3.18 | -51.18 ± 17.93 |
| bw = 0.8 | 0.95 ± 0.00 | 0.13 ± 4.57 | 1.75 ± 3.18 | -51.18 ± 17.93 |
| bw = 0.9 | 0.95 ± 0.00 | 0.13 ± 4.57 | 1.75 ± 3.18 | -49.43 ± 20.40 |
| bw = 1 | 0.95 ± 0.00 | 0.13 ± 4.57 | 1.75 ± 3.18 | -49.43 ± 20.40 |
| ConfexTree, $\alpha = 0.1$ | | | | |
| bw = 0.1 | 1.00 ± 0.00 | 10.00 ± 0.00 | 10.00 ± 0.00 | 10.00 ± nan |
| bw = 0.2 | 1.00 ± 0.00 | 9.82 ± 0.25 | 9.75 ± 0.35 | 7.50 ± nan |
| bw = 0.3 | 0.98 ± 0.04 | 5.59 ± 1.23 | 6.75 ± 0.35 | -36.75 ± 49.85 |
| bw = 0.4 | 0.98 ± 0.04 | 3.31 ± 0.33 | 5.50 ± 0.00 | -36.75 ± 48.44 |
| bw = 0.5 | 0.95 ± 0.00 | -0.10 ± 2.32 | 3.00 ± 1.41 | -23.75 ± 22.27 |
| bw = 0.6 | 0.95 ± 0.00 | -0.51 ± 3.41 | 2.75 ± 2.47 | -19.50 ± 12.73 |
| bw = 0.7 | 0.92 ± 0.04 | -0.79 ± 1.48 | 3.00 ± 0.71 | -34.00 ± 12.02 |
| bw = 0.8 | 0.92 ± 0.04 | -0.79 ± 1.48 | 3.00 ± 0.71 | -34.00 ± 12.02 |
| bw = 0.9 | 0.92 ± 0.04 | -0.79 ± 1.48 | 3.00 ± 0.71 | -32.75 ± 13.79 |
| bw = 1 | 0.92 ± 0.04 | -0.79 ± 1.48 | 3.00 ± 0.71 | -32.75 ± 13.79 |

Table 14: Conformal evaluation results, GermanCredit, RandomForest

## D.5 GIVEMESOMECREDIT

This dataset, obtained through Kaggle[7], contains credit scoring data with 8 numeric features that were scaled to $(0, 1)$ using MinMax scaling.

We find that CONFEX-Tree methods outperform all other methods on plausibility and sensitivity. Our results roughly show the same pattern that distance decreases as the kernel bandwidth increases, however plausibility remains quite good for all tested (small) choices of kernel bandwidth.

### D.5.1 PLOTS

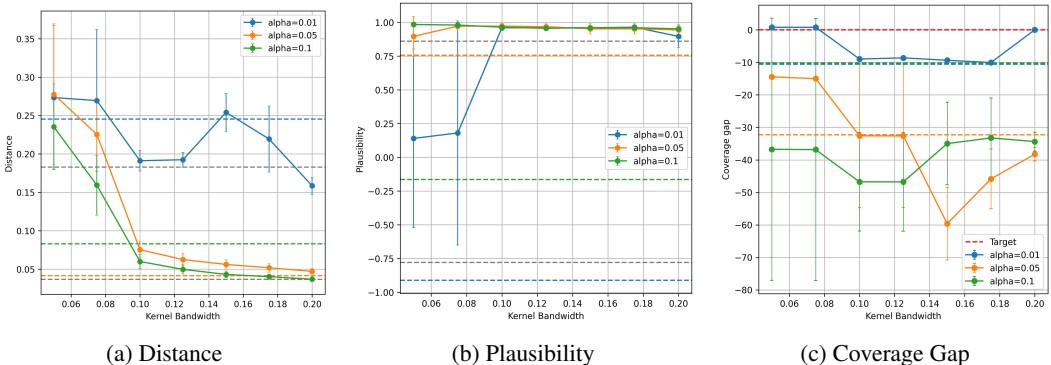

(a) Distance      (b) Plausibility      (c) Coverage Gap

Figure 13: Effect of coverage rate and kernel bandwidth on metrics for CONFEX-Tree on the GiveMeSomeCredit dataset, MLP. CONFEX-Naive is represented by dashed horizontal lines.

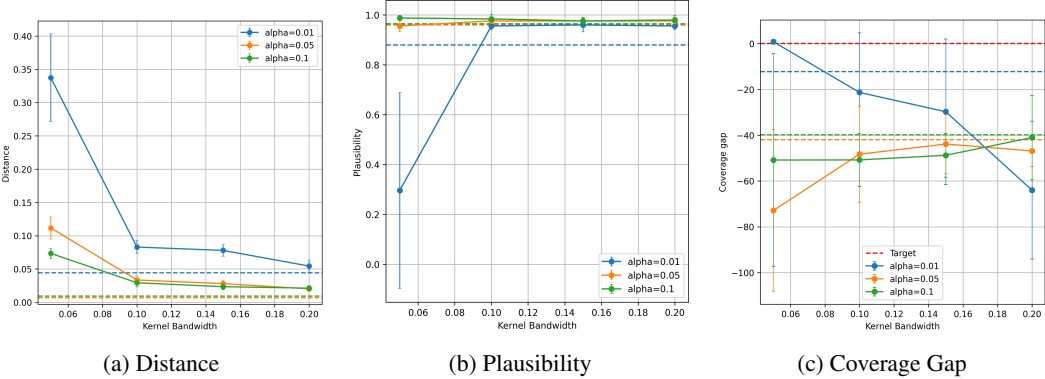

(a) Distance      (b) Plausibility      (c) Coverage Gap

Figure 14: Effect of coverage rate and kernel bandwidth on metrics for CONFEX-Tree on the GiveMeSomeCredit dataset, RandomForest. CONFEX-Naive is represented by dashed horizontal lines.

---

[7]https://www.kaggle.com/competitions/GiveMeSomeCredit

### D.5.2  MODEL EVALUATION RESULTS

| Repeat | Accuracy (%) | Precision (%) | F1 Score (%) |
|---|---|---|---|
| repeat0,MLP | 93.54 | 91.82 | 91.81 |
| repeat1,MLP | 93.49 | 91.79 | 91.96 |
| repeat2,MLP | 93.61 | 91.98 | 91.87 |
| repeat3,MLP | 93.57 | 91.89 | 91.72 |
| repeat4,MLP | 93.48 | 91.73 | 91.84 |
| repeat0,RF | 93.40 | 91.57 | 91.78 |
| repeat1,RF | 93.40 | 91.53 | 91.69 |
| repeat2,RF | 93.41 | 91.61 | 91.82 |
| repeat3,RF | 93.37 | 91.49 | 91.68 |
| repeat4,RF | 93.53 | 91.83 | 91.89 |

Table 15: Model evaluation results, GiveMeSomeCredit.

### D.5.3 CFX GENERATION RESULTS

| Generator | Distance | Plausibility | Implausibility | Sensitivity $(10^{-1})$ | Stability |
|---|---|---|---|---|---|
| **MLP** | | | | | |
| MinDist | 0.03 ± 0.00 | 0.93 ± 0.04 | 0.09 ± 0.00 | 1.39 ± 0.57 | 0.18 ± 0.06 |
| Wachter | 0.09 ± 0.01 | 0.93 ± 0.02 | 0.09 ± 0.00 | 0.95 ± 0.07 | 0.18 ± 0.07 |
| Greedy | 0.13 ± 0.07 | -0.02 ± 0.38 | 0.14 ± 0.03 | 1.05 ± 0.81 | 0.17 ± 0.06 |
| ConfexNaive | | | | | |
| $\alpha = 0.01$ | 0.25 ± 0.01 | -0.91 ± 0.05 | 0.21 ± 0.01 | 0.21 ± 0.05 | 0.18 ± 0.11 |
| $\alpha = 0.05$ | 0.04 ± 0.00 | 0.76 ± 0.05 | 0.09 ± 0.00 | 0.78 ± 0.11 | 0.18 ± 0.06 |
| $\alpha = 0.1$ | 0.08 ± 0.01 | -0.16 ± 0.32 | 0.11 ± 0.01 | 0.52 ± 0.09 | 0.16 ± 0.06 |
| ECCCo | | | | | |
| $\alpha = 0.01$ | 0.71 ± 0.28 | -0.98 ± 0.03 | 0.32 ± 0.15 | 0.21 ± 0.06 | 0.20 ± 0.08 |
| $\alpha = 0.05$ | 0.69 ± 0.28 | -0.97 ± 0.03 | 0.31 ± 0.14 | 0.21 ± 0.06 | 0.20 ± 0.08 |
| $\alpha = 0.1$ | 0.69 ± 0.28 | -0.97 ± 0.03 | 0.31 ± 0.14 | 0.21 ± 0.06 | 0.20 ± 0.07 |
| ConfexTree, $\alpha = 0.01$ | | | | | |
| bw = 0.05 | 0.27 ± 0.09 | 0.14 ± 0.66 | 0.08 ± 0.00 | nan ± nan | 0.07 ± 0.05 |
| bw = 0.075 | 0.27 ± 0.09 | 0.18 ± 0.83 | 0.08 ± 0.00 | nan ± nan | 0.07 ± 0.05 |
| bw = 0.1 | 0.19 ± 0.01 | 0.96 ± 0.02 | 0.07 ± 0.00 | 0.09 ± 0.02 | 0.20 ± 0.08 |
| bw = 0.125 | 0.19 ± 0.01 | 0.96 ± 0.01 | 0.07 ± 0.00 | 0.10 ± 0.03 | 0.20 ± 0.07 |
| bw = 0.15 | 0.25 ± 0.02 | 0.96 ± 0.01 | 0.08 ± 0.00 | 0.09 ± 0.03 | 0.18 ± 0.06 |
| bw = 0.175 | 0.22 ± 0.04 | 0.96 ± 0.01 | 0.08 ± 0.00 | 0.11 ± 0.04 | 0.19 ± 0.07 |
| bw = 0.2 | 0.16 ± 0.01 | 0.90 ± 0.08 | 0.08 ± 0.00 | 0.19 ± 0.02 | 0.18 ± 0.07 |
| bw = 0.25 | 0.29 ± 0.02 | 0.88 ± 0.15 | 0.11 ± 0.01 | 0.10 ± 0.02 | 0.17 ± 0.07 |
| ConfexTree, $\alpha = 0.05$ | | | | | |
| bw = 0.05 | 0.28 ± 0.09 | 0.90 ± 0.15 | 0.06 ± 0.01 | 0.10 ± 0.09 | 0.25 ± 0.07 |
| bw = 0.075 | 0.23 ± 0.05 | 0.97 ± 0.02 | 0.07 ± 0.01 | 0.14 ± 0.04 | 0.22 ± 0.09 |
| bw = 0.1 | 0.08 ± 0.01 | 0.97 ± 0.02 | 0.08 ± 0.00 | 0.47 ± 0.08 | 0.19 ± 0.07 |
| bw = 0.125 | 0.06 ± 0.01 | 0.97 ± 0.02 | 0.08 ± 0.00 | 0.53 ± 0.09 | 0.19 ± 0.07 |
| bw = 0.15 | 0.06 ± 0.01 | 0.95 ± 0.04 | 0.08 ± 0.00 | 0.56 ± 0.07 | 0.19 ± 0.07 |
| bw = 0.175 | 0.05 ± 0.01 | 0.95 ± 0.04 | 0.08 ± 0.00 | 0.59 ± 0.07 | 0.19 ± 0.07 |
| bw = 0.2 | 0.05 ± 0.00 | 0.94 ± 0.04 | 0.09 ± 0.00 | 0.62 ± 0.13 | 0.18 ± 0.07 |
| bw = 0.25 | 0.05 ± 0.00 | 0.91 ± 0.02 | 0.09 ± 0.00 | 0.70 ± 0.31 | 0.18 ± 0.07 |
| ConfexTree, $\alpha = 0.1$ | | | | | |
| bw = 0.05 | 0.24 ± 0.06 | 0.98 ± 0.01 | 0.08 ± 0.01 | 0.14 ± 0.03 | 0.22 ± 0.09 |
| bw = 0.075 | 0.16 ± 0.04 | 0.98 ± 0.02 | 0.08 ± 0.01 | 0.27 ± 0.07 | 0.20 ± 0.07 |
| bw = 0.1 | 0.06 ± 0.01 | 0.96 ± 0.01 | 0.08 ± 0.00 | 0.59 ± 0.09 | 0.19 ± 0.07 |
| bw = 0.125 | 0.05 ± 0.01 | 0.96 ± 0.01 | 0.08 ± 0.00 | 0.64 ± 0.08 | 0.19 ± 0.07 |
| bw = 0.15 | 0.04 ± 0.00 | 0.96 ± 0.01 | 0.09 ± 0.00 | 0.71 ± 0.11 | 0.18 ± 0.07 |
| bw = 0.175 | 0.04 ± 0.00 | 0.96 ± 0.01 | 0.09 ± 0.00 | 0.73 ± 0.13 | 0.18 ± 0.07 |
| bw = 0.2 | 0.04 ± 0.00 | 0.95 ± 0.02 | 0.09 ± 0.00 | 0.84 ± 0.33 | 0.18 ± 0.07 |
| bw = 0.25 | 0.03 ± 0.00 | 0.95 ± 0.02 | 0.09 ± 0.00 | 0.85 ± 0.33 | 0.18 ± 0.07 |
| FACE | 0.12 ± 0.01 | 0.95 ± 0.02 | 0.09 ± 0.00 | 0.35 ± 0.04 | 0.19 ± 0.07 |
| C-CHVAE | 1.32 ± 0.08 | -0.92 ± 0.06 | 0.69 ± 0.05 | 0.09 ± 0.02 | 0.27 ± 0.20 |

Table 16: CFX generation results, GiveMeSomeCredit, MLP. Validity 71% for Wachter and 80% for Schut.

| Generator | Distance | Plausibility | Implausibility | Sensitivity ($10^{-1}$) | Stability |
|---|---|---|---|---|---|
| **RandomForest** | | | | | |
| MinDist | 0.01 ± 0.00 | 0.96 ± 0.01 | 0.09 ± 0.00 | 96.79 ± 35.89 | 0.22 ± 0.07 |
| ConfexNaive | | | | | |
| $\alpha = 0.01$ | 0.04 ± 0.01 | 0.88 ± 0.07 | 0.09 ± 0.00 | 1.28 ± 0.10 | 0.22 ± 0.07 |
| $\alpha = 0.05$ | 0.01 ± 0.00 | 0.96 ± 0.01 | 0.09 ± 0.00 | 67.00 ± 45.36 | 0.22 ± 0.07 |
| $\alpha = 0.1$ | 0.01 ± 0.00 | 0.96 ± 0.01 | 0.09 ± 0.00 | 28.68 ± 25.01 | 0.22 ± 0.07 |
| ConfexTree, $\alpha = 0.01$ | | | | | |
| bw = 0.05 | 0.34 ± 0.07 | 0.30 ± 0.39 | 0.07 ± 0.01 | nan ± nan | 0.29 ± 0.07 |
| bw = 0.1 | 0.08 ± 0.01 | 0.96 ± 0.01 | 0.07 ± 0.00 | 0.50 ± 0.20 | 0.22 ± 0.07 |
| bw = 0.15 | 0.08 ± 0.01 | 0.96 ± 0.03 | 0.07 ± 0.00 | 0.60 ± 0.22 | 0.22 ± 0.07 |
| bw = 0.2 | 0.05 ± 0.01 | 0.96 ± 0.01 | 0.08 ± 0.00 | 0.73 ± 0.14 | 0.22 ± 0.07 |
| ConfexTree, $\alpha = 0.05$ | | | | | |
| bw = 0.05 | 0.11 ± 0.02 | 0.96 ± 0.02 | 0.07 ± 0.00 | 0.34 ± 0.08 | 0.22 ± 0.06 |
| bw = 0.1 | 0.03 ± 0.01 | 0.98 ± 0.01 | 0.08 ± 0.00 | 2.90 ± 2.12 | 0.22 ± 0.07 |
| bw = 0.15 | 0.03 ± 0.00 | 0.98 ± 0.01 | 0.08 ± 0.00 | 3.35 ± 2.19 | 0.22 ± 0.07 |
| bw = 0.2 | 0.02 ± 0.00 | 0.98 ± 0.01 | 0.08 ± 0.00 | 2.85 ± 1.09 | 0.22 ± 0.07 |
| ConfexTree, $\alpha = 0.1$ | | | | | |
| bw = 0.05 | 0.07 ± 0.01 | 0.99 ± 0.01 | 0.08 ± 0.00 | 0.55 ± 0.14 | 0.22 ± 0.06 |
| bw = 0.1 | 0.03 ± 0.01 | 0.98 ± 0.02 | 0.08 ± 0.00 | 2.20 ± 1.93 | 0.22 ± 0.07 |
| bw = 0.15 | 0.02 ± 0.00 | 0.98 ± 0.01 | 0.08 ± 0.00 | 3.71 ± 2.69 | 0.22 ± 0.07 |
| bw = 0.2 | 0.02 ± 0.00 | 0.98 ± 0.02 | 0.09 ± 0.00 | 4.14 ± 2.67 | 0.22 ± 0.07 |
| FeatureTweak | 0.03 ± 0.01 | 0.96 ± 0.02 | 0.09 ± 0.00 | 1.40 ± 0.16 | 0.20 ± 0.06 |
| FOCUS | 0.05 ± 0.00 | 0.91 ± 0.05 | 0.09 ± 0.00 | 2.27 ± 0.79 | 0.22 ± 0.07 |
| FACE | 0.10 ± 0.01 | 0.97 ± 0.01 | 0.08 ± 0.00 | 0.46 ± 0.08 | 0.22 ± 0.07 |

Table 17: CFX generation results, GiveMeSomeCredit, RF. ConfexTree with alpha=0.01,bw=0.05 had 78% failures (i.e. one class had no singleton regions). Valdiity 50% for FeatureTweak.

### D.5.4 CONFORMAL EVALUATION RESULTS

| Generator | Marginal CovGap | Binning CovGap | Class Cond CovGap | Simulated CovGap |
|---|---|---|---|---|
| **MLP** | | | | |
| ConfexNaive | | | | |
| $\alpha = 0.01$ | 0.99 ± 0.00 | -5.24 ± 0.05 | 0.15 ± 0.01 | -10.52 ± 5.95 |
| $\alpha = 0.05$ | 0.96 ± 0.00 | -30.19 ± 0.86 | 0.02 ± 0.13 | -32.31 ± 9.63 |
| $\alpha = 0.1$ | 0.90 ± 0.00 | -41.43 ± 0.21 | -0.04 ± 0.04 | -10.15 ± 15.42 |
| ConfexTree, $\alpha = 0.01$ | | | | |
| bw = 0.05 | 1.00 ± 0.00 | 0.98 ± 0.00 | 1.00 ± 0.00 | 0.80 ± 0.00 |
| bw = 0.075 | 1.00 ± 0.00 | 0.98 ± 0.00 | 1.00 ± 0.00 | 0.80 ± 0.00 |
| bw = 0.1 | 1.00 ± 0.00 | -1.84 ± 0.09 | 0.61 ± 0.01 | -8.96 ± 0.20 |
| bw = 0.125 | 1.00 ± 0.00 | -1.73 ± 0.08 | 0.63 ± 0.01 | -8.60 ± 0.17 |
| bw = 0.15 | 1.00 ± 0.00 | -2.03 ± 0.09 | 0.59 ± 0.01 | -9.35 ± 0.18 |
| bw = 0.175 | 1.00 ± 0.00 | -2.15 ± 0.09 | 0.57 ± 0.01 | -10.04 ± 0.32 |
| bw = 0.2 | 1.00 ± 0.00 | -3.88 ± 0.16 | 0.34 ± 0.02 | 0.04 ± 0.03 |
| bw = 0.25 | 1.00 ± 0.00 | -4.21 ± 0.09 | 0.29 ± 0.01 | 0.07 ± 0.03 |
| bw = 0.4 | 0.99 ± 0.00 | -5.53 ± 0.10 | 0.11 ± 0.01 | -26.76 ± 39.37 |
| ConfexTree, $\alpha = 0.05$ | | | | |
| bw = 0.05 | 1.00 ± 0.00 | -0.48 ± 0.02 | 4.25 ± 0.00 | -14.45 ± 0.11 |
| bw = 0.075 | 1.00 ± 0.00 | -0.70 ± 0.02 | 4.22 ± 0.00 | -14.99 ± 0.11 |
| bw = 0.1 | 0.98 ± 0.00 | -11.87 ± 0.08 | 2.70 ± 0.01 | -32.61 ± 22.10 |
| bw = 0.125 | 0.98 ± 0.00 | -11.94 ± 0.08 | 2.69 ± 0.01 | -32.61 ± 22.10 |
| bw = 0.15 | 0.98 ± 0.00 | -14.15 ± 0.13 | 2.36 ± 0.02 | -59.62 ± 11.20 |
| bw = 0.175 | 0.98 ± 0.00 | -14.64 ± 0.14 | 2.30 ± 0.01 | -45.84 ± 9.20 |
| bw = 0.2 | 0.97 ± 0.00 | -18.83 ± 0.30 | 1.67 ± 0.03 | -38.26 ± 2.07 |
| ConfexTree, $\alpha = 0.1$ | | | | |
| bw = 0.05 | 0.99 ± 0.00 | -1.06 ± 0.05 | 8.48 ± 0.01 | -36.74 ± 40.35 |
| bw = 0.075 | 0.99 ± 0.00 | -1.37 ± 0.05 | 8.44 ± 0.01 | -36.85 ± 40.31 |
| bw = 0.1 | 0.97 ± 0.00 | -14.40 ± 0.13 | 6.63 ± 0.01 | -46.74 ± 15.17 |
| bw = 0.125 | 0.97 ± 0.00 | -14.62 ± 0.13 | 6.60 ± 0.01 | -46.78 ± 15.17 |
| bw = 0.15 | 0.96 ± 0.00 | -18.02 ± 0.22 | 6.07 ± 0.02 | -34.99 ± 12.69 |
| bw = 0.175 | 0.96 ± 0.00 | -18.51 ± 0.23 | 6.01 ± 0.02 | -33.24 ± 12.32 |
| bw = 0.2 | 0.96 ± 0.00 | -22.20 ± 0.37 | 5.42 ± 0.02 | -34.36 ± 2.89 |

Table 18: Conformal evaluation results, GiveMeSomeCredit, MLP

| Generator | Marginal CovGap | Binning CovGap | Class Cond CovGap | Simulated CovGap |
|---|---|---|---|---|
| ConfexNaive | | | | |
| $\alpha = 0.01$ | 0.99 ± 0.00 | -6.14 ± 0.33 | 0.03 ± 0.04 | -12.25 ± 6.80 |
| $\alpha = 0.05$ | 0.95 ± 0.00 | -30.33 ± 0.54 | -0.12 ± 0.09 | -41.99 ± 4.34 |
| $\alpha = 0.1$ | 0.90 ± 0.00 | -40.10 ± 0.33 | -0.09 ± 0.07 | -39.82 ± 6.80 |
| ConfexTree, $\alpha = 0.01$ | | | | |
| bw = 0.05 | 1.00 ± 0.00 | 0.98 ± 0.01 | 1.00 ± 0.00 | 0.84 ± 0.09 |
| bw = 0.1 | 1.00 ± 0.00 | -1.90 ± 0.09 | 0.57 ± 0.02 | -21.32 ± 26.00 |
| bw = 0.15 | 1.00 ± 0.00 | -1.93 ± 0.10 | 0.56 ± 0.02 | -29.74 ± 31.74 |
| bw = 0.2 | 0.99 ± 0.00 | -3.46 ± 0.06 | 0.35 ± 0.02 | -64.01 ± 30.10 |
| ConfexTree, $\alpha = 0.05$ | | | | |
| bw = 0.05 | 0.99 ± 0.00 | -0.74 ± 0.11 | 3.59 ± 0.06 | -72.90 ± 35.28 |
| bw = 0.1 | 0.97 ± 0.00 | -12.33 ± 0.29 | 1.52 ± 0.09 | -48.29 ± 20.97 |
| bw = 0.15 | 0.97 ± 0.00 | -14.46 ± 0.30 | 1.20 ± 0.09 | -43.87 ± 12.96 |
| bw = 0.2 | 0.96 ± 0.00 | -19.58 ± 0.17 | 0.53 ± 0.07 | -46.87 ± 6.75 |
| ConfexTree, $\alpha = 0.1$ | | | | |
| bw = 0.05 | 0.96 ± 0.00 | -2.32 ± 0.17 | 5.78 ± 0.10 | -50.85 ± 46.46 |
| bw = 0.1 | 0.93 ± 0.00 | -16.78 ± 0.11 | 2.27 ± 0.09 | -50.79 ± 11.62 |
| bw = 0.15 | 0.92 ± 0.00 | -20.68 ± 0.12 | 1.64 ± 0.10 | -48.78 ± 9.57 |
| bw = 0.2 | 0.91 ± 0.00 | -25.84 ± 0.14 | 0.96 ± 0.09 | -41.02 ± 18.42 |

Table 19: Conformal evaluation results, GiveMeSomeCredit, RandomForest

### D.6 ADULTINCOME

This dataset Becker & Kohavi (1996), obtained through Kaggle[8], predicts whether an individual's income exceeds $50,000. We processed the following features: numeric features (Age, Capital Gain, Capital Loss, Hours per week) scaled to $(0, 1)$, ordinal features (education), and categorical features (Workclass, Occupation, Race, Relationship, Gender, Marital status) using one-hot encoding. In CONFEXTree, to avoid splitting over many categorical feature combinations, we consider the first (Workclass) as a feature to split by and do not split the rest.

For this dataset we find that CONFEX-Tree methods generally obtain worse plausibility than competing methods (although we have comparable sensitivity). This could be attributed to an insufficient kernel: further tuning to obtain a better kernel (bandwidth, features contributing to the kernel) etc. to better define locality could help with this.

#### D.6.1 PLOTS

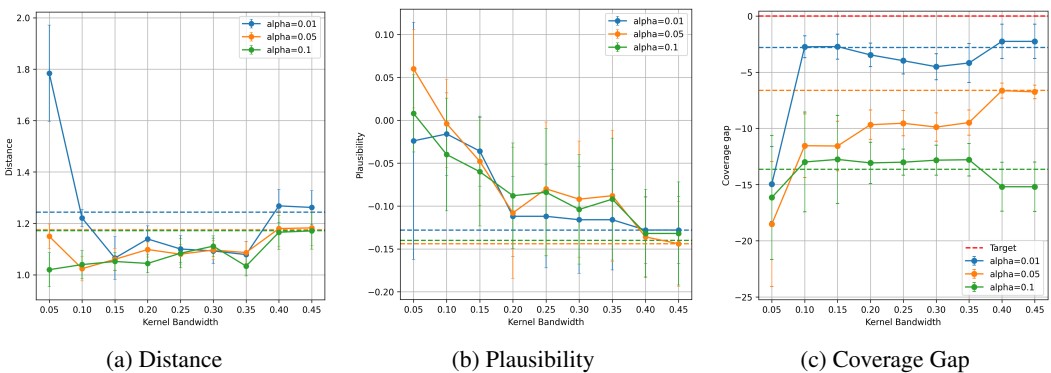

| (a) Distance | (b) Plausibility | (c) Coverage Gap |

Figure 15: Effect of coverage rate and kernel bandwidth on metrics for CONFEX-Tree on the Adult-Income dataset, MLP. CONFEX-Naive is represented by dashed horizontal lines.

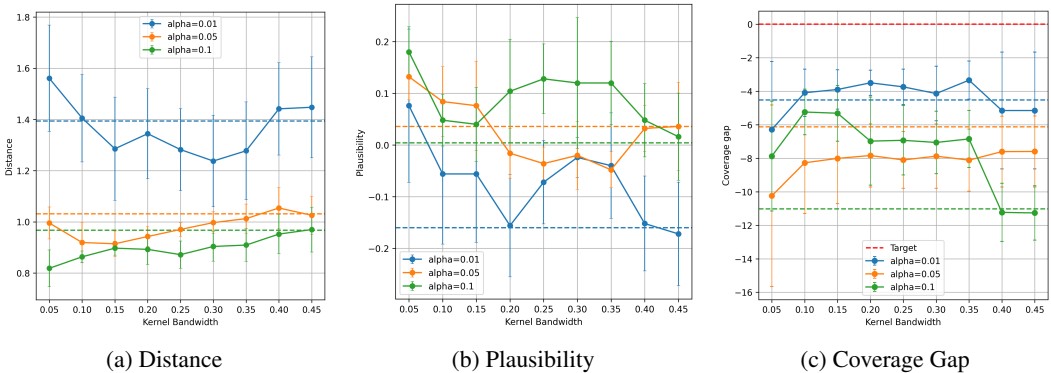

| (a) Distance | (b) Plausibility | (c) Coverage Gap |

Figure 16: Effect of coverage rate and kernel bandwidth on metrics for CONFEX-Tree on the Adult-Income dataset, RandomForest. CONFEX-Naive is represented by dashed horizontal lines.

---

[8]https://www.kaggle.com/datasets/wenruliu/adult-income-dataset

### D.6.2 MODEL EVALUATION RESULTS

| Repeat | Accuracy (%) | Precision (%) | F1 Score (%) |
|---|---|---|---|
| repeat0,MLP | 85.41 | 85.05 | 85.17 |
| repeat1,MLP | 85.04 | 84.70 | 84.83 |
| repeat2,MLP | 84.89 | 84.32 | 84.40 |
| repeat3,MLP | 85.03 | 84.58 | 84.71 |
| repeat4,MLP | 84.96 | 84.35 | 84.29 |
| repeat0,RF | 85.73 | 85.20 | 85.14 |
| repeat1,RF | 85.32 | 84.76 | 84.72 |
| repeat2,RF | 85.70 | 85.18 | 85.06 |
| repeat3,RF | 85.51 | 84.96 | 84.87 |
| repeat4,RF | 85.48 | 84.93 | 84.89 |

Table 20: Model evaluation results, AdultIncome.

### D.6.3 CFX GENERATION RESULTS

| Generator | Distance | Plausibility | Implausibility | Sensitivity $(10^{-1})$ | Stability |
|---|---|---|---|---|---|
| **MLP** | | | | | |
| MinDist | 1.16 ± 0.07 | -0.13 ± 0.05 | 1.97 ± 0.04 | 0.11 ± 0.11 | 0.33 ± 0.02 |
| Wachter | 0.39 ± 0.05 | 0.36 ± 0.02 | 1.87 ± 0.01 | 0.14 ± 0.01 | 0.14 ± 0.02 |
| Greedy | 0.95 ± 0.08 | 0.02 ± 0.10 | 2.06 ± 0.04 | 24190.61 ± 48381.15 | 0.86 ± 0.03 |
| ConfexNaive | | | | | |
| $\alpha = 0.01$ | 1.24 ± 0.07 | -0.13 ± 0.04 | 1.97 ± 0.04 | 0.05 ± 0.01 | 0.44 ± 0.04 |
| $\alpha = 0.05$ | 1.17 ± 0.07 | -0.14 ± 0.05 | 1.97 ± 0.04 | 0.05 ± 0.01 | 0.37 ± 0.02 |
| $\alpha = 0.1$ | 1.17 ± 0.06 | -0.14 ± 0.04 | 1.97 ± 0.04 | 0.08 ± 0.05 | 0.35 ± 0.02 |
| ECCCo | | | | | |
| $\alpha = 0.01$ | 0.57 ± 0.01 | 0.13 ± 0.01 | 1.88 ± 0.01 | 0.05 ± 0.00 | 0.37 ± 0.01 |
| $\alpha = 0.01$ | 0.73 ± 0.11 | -0.05 ± 0.07 | 1.87 ± 0.06 | 0.05 ± 0.00 | 0.63 ± 0.14 |
| $\alpha = 0.05$ | 0.57 ± 0.00 | 0.12 ± 0.02 | 1.88 ± 0.01 | 0.05 ± 0.00 | 0.37 ± 0.02 |
| $\alpha = 0.05$ | 0.72 ± 0.11 | -0.04 ± 0.06 | 1.87 ± 0.06 | 0.05 ± 0.01 | 0.61 ± 0.15 |
| $\alpha = 0.1$ | 0.56 ± 0.01 | 0.12 ± 0.02 | 1.88 ± 0.01 | 0.05 ± 0.00 | 0.37 ± 0.02 |
| $\alpha = 0.1$ | 0.72 ± 0.11 | -0.04 ± 0.06 | 1.87 ± 0.06 | 0.05 ± 0.01 | 0.61 ± 0.15 |
| ConfexTree, $\alpha = 0.01$ | | | | | |
| bw = 0.05 | 1.78 ± 0.19 | -0.02 ± 0.14 | 1.80 ± 0.04 | 0.05 ± 0.02 | 0.29 ± 0.02 |
| bw = 0.1 | 1.22 ± 0.03 | -0.02 ± 0.05 | 1.84 ± 0.04 | 0.06 ± 0.01 | 0.34 ± 0.02 |
| bw = 0.15 | 1.06 ± 0.08 | -0.04 ± 0.04 | 1.86 ± 0.05 | 0.05 ± 0.01 | 0.36 ± 0.04 |
| bw = 0.2 | 1.14 ± 0.05 | -0.11 ± 0.05 | 1.93 ± 0.05 | 0.05 ± 0.01 | 0.36 ± 0.03 |
| bw = 0.25 | 1.10 ± 0.05 | -0.11 ± 0.06 | 1.93 ± 0.04 | 0.05 ± 0.01 | 0.36 ± 0.03 |
| bw = 0.3 | 1.09 ± 0.05 | -0.12 ± 0.06 | 1.94 ± 0.04 | 0.05 ± 0.01 | 0.35 ± 0.03 |
| bw = 0.35 | 1.08 ± 0.05 | -0.12 ± 0.06 | 1.93 ± 0.04 | 0.05 ± 0.01 | 0.37 ± 0.03 |
| bw = 0.4 | 1.27 ± 0.06 | -0.13 ± 0.04 | 1.97 ± 0.04 | 0.05 ± 0.01 | 0.46 ± 0.04 |
| bw = 0.45 | 1.26 ± 0.07 | -0.13 ± 0.04 | 1.97 ± 0.04 | 0.04 ± 0.01 | 0.46 ± 0.04 |
| ConfexTree, $\alpha = 0.05$ | | | | | |
| bw = 0.05 | 1.15 ± 0.05 | 0.06 ± 0.05 | 1.87 ± 0.03 | 0.08 ± 0.03 | 0.28 ± 0.02 |
| bw = 0.1 | 1.02 ± 0.05 | -0.00 ± 0.05 | 1.91 ± 0.05 | 0.07 ± 0.02 | 0.30 ± 0.02 |
| bw = 0.15 | 1.06 ± 0.04 | -0.05 ± 0.05 | 1.93 ± 0.04 | 0.07 ± 0.01 | 0.33 ± 0.03 |
| bw = 0.2 | 1.10 ± 0.03 | -0.11 ± 0.08 | 1.95 ± 0.04 | 0.06 ± 0.02 | 0.35 ± 0.02 |
| bw = 0.25 | 1.08 ± 0.04 | -0.08 ± 0.08 | 1.95 ± 0.03 | 0.07 ± 0.02 | 0.34 ± 0.02 |
| bw = 0.3 | 1.10 ± 0.04 | -0.09 ± 0.07 | 1.95 ± 0.03 | 0.07 ± 0.02 | 0.35 ± 0.02 |
| bw = 0.35 | 1.09 ± 0.04 | -0.09 ± 0.08 | 1.94 ± 0.03 | 0.06 ± 0.03 | 0.35 ± 0.02 |
| bw = 0.4 | 1.18 ± 0.07 | -0.14 ± 0.05 | 1.97 ± 0.04 | 0.05 ± 0.01 | 0.37 ± 0.02 |
| bw = 0.45 | 1.18 ± 0.07 | -0.14 ± 0.05 | 1.97 ± 0.04 | 0.05 ± 0.01 | 0.37 ± 0.02 |
| ConfexTree, $\alpha = 0.1$ | | | | | |
| bw = 0.05 | 1.02 ± 0.07 | 0.01 ± 0.04 | 1.91 ± 0.04 | 0.08 ± 0.01 | 0.27 ± 0.02 |
| bw = 0.1 | 1.04 ± 0.06 | -0.04 ± 0.07 | 1.93 ± 0.04 | 0.08 ± 0.02 | 0.31 ± 0.02 |
| bw = 0.15 | 1.05 ± 0.04 | -0.06 ± 0.06 | 1.93 ± 0.04 | 0.08 ± 0.02 | 0.32 ± 0.02 |
| bw = 0.2 | 1.04 ± 0.04 | -0.09 ± 0.06 | 1.92 ± 0.03 | 0.07 ± 0.02 | 0.33 ± 0.02 |
| bw = 0.25 | 1.08 ± 0.06 | -0.08 ± 0.07 | 1.96 ± 0.03 | 0.07 ± 0.02 | 0.34 ± 0.02 |
| bw = 0.3 | 1.11 ± 0.04 | -0.10 ± 0.06 | 1.96 ± 0.03 | 0.07 ± 0.03 | 0.34 ± 0.02 |
| bw = 0.35 | 1.03 ± 0.04 | -0.09 ± 0.07 | 1.93 ± 0.04 | 0.07 ± 0.03 | 0.34 ± 0.02 |
| bw = 0.4 | 1.17 ± 0.07 | -0.13 ± 0.05 | 1.97 ± 0.04 | 0.08 ± 0.07 | 0.35 ± 0.02 |
| bw = 0.45 | 1.17 ± 0.07 | -0.13 ± 0.06 | 1.97 ± 0.04 | 0.08 ± 0.07 | 0.35 ± 0.02 |
| FACE | 1.36 ± 0.16 | 0.34 ± 0.12 | 1.77 ± 0.03 | 0.06 ± 0.02 | 0.32 ± 0.01 |
| C-CHVAE | 6.39 ± 0.55 | 0.33 ± 0.15 | 1.18 ± 0.18 | 0.04 ± 0.01 | 0.46 ± 0.08 |

Table 21: CFX generation results, AdultIncome, MLP. Validity 80% for Wachter, 84-85% for all ECCCo methods, 89% for Greedy, 54% for C-CHVAE.

| Generator | Distance | Plausibility | Implausibility | Sensitivity $(10^{-1})$ | Stability |
|---|---|---|---|---|---|
| **RandomForest** | | | | | |
| ConfexNaive | | | | | |
| $\alpha = 0.01$ | 1.39 ± 0.19 | -0.16 ± 0.10 | 1.89 ± 0.07 | 0.14 ± 0.11 | 0.29 ± 0.05 |
| $\alpha = 0.05$ | 1.03 ± 0.08 | 0.04 ± 0.11 | 1.91 ± 0.05 | 0.12 ± 0.05 | 0.26 ± 0.04 |
| $\alpha = 0.1$ | 0.97 ± 0.09 | 0.00 ± 0.10 | 1.91 ± 0.04 | 0.12 ± 0.02 | 0.26 ± 0.03 |
| ConfexTree, $\alpha = 0.01$ | | | | | |
| bw = 0.05 | 1.56 ± 0.21 | 0.08 ± 0.15 | 1.80 ± 0.04 | 0.06 ± 0.03 | 0.18 ± 0.02 |
| bw = 0.1 | 1.41 ± 0.17 | -0.06 ± 0.14 | 1.79 ± 0.05 | 0.11 ± 0.03 | 0.24 ± 0.04 |
| bw = 0.15 | 1.29 ± 0.20 | -0.06 ± 0.13 | 1.82 ± 0.06 | 0.09 ± 0.03 | 0.24 ± 0.04 |
| bw = 0.2 | 1.34 ± 0.18 | -0.16 ± 0.10 | 1.89 ± 0.07 | 0.12 ± 0.03 | 0.25 ± 0.04 |
| bw = 0.25 | 1.28 ± 0.16 | -0.07 ± 0.08 | 1.91 ± 0.07 | 0.12 ± 0.05 | 0.26 ± 0.04 |
| bw = 0.3 | 1.24 ± 0.18 | -0.02 ± 0.04 | 1.93 ± 0.06 | 0.10 ± 0.05 | 0.24 ± 0.03 |
| bw = 0.35 | 1.28 ± 0.19 | -0.04 ± 0.10 | 1.93 ± 0.06 | 0.14 ± 0.10 | 0.25 ± 0.04 |
| bw = 0.4 | 1.44 ± 0.18 | -0.15 ± 0.09 | 1.90 ± 0.07 | 0.08 ± 0.03 | 0.29 ± 0.06 |
| bw = 0.45 | 1.45 ± 0.20 | -0.17 ± 0.10 | 1.90 ± 0.07 | 0.09 ± 0.03 | 0.29 ± 0.06 |
| ConfexTree, $\alpha = 0.05$ | | | | | |
| bw = 0.05 | 1.00 ± 0.06 | 0.13 ± 0.04 | 1.86 ± 0.05 | 0.12 ± 0.04 | 0.19 ± 0.02 |
| bw = 0.1 | 0.92 ± 0.08 | 0.08 ± 0.07 | 1.89 ± 0.05 | 0.12 ± 0.04 | 0.22 ± 0.02 |
| bw = 0.15 | 0.91 ± 0.05 | 0.08 ± 0.09 | 1.90 ± 0.04 | 0.15 ± 0.05 | 0.22 ± 0.02 |
| bw = 0.2 | 0.94 ± 0.04 | -0.02 ± 0.05 | 1.93 ± 0.04 | 0.14 ± 0.04 | 0.24 ± 0.03 |
| bw = 0.25 | 0.97 ± 0.03 | -0.04 ± 0.03 | 1.94 ± 0.03 | 0.09 ± 0.01 | 0.24 ± 0.03 |
| bw = 0.3 | 1.00 ± 0.04 | -0.02 ± 0.07 | 1.95 ± 0.03 | 0.11 ± 0.04 | 0.24 ± 0.02 |
| bw = 0.35 | 1.01 ± 0.06 | -0.05 ± 0.03 | 1.95 ± 0.03 | 0.11 ± 0.03 | 0.24 ± 0.02 |
| bw = 0.4 | 1.05 ± 0.08 | 0.03 ± 0.04 | 1.91 ± 0.04 | 0.13 ± 0.04 | 0.26 ± 0.04 |
| bw = 0.45 | 1.03 ± 0.07 | 0.04 ± 0.09 | 1.91 ± 0.04 | 0.15 ± 0.06 | 0.26 ± 0.04 |
| ConfexTree, $\alpha = 0.1$ | | | | | |
| bw = 0.05 | 0.82 ± 0.07 | 0.18 ± 0.05 | 1.88 ± 0.04 | 0.20 ± 0.06 | 0.20 ± 0.03 |
| bw = 0.1 | 0.86 ± 0.02 | 0.05 ± 0.05 | 1.90 ± 0.03 | 0.18 ± 0.10 | 0.22 ± 0.02 |
| bw = 0.15 | 0.90 ± 0.03 | 0.04 ± 0.07 | 1.92 ± 0.04 | 0.19 ± 0.13 | 0.24 ± 0.02 |
| bw = 0.2 | 0.89 ± 0.06 | 0.10 ± 0.10 | 1.88 ± 0.04 | 0.15 ± 0.08 | 0.25 ± 0.02 |
| bw = 0.25 | 0.87 ± 0.05 | 0.13 ± 0.07 | 1.87 ± 0.03 | 0.15 ± 0.05 | 0.25 ± 0.03 |
| bw = 0.3 | 0.90 ± 0.06 | 0.12 ± 0.13 | 1.89 ± 0.03 | 0.17 ± 0.05 | 0.25 ± 0.02 |
| bw = 0.35 | 0.91 ± 0.06 | 0.12 ± 0.08 | 1.88 ± 0.03 | 0.18 ± 0.10 | 0.25 ± 0.02 |
| bw = 0.4 | 0.95 ± 0.08 | 0.05 ± 0.07 | 1.91 ± 0.04 | 1.92 ± 3.55 | 0.25 ± 0.02 |
| bw = 0.45 | 0.97 ± 0.09 | 0.02 ± 0.08 | 1.92 ± 0.04 | 2.62 ± 4.87 | 0.25 ± 0.03 |
| FeatureTweak | 0.24 ± 0.10 | 0.30 ± 0.11 | 1.84 ± 0.04 | 0.06 ± 0.01 | 0.13 ± 0.01 |
| FOCUS | 0.58 ± 0.34 | 0.40 ± 0.06 | 1.84 ± 0.04 | 0.29 ± 0.14 | 0.17 ± 0.01 |
| FACE | 1.50 ± 0.07 | 0.39 ± 0.07 | 1.74 ± 0.03 | 0.06 ± 0.02 | 0.27 ± 0.01 |

Table 22: CFX generation results, AdultIncome, RandomForest. Methods with nan values had 100% failures. Validity 61% for FeatureTweak.

### D.6.4 CONFORMAL EVALUATION RESULTS

| Generator | Marginal CovGap | Binning CovGap | Class Cond CovGap | Simulated CovGap |
|---|---|---|---|---|
| **MLP** | | | | |
| ConfexNaive | | | | |
| $\alpha = 0.01$ | 0.99 ± 0.00 | -0.88 ± 0.08 | -0.01 ± 0.05 | -2.81 ± 0.88 |
| $\alpha = 0.05$ | 0.96 ± 0.00 | -2.64 ± 0.19 | 0.33 ± 0.09 | -6.63 ± 0.94 |
| $\alpha = 0.1$ | 0.90 ± 0.00 | -4.74 ± 0.78 | 0.21 ± 0.08 | -13.65 ± 2.19 |
| ConfexTree, $\alpha = 0.01$ | | | | |
| bw = 0.05 | 1.00 ± 0.00 | 0.10 ± 0.09 | 0.52 ± 0.06 | -14.95 ± 3.34 |
| bw = 0.1 | 0.99 ± 0.00 | -0.36 ± 0.16 | 0.28 ± 0.09 | -2.74 ± 0.97 |
| bw = 0.15 | 0.99 ± 0.00 | -0.32 ± 0.17 | 0.30 ± 0.09 | -2.72 ± 1.11 |
| bw = 0.2 | 0.99 ± 0.00 | -1.03 ± 0.10 | -0.08 ± 0.06 | -3.46 ± 1.05 |
| bw = 0.25 | 0.99 ± 0.00 | -0.97 ± 0.17 | -0.06 ± 0.09 | -3.98 ± 1.18 |
| bw = 0.3 | 0.99 ± 0.00 | -1.07 ± 0.15 | -0.13 ± 0.08 | -4.51 ± 1.16 |
| bw = 0.35 | 0.99 ± 0.00 | -0.93 ± 0.14 | -0.04 ± 0.08 | -4.18 ± 1.73 |
| bw = 0.4 | 0.99 ± 0.00 | -0.76 ± 0.20 | 0.07 ± 0.11 | -2.26 ± 1.53 |
| bw = 0.45 | 0.99 ± 0.00 | -0.76 ± 0.20 | 0.07 ± 0.11 | -2.26 ± 1.53 |
| ConfexTree, $\alpha = 0.05$ | | | | |
| bw = 0.05 | 0.98 ± 0.00 | 0.61 ± 0.18 | 2.48 ± 0.05 | -18.52 ± 5.56 |
| bw = 0.1 | 0.96 ± 0.00 | -1.50 ± 0.16 | 1.11 ± 0.12 | -11.54 ± 2.82 |
| bw = 0.15 | 0.96 ± 0.00 | -1.92 ± 0.14 | 0.86 ± 0.14 | -11.57 ± 2.21 |
| bw = 0.2 | 0.95 ± 0.00 | -2.55 ± 0.37 | 0.47 ± 0.14 | -9.68 ± 1.30 |
| bw = 0.25 | 0.95 ± 0.00 | -2.61 ± 0.31 | 0.35 ± 0.10 | -9.54 ± 1.12 |
| bw = 0.3 | 0.95 ± 0.00 | -2.71 ± 0.30 | 0.28 ± 0.08 | -9.89 ± 1.27 |
| bw = 0.35 | 0.95 ± 0.00 | -2.71 ± 0.37 | 0.30 ± 0.14 | -9.48 ± 1.13 |
| bw = 0.4 | 0.95 ± 0.01 | -2.88 ± 0.27 | 0.14 ± 0.19 | -6.64 ± 0.66 |
| bw = 0.45 | 0.95 ± 0.01 | -2.88 ± 0.27 | 0.14 ± 0.19 | -6.76 ± 0.60 |
| ConfexTree, $\alpha = 0.1$ | | | | |
| bw = 0.05 | 0.94 ± 0.00 | 1.33 ± 0.38 | 4.61 ± 0.06 | -16.15 ± 5.52 |
| bw = 0.1 | 0.93 ± 0.00 | -0.44 ± 0.47 | 3.46 ± 0.05 | -12.99 ± 4.44 |
| bw = 0.15 | 0.93 ± 0.00 | -0.91 ± 0.53 | 3.16 ± 0.11 | -12.76 ± 3.92 |
| bw = 0.2 | 0.92 ± 0.00 | -2.51 ± 0.51 | 1.93 ± 0.17 | -13.07 ± 1.83 |
| bw = 0.25 | 0.92 ± 0.00 | -2.57 ± 0.53 | 1.78 ± 0.13 | -13.01 ± 1.16 |
| bw = 0.3 | 0.92 ± 0.00 | -2.68 ± 0.55 | 1.72 ± 0.12 | -12.83 ± 1.34 |
| bw = 0.35 | 0.92 ± 0.00 | -2.81 ± 0.54 | 1.64 ± 0.12 | -12.79 ± 1.44 |
| bw = 0.4 | 0.90 ± 0.00 | -5.01 ± 0.90 | 0.13 ± 0.17 | -15.19 ± 2.17 |
| bw = 0.45 | 0.90 ± 0.00 | -5.01 ± 0.90 | 0.13 ± 0.17 | -15.19 ± 2.20 |

Table 23: Conformal evaluation results, AdultIncome, MLP

| Generator | Marginal CovGap | Binning CovGap | Class Cond CovGap | Simulated CovGap |
|---|---|---|---|---|
| **RandomForest** | | | | |
| ConfexNaive | | | | |
| $\alpha = 0.01$ | 0.99 ± 0.00 | -0.90 ± 0.22 | 0.01 ± 0.09 | -5.70 ± 5.11 |
| $\alpha = 0.05$ | 0.96 ± 0.00 | -2.81 ± 0.08 | 0.35 ± 0.12 | -5.49 ± 2.52 |
| $\alpha = 0.1$ | 0.91 ± 0.00 | -5.78 ± 0.58 | 0.13 ± 0.14 | -10.19 ± 0.48 |
| ConfexTree, $\alpha = 0.01$ | | | | |
| bw = 0.05 | 1.00 ± 0.00 | 0.20 ± 0.02 | 0.56 ± 0.01 | -5.11 ± 2.96 |
| bw = 0.1 | 0.99 ± 0.00 | -0.43 ± 0.08 | 0.23 ± 0.03 | -2.95 ± 0.14 |
| bw = 0.15 | 0.99 ± 0.00 | -0.45 ± 0.12 | 0.23 ± 0.05 | -2.67 ± 0.17 |
| bw = 0.2 | 0.99 ± 0.00 | -0.80 ± 0.12 | 0.05 ± 0.04 | -3.32 ± 0.27 |
| bw = 0.25 | 0.99 ± 0.00 | -0.78 ± 0.13 | 0.04 ± 0.03 | -3.89 ± 0.55 |
| bw = 0.3 | 0.99 ± 0.00 | -0.87 ± 0.19 | -0.00 ± 0.07 | -4.68 ± 0.38 |
| bw = 0.35 | 0.99 ± 0.00 | -0.79 ± 0.20 | 0.04 ± 0.08 | -3.76 ± 0.17 |
| bw = 0.4 | 0.99 ± 0.00 | -0.91 ± 0.24 | -0.00 ± 0.10 | -7.54 ± 4.96 |
| bw = 0.45 | 0.99 ± 0.00 | -0.91 ± 0.24 | -0.00 ± 0.10 | -7.54 ± 4.96 |
| ConfexTree, $\alpha = 0.05$ | | | | |
| bw = 0.05 | 0.97 ± 0.01 | 0.03 ± 0.06 | 1.84 ± 0.08 | -6.43 ± 1.84 |
| bw = 0.1 | 0.96 ± 0.00 | -1.60 ± 0.23 | 0.88 ± 0.11 | -5.54 ± 1.04 |
| bw = 0.15 | 0.95 ± 0.00 | -2.03 ± 0.04 | 0.58 ± 0.05 | -5.39 ± 0.67 |
| bw = 0.2 | 0.95 ± 0.00 | -2.59 ± 0.01 | 0.44 ± 0.02 | -6.10 ± 0.05 |
| bw = 0.25 | 0.96 ± 0.00 | -2.66 ± 0.19 | 0.38 ± 0.10 | -6.74 ± 0.05 |
| bw = 0.3 | 0.96 ± 0.00 | -2.59 ± 0.12 | 0.43 ± 0.05 | -6.17 ± 0.05 |
| bw = 0.35 | 0.96 ± 0.00 | -2.52 ± 0.08 | 0.47 ± 0.02 | -6.79 ± 0.05 |
| bw = 0.4 | 0.95 ± 0.00 | -2.82 ± 0.10 | 0.31 ± 0.18 | -6.25 ± 0.14 |
| bw = 0.45 | 0.95 ± 0.00 | -2.82 ± 0.10 | 0.31 ± 0.18 | -6.24 ± 0.12 |
| ConfexTree, $\alpha = 0.1$ | | | | |
| bw = 0.05 | 0.93 ± 0.00 | -0.11 ± 0.80 | 2.91 ± 0.35 | -4.98 ± 0.44 |
| bw = 0.1 | 0.91 ± 0.00 | -2.48 ± 0.49 | 1.38 ± 0.18 | -4.62 ± 1.45 |
| bw = 0.15 | 0.91 ± 0.01 | -3.14 ± 0.83 | 1.00 ± 0.34 | -4.33 ± 1.62 |
| bw = 0.2 | 0.91 ± 0.00 | -4.33 ± 0.10 | 0.66 ± 0.05 | -4.80 ± 0.47 |
| bw = 0.25 | 0.91 ± 0.00 | -4.50 ± 0.07 | 0.54 ± 0.16 | -5.34 ± 0.81 |
| bw = 0.3 | 0.91 ± 0.00 | -4.67 ± 0.15 | 0.38 ± 0.20 | -5.73 ± 1.22 |
| bw = 0.35 | 0.91 ± 0.00 | -4.88 ± 0.36 | 0.25 ± 0.33 | -5.56 ± 1.45 |
| bw = 0.4 | 0.91 ± 0.00 | -6.14 ± 0.35 | 0.04 ± 0.06 | -10.77 ± 3.43 |
| bw = 0.45 | 0.91 ± 0.00 | -6.14 ± 0.34 | 0.03 ± 0.05 | -10.87 ± 3.20 |

Table 24: Conformal evaluation results, AdultIncome, RandomForest

