# OpenReview forum: "CONFEX: Uncertainty-Aware Counterfactual Explanations with Conformal Guarantees"
_ICLR.cc/2026/Conference — Submitted to ICLR 2026_

### Official Review · Reviewer_wspM · 2025-10-25

**Soundness:** 2
**Presentation:** 2
**Contribution:** 2
**Rating:** 4
**Confidence:** 5

**Summary:**

This paper introduces CONFEX, a novel method for generating counterfactual explanations (CFXs) that incorporates uncertainty quantification through Conformal Prediction (CP). The core motivation is that existing CFX methods often fail to account for predictive uncertainty, potentially leading to unreliable explanations that suggest changes in regions where the model's predictions are uncertain or unsupported by data. The authors identify a critical flaw in naively applying CP to CFX generation: the generated counterfactuals may not be exchangeable with the calibration data, violating CP's fundamental assumptions. To address this, they propose using localized CP (LCP) to enforce approximate conditional (test-conditional) coverage guarantees rather than just marginal guarantees. The paper presents three variants: CONFEX-Naive, CONFEX-LCP, and CONFEX-Tree. All methods use Mixed-Integer Linear Programming (MILP) to guarantee optimal solutions and enforce that CFXs yield singleton prediction regions for the target class, indicating high confidence predictions.

**Strengths:**

The paper demonstrates technical rigor in identifying and addressing a fundamental problem with applying conformal prediction to counterfactual explanation generation. The recognition that CFX search violates the exchangeability assumption which a cornerstone of CP's validity which represents genuine theoretical insight. The authors don't simply note this problem but provide a principled solution through localized CP that enforces approximate conditional guarantees. The mathematical formulation is clear and well-justified, with formal proofs of group-conditional coverage guarantees for CONFEX-Tree. This theoretical grounding distinguishes the work from heuristic approaches to uncertainty-aware CFX generation.The MILP encoding is another strength, ensuring both optimality of solutions and hard satisfaction of CP constraints. CONFEX provides provable guarantees on both the validity and minimality of explanations which is unlike gradient-based methods that may fail to find valid counterfactuals or converge to suboptimal solutions. The formulation integrates uncertainty constraints into the optimization framework and thus making uncertainty in the CFX generation process rather than an afterthought.

**Weaknesses:**

The novelty of this work is fundamentally incremental rather than groundbreaking. The paper essentially combines two existing techniques i.e., localized conformal prediction (Guan, 2023) and MILP-based counterfactual generation (Kanamori et al., 2020). in a principled way. While the combination is non-trivial and the application domain is new, neither the core CP methodology nor the optimization framework represents novel technical contributions. The localized CP procedure is directly borrowed from prior work, and the MILP encoding follows established patterns for constraint satisfaction problems.

The tree-based quantile computation (CONFEX-Tree) is the most original technical contribution, but even this builds on well-known spatial data structures (KD-trees). The insight to use offline partitioning is clever engineering rather than algorithmic innovation. The connection to existing group-conditional CP methods further diminishes the novelty, as the approach can be viewed as a specific instantiation of known conditional CP frameworks with spatial grouping.

A critical limitation affecting the contribution's impact is the restrictive scope of applicable models. CONFEX is limited to models with MILP-encodable architectures such as linear models, MLPs with ReLU activations, and decision trees. This restriction severely limits the practical applicability of CONFEX to contemporary machine learning systems. The experiments are conducted only on tabular datasets with relatively simple models (50-unit MLPs, small random forests), which may not reflect real-world deployment scenarios where deeper, more complex models are standard.

The paper doesn't adequately address why formal coverage guarantees are necessary for CFX generation in practice. While theoretically appealing, the practical benefit of having a 90% coverage guarantee versus a heuristic that produces empirically plausible counterfactuals isn't convincingly demonstrated.

**Questions:**

1. What is the formal approximation quality of CONFEX-Tree's conditional coverage guarantees, and how does it degrade with bandwidth selection?

2. How does computational cost scale with model complexity, and what is the practical runtime-quality tradeoff compared to gradient-based methods?

3. Can the method be extended to models with non-MILP-encodable components, and what theoretical guarantees would remain?

4. How should bandwidth be selected systematically, and what is the sensitivity of coverage guarantees and counterfactual quality to this choice?

**Details Of Ethics Concerns:**

No ethics concerns have been identified in the work.

---

> ### Author Response · Authors · 2025-11-20
> **Official Comment by Authors (part 1)**
>
> ## Need for guarantees
> We think that formal uncertainty guarantees are very important in the context of CFXs for the following reasons.
> 1) Counterfactuals are often used for recourse recommendations. Without uncertainty guarantees, the CFX method may suggest CFXs where the model is uncertain, misleading the user into making changes that do not actually alter the outcome. Our method instead guarantees that the decision flip happens with (arbitrary) high probability.
> 2) Our local (i.e., approx. conditional) guarantees imply that our CFXs are valid with high probability for any individual, even for out-of-distribution ones. This confers our method’s robustness to distribution shifts. On the other hand, a heuristic notion of uncertainty (or plausibility etc.) wouldn't provide reliable explanations for every individual, even if this heuristic performed well empirically on a test set, and it might result in wildly implausible explanations when distributions shift.
>
> We thank the reviewer for highlighting the need for further motivating the formal guarantees. We will incorporate these points in the paper. We would like to note that, based on reviewer bi8T's suggestion, the paper revision includes a formal problem statement that should better clarify the nature of our guarantees (see Problem 1).
>
> ## Novelty of work
>
> We believe that our work provides a meaningful contribution to the field of counterfactual explanations.
>
> Prior work such as [1] and [2] have attempted to incorporate conformal prediction in CFX generation, however these methods fail to retain formal guarantees. Effectively, they use conformal prediction as a heuristic, which is contrary to the purpose of conformal prediction.
> Both papers [1] and [2] conclude that they both require further analysis into conformal prediction in order to retain formal guarantees.
>
> - "the scope of this work has prevented us from investigating the role of conformal prediction in this context more thoroughly" [1]
> - "Another line of future research relates to the theoretical underpinning of the CPICF generation. Standard conformal prediction requires the input data to be exchangeable. Thus, once the counterfactual is added, the conformal prediction guarantees no longer hold. [...] While the CPICFs and the calibration data are not exchangeable, it may be possible to apply ideas from Hore and Barber (2025) to randomise the intervals in order to obtain theoretical guarantees; investigating this will be part of future work." [2]
>
> Our contribution lies in solving this open problem, by providing a method for uncertainty-aware CFXs which retains conformal guarantees. As the reviewer points out, our solution builds on a non-trivial combination of LCP and MILP, and in particular, on an efficient tree-based encoding of local quantiles (CONFEX-Tree). We respectfully disagree with the statement that the group-conditional validity of CONFEX-Tree diminishes our contribution, as CONFEX-Tree provides both local (approx conditional) and group-conditional guarantees (see Eq. 16 and 17 in the paper).
>
>
> ## Model limitations
> In practical algorithmic recourse settings, tabular data is most common (e.g., loan and hiring applications). For tabular datasets, methods like random forest or gradient boosted trees are popular and quite competitive and CONFEX is compatible with both (e.g. see https://arxiv.org/abs/2305.02997 and https://arxiv.org/abs/2402.03970v2)
> Note that we have an implementation of boosted trees for CONFEX, but we do not include it in the results because implementations are unavailable in the generators we benchmark against.
>
> On model size: our focus is on providing strong guarantees on the counterfactuals, which requires using exhaustive methods like constraints solving. These exhaustive methods do not scale to large or very large neural networks, where it is difficult to obtain any guarantees. In high-risk settings like recourse, we argue it is more desirable to have formally proven / certified outcomes than sacrifice the guarantees for (often unnecessarily) large neural nets.
>
> ## Can the method be extended to models with non-MILP-encodable components, and what theoretical guarantees would remain?
>
> Due to the black-box nature of conformal prediction (i.e. it works with any predictive model), our method could be extended to non-MILP-encodable components. This would be possible by utilising gradient-based search methods or black-box stochastic optimisation methods like evolutionary algorithms instead of MILP. However, the method would no longer be complete, i.e. it may return a sub-optimal CFX or no CFX at all even if one exists that satisfies the conformal prediction constraints.
>
> That said, the classifier may include non-MILP-encodable components as long as the score function $s(x,y)$ remains so. In the paper, indeed we consider NN models with a final softmax component, but we use a log-likelihood-ratio score function that is linear as it only uses the network’s logits.

---

> ### Author Response · Authors · 2025-11-20
> **Official Comment by Authors (part 2)**
>
> ## Formal approximation quality of CONFEX-Tree’s conditional coverage guarantees.
> In the paper, we already include an analysis of how the conditional coverage guarantees of CONFEX change w.r.t. the kernel bandwidth. To do so, we measure the so-called “coverage gap”, i.e., the difference between the nominal coverage $1-\alpha$ and the empirical coverage (see also Appendix C, eq. 18, for a definition of this metric). Results for this analysis can be found in the plots c) of figure 2 (main paper), and Appendix C.
>
> ## Scalability analysis
> We thank the reviewer for the suggestion to include a scalability analysis of our method, this suggestion was also raised by Reviewer 55xF. In response, we are currently working on producing this analysis and will add it to the appendix as another section.
>
> ## Choosing bandwidth systematically
>
> As explained in the paper, one could tune the bandwidth by running the method on a held-out dataset. In our experiments, we observe that plausibility, sensitivity, and coverage degrade as the bandwidth increases (i.e., as the conformal prediction constraints become less local/conditional). We also see that choosing bandwidths where the target coverage is met generally leads to good plausibility and sensitivity.
>
> ## References
> [1] Altmeyer, Patrick, et al. "Faithful model explanations through energy-constrained conformal counterfactuals." Proceedings of the AAAI conference on artificial intelligence. Vol. 38. No. 10. 2024.
>
> [2] Adams, James M., et al. "Individualised Counterfactual Examples Using Conformal Prediction Intervals." arXiv preprint arXiv:2505.22326 (2025).

---

> ### Author Response · Authors · 2025-12-03
> **Summary Comment by Authors**
>
> We sincerely thank reviewer wspM for their review and note that the reviewer did not respond to our rebuttal during the time in which this was possible. Nevertheless, we have used their suggestions to improve the paper and describe the changes made below:
> - We further clarified the importance and novelty of our contribution in the Introduction (Lines 90, 170).
> - We further motivate the need for formal guarantees in CFXs (Lines 170, 404).
> - We add further information regarding the scope of applicable models (Line 530).
> - We refined wording to clarify the procedure to systematically select bandwidth (Line 537).
> - Appendix C was included, which contains the scalability results requested by this reviewer and Reviewer 55xF. Appendix C considers the effect that the dimensionality of the data, size of the calibration set, and model complexity (evaluated for MLPs, random forests and gradient boosted trees) has on the time taken to generate CFXs. Additionally we consider how the size of the calibration set affects distance and plausibility.

---

### Official Review · Reviewer_55xF · 2025-10-29

**Soundness:** 1
**Presentation:** 2
**Contribution:** 2
**Rating:** 2
**Confidence:** 3

**Summary:**

The paper is concerned with finding counterfactual explanations that are from regions of high certainty of the model. Specifically, the paper argues for including the information about data support to avoid counterfactuals in regions without data. They utilize Mixed-Integer Programming (MIP) to find counterfactuals with high certainty according to the Localised Conformal Prediction (LCP). To improve scalability, they propose approximating the true LCP with a tree model used to split the input space into a more manageable number of strata, rather than expressing the full calibration set in the formulation.

**Strengths:**

The paper tackles an interesting question and reads okay. I believe the paper addresses an important problem with using CP for counterfactual generation. The authors also cleverly simplify the problem to enable better scalability.

**Weaknesses:**

There are multiple issues with the paper. In order of perceived importance:

- The paper seems to wrongly claim that the MIP formulation is linear. Algorithm 2 contains the multiplication of two variables ($in_i$ and $w_i$), specifically on lines 8 and 9. The paper should specify the exact and complete MIP formulation, as is otherwise common in MIP literature, at least in appendix. Especially, the formulation of CONFEX-Tree is essential. Algorithm 1 specifies only a general idea and no further explanation or the actual mathematical formulation is provided. This is promised on line 90, sa the first contribution.
- The comparison to the state of the art is dubious. On line 373, four datasets are mentioned, but results on two of them are postponed to the appendix, where they are presented without commentary. It is unclear why two out of four datasets were not included in the main body, though the poor plausibility results on the Adult dataset suggest potential selection bias. Furthermore, the comparison does not include any other method that considers plausibility, despite there being plenty of them, including the cited FACE (Poyiadzi et al., 2020) or some MILP-based, e.g., DACE using the LOF (Kanamori et al., 2020), or more recent LiCE (https://openreview.net/pdf?id=rGyi8NNqB0 - ICLR 2025). Alternatively, C-CHVAE (Pawelczyk et al., 2020) is implemented inside the Carla library that was used for experiments. A similar thing could be said about robustness, where one could look at e.g., PROPLACE (https://proceedings.mlr.press/v222/jiang24a/jiang24a.pdf). Despite this lack of a plausible CFX method, the authors claim to provide an extensive evaluation that demonstrates CONFEX provides more plausible and stable explanations than competing generators. Note that results on stability are not discussed, just presented in the appendix, and their interpretation is inconclusive.
- The scalability and performance of the Tree variant in relation to the LCP variant should be evaluated. The evaluation of runtime with respect to calibration set size could be useful in deciding at which point the (assumed) decrease in performance is worth the improvement in scalability. Three minutes per instance for 100 instances seems to be computable.
- The results in Table 1 are non-trivial to compare, since the mean is reported over different sets of factuals (only those where each method was valid).
- The definition of sensitivity and the particular distance used in Table 1 could be stated in the main body, and the Coverage Gap should also be defined more clearly.

The issues I found led me to the recommendation to reject the contribution. I am certainly willing to be swayed if my concerns are alleviated.

Minor comments:
- Figure 1 should be better described.
- On line 178, a confusion might arise, as the f can be non-linear, while still being MILP-representable. I would recommend rewording this to avoid confusion.
- line 226-227, a confusingly worded section
- line 359 eq. (14) - A colon should follow the g, possibly?
- Distance result of the FOCUS method on the Credit datasets in Table 1 should probably be bold.

**Questions:**

Most pressing questions:
- Can the MIP formulation of LCP be linearized?
- Why were the results of stability not discussed and included in the main body? They seem to be the most relevant measure of certainty of the counterfactual.
- Why were results on Adult and GMSC omitted from the main body?
- Are the results statistically significant?
- How do the methods compare when taking the intersections of factuals for which all methods were successful?

Minor questions:
- Is the ratio (6) a standard score function?
- Why was CLUE not used in evaluations?
- The +- information and the bands in Figure 2 are the standard deviation? Over how many runs under what configuration?
- What does the (10^-1) by the Sensitivity mean?
- How exactly is the coverage gap defined?

---

> ### Author Response · Authors · 2025-11-20
> **Official Comment by Authors (part 1)**
>
> ## Linearity of the MIP formulation
> The CONFEX methods and the MIP formulation in Algorithm 2 is indeed linear.
>
> Whilst 8 and 9 contain the multiplication of two variables, the multiplications comprise of at least one binary variable (in). In such cases, it is possible to linearise the operation, and the solver does this in the pre-solve phase. See [1].
>
> In particular, when using a box kernel, both $in$ and $w$ are binary, and the multiplication $z$ of two binary variables $x$ and $y$ can be determined with these constraints: $ z \leq x; z \leq y; z \geq y + z - 1  $.
>
> We understand how the confusion can arise and thank the reviewer for pointing this out. In response, we have added a short clarification on lines 899-905.
>
> ## Exact and complete MIP formulation
> We thank the reviewer for this suggestion and agree that adding the complete MIP formulation would add to the paper. This has now been included as Appendix B. Appendix B now also includes a description of the MIP formulation of our CONFEX-Tree algorithm (Algorithm 3).
>
> ## Comparison to state of the art
> In our comparison with the state of the art, we chose to compare only against uncertainty-aware CFX generators (like ours). We are aware that CFX methods exist that directly optimise for the desired metric (e.g., plausibility and robustness). Our aim is to show that, thanks to its rigorous conformal prediction constraints, our CONFEX method yields better plausibility and robustness than existing uncertainty-aware methods, without directly optimising for these metrics. We can emphasise this point further in the results/introduction.
>
> Nevertheless, following the reviewer’s suggestion, we were able to implement and evaluate FACE and C-CHVAE. The updated results in Table 1 confirm that our method (CTree in the table) outperforms FACE and C-CHVAE in both plausibility and robustness/sensitivity, except for the CaliforniaHousing dataset where CONFEX-Tree comes second to FACE (outperforming C-CHVAE) in terms of plausibility but has the best sensitivity. But again, CONFEX doesn’t directly optimize for plausibility unlike FACE and C-CHVAE.
>
> The reviewer gave additional suggestions for other generators:
> - We do not include CLUE: CLUE is an explanation method for Bayesian Neural Networks (or more generally, models which provide a differentiable estimate of uncertainty H, see [2]). This limits the class of models that can be used for CFX generation. Also, CLUE is not compatible with the (deterministic) neural networks or random forests used in our experiments.
> - We do not include DACE: their formulation relies on a finite action set A of counterfactual perturbations. (See section Action and Ordered Action in [3]) Additionally, no code is provided by the authors.
> - PROPLACE is a method which targets a different notion of robustness than what we measure: PROPLACE targets robustness to model parameter changes whereas we measure robustness to input perturbation (sensitivity).
>
> A final minor clarification: we do not use the CARLA library, as it is incompatible with modern Python versions.
>
> ## Results and metrics postponed to the appendix
> We postponed Adult, GMSC, the definition of metrics and the Sensitivity metrics results to the appendix purely due to space constraints.
>
> In response to omitting the datasets, we have rewritten the appendices: each dataset is now provided with the same plots we find in the main paper, is accompanied by a short commentary and tables were reformatted to be more readable.
>
> Regarding omitting the stability metric: our preliminary tests on this metrics suggested that it could have been a promising indicator to include in the main evaluation. However, as pointed out by the reviewer, the results are inconclusive: with some datasets (e.g. GMSC) we find little variation in the value of this metric and with others (e.g. GC) we find that CONFEX methods perform better than competitors on stability. Since we did not want to discard the results, and were short on space in the main paper, we left them in the appendix.
>
> We respectfully disagree with the idea that stability seems “to be the most relevant measure of certainty of the counterfactual.”: it is not an established metric in the CFX literature; it was introduced as an optimisation objective rather than a metric in [4], and we found it to be quite sensitive to the tuning of the width parameter.
>
> ## Scalability analysis of CONFEX-Tree vs CONFEX-LCP
> We thank the reviewers for this suggestion and agree that it would be useful to include this information. We are currently working to get these results and will add them as a section in in the appendix. We would like to stress, however, that CONFEX-Tree should not be seen as an approximation of CONFEX-LCP, as CONFEX-Tree also provides local coverage guarantees (but under a different kernel definition, see Eq. 17). Moreover, it provides (unlike CONFEX-LCP) group-conditional guarantees (see Eq. 16).

---

> ### Author Response · Authors · 2025-11-20
> **Official Comment by Authors (part 2)**
>
> ## Comparison across different sets of factuals
> We thank the reviewers for pointing this out. We agree that, by excluding factuals with non-valid CFXs, in some cases we may compare different sets of factuals.
> We stress this is not the case for the CaliforniaHousing dataset, where all methods had 100% validity. In the GermanCredit dataset, instead, some of the incomplete methods can’t always produce valid CFXs.
>
> However, we do not believe that taking the intersection of factuals for which all methods are successful would be the correct approach, as it would unfairly favour methods with low validity. Specifically, it would be a form of selection bias because we would retain only the points were these incomplete methods perform well. Also, it would result in discarding a lot of useful data (some methods have validity as low as 50%, and their overall intersection would be even smaller).
>
> We resolve this issue by more clearly stating the validity of each method in the caption of Table 1.
>
> ## Are the results statistically significant?
> To increase our confidence in our results, we have increased the number of runs included in our evaluation to 5 and have updated Table 1 and Figure 2 to reflect this.
>
> To test the significance of our results in Table 1, we perform t-tests to check whether the mean of each generator’s metric is significantly different from the mean of the best performing generator’s metric.
>
> The results are shown in Table 2, which can be found in Appendix A. We find that across most metrics, the p-values are small (< 0.05), indicating that the observed gaps exceed run-to-run variability. The main exception is for the Sensitivity metric on CaliforniaHousing, where a few methods are statistically indistinguishable from the best method.
>
> We will update the results discussion in the paper with this information and the results on the newly added generators FACE and C-CHVAE.
>
> ## The +- information and the bands in Figure 2 are the standard deviation? Over how many runs under what configuration?
> Yes, standard deviation over 5 runs. Each run, uses a different random seed to train the model, and uses a different random sample of 200 points from the test set, as explained in Appendix C.
>
> ## Is the ratio (6) a standard score function?
>
> In conformal prediction, there is no “standard” score function for classification, but the most popular score is perhaps $S(x,y) = 1-p(x)_y$. However, for neural networks, this score can’t be expressed in MILP as it uses softmax probabilities. Our log-likelihood ratio score is instead linear as it reduces to a difference between logits. Moreover, our score is more expressive than the above because, as explained in the paper, it increases as the discrepancy between the probability of the predicted class and that of the true class $y$ grows bigger. That is, our score correctly penalises predictions where the model is wrongly over-confident. That said, the conformal prediction guarantees are independent of the specific choice of score function.
>
> ## What does the (10^-1) by the Sensitivity mean?
> The values in that column were multiplied by 0.1 to make them easier to read.
>
> ## Definition of coverage gap
> The coverage gap is computed as `100 * (empirical coverage – target coverage)`
>
> ## Minor weaknesses
> In the newly uploaded revision of the paper, we fixed minor wording and formatting changes suggested by the reviewer and updated the captions of some figures to increase clarity.
>
> ## References
> [1] Klotz, E. (2021) Specialized strategies for products of binary variables, Gurobi. Available at: https://cdn.gurobi.com/wp-content/uploads/Models-with-Products-of-Binary-Variables.pdf (Accessed: 18 November 2025).
>
> [2] Antorán, Javier, et al. "Getting a clue: A method for explaining uncertainty estimates." arXiv preprint arXiv:2006.06848(2020).
>
> [3] Kanamori, Kentaro, et al. "Ordered counterfactual explanation by mixed-integer linear optimization." Proceedings of the AAAI Conference on Artificial Intelligence. Vol. 35. No. 13. 2021.
>
> [4] Dutta, Sanghamitra, et al. "Robust counterfactual explanations for tree-based ensembles." International conference on machine learning. PMLR, 2022.

---

> > ### Comment · Reviewer_55xF · 2025-11-21
> >
> > I thank the authors for the thorough rebuttal and appreciate the inclusion of more details and the clarifications, especially the linearization of the quadratic terms in the formulation. I suspected as much.
> >
> > ### Plausibility
> > I understand that plausibility is not the primary goal of CONFEX, but in that case, the wording of some claims should be made more nuanced. When comparing to "competing generators", which the CONFEX outperforms by "providing more plausible and stable explanations" (L98), the fact that the authors consider only the uncertainty-aware CFX generators should be made clearer. Notably, the abstract suggests a comparison to state-of-the-art methods in terms of plausibility and stability, which would also include other models.
> >
> > Nevertheless, it could be argued, that the optimization task here (5) is closely related to the task of finding plausible counterfactuals. The difference is that plausibility constrains the joint probability p(y', x'), while here, the authors constrain p(y', x')/p(x').
> > And indeed, the C-CHVAE and FACE (which consider plausibility) both show competitive results in terms of sensitivity on CaliforniaHousing, suggesting a correlation between plausibility and sensitivity. This comparable stability performance of C-CHVAE and FACE also seems to be the case on Adult and GMSC datasets in the Appendices. The CONFEX sensitivity results are/seem to be not statistically significant, when compared to these plausibility methods.
> >
> > As a side note, it is not true that DACE does not have an implementation (https://github.com/kelicht/dace), and LiCE was outright omitted from the rebuttal, while precisely those MILP-based plausibility-oriented CFX generators would be the best candidates for comparison in terms of plausibility.
> >
> > ### How does CONFEX compare?
> > If stability is not the best metric for (un)certainty, despite it measuring the probability of counterfactual remaining valid when slightly perturbed, what metric do you use to compare to other approaches in terms of uncertainty? The CONFEX shows better performance only on sensitivity, and even that seems to be statistically significant only on one dataset, as mentioned before. I did not find a comparison of the coverage gap to other methods, so it is difficult to evaluate if the results show improvement in that dimension.
> >
> > I am willing to increase my score if the claims made in the paper match the experimental results, since I find the method interesting and possibly useful.

---

> ### Author Response · Authors · 2025-11-26
> **Official Comment by Authors**
>
> We thank the reviewer for their response to our rebuttal. We have used the feedback to further improve the paper, and have uploaded a new revision: https://openreview.net/pdf?id=FE3FVddJaT
>
> ## Wording of claims
> Following the reviewer’s suggestion, we have refined the claims made in the paper to specify that we outperform competing uncertainty-aware generators in terms of plausibility and robustness.
>
> ## Other plausibility-targeting generators
>
> We thank the reviewer for the correction regarding the availability of DACE’s code. However, we believe DACE is not the most informative baseline for our setting, since its use of a discrete action set requires discretising all real-valued variables, which changes the problem formulation and can make comparisons less direct and less representative of performance in the continuous-feature regimes we target.
>
> Currently, our experiments include a variety of plausibility-targeting generators: FACE, which is black-box, ECCCo, which targets plausibility as well as uncertainty, and C-CHVAE, which uses an auxiliary generative model to optimise for plausibility. Regarding LiCE, we can add it to the final version of the paper; however, during the current rebuttal period - due to time and resource constraints - we have focused our efforts on bringing other results, such as the certainty results, described below.
>
> ## Difference between optimising for plausibility and our method
> As the reviewer correctly points out, there are analogies between our method and methods optimising for plausibility.
>
> A constraint for plausibility would be that $P_{X,Y}(X=x' | Y=y^+) \geq threshold$, i.e., that the probability of observing a counterfactual point $x’$ is higher than a given threshold, given that we restrict to the target class.
>
> Our constraint is of the form $P_{X,Y}(Y=y^+ | X=x) \geq threshold$, i.e., that the counterfactual point $x’$ yields the target class with high probability, indicating certainty.
>
> However, these properties are distinct, and measures for plausibility (or sensitivity, stability, etc.) do not directly measure certainty in the counterfactual.
>
> We had chosen these metrics to compare against since we wanted to evaluate the performance on CONFEX on metrics that it does not directly optimise for. However, we agree with the reviewer’s suggestion that including a metric of uncertainty would be a good addition to clearly show the effect of our method and its design.
>
> ## A metric for certainty
>
> To quantify the certainty of the counterfactuals in a principled way, we use the conformal p-values from the CONFEX-Tree procedure. In a conformal prediction procedure, the conformal p-value of a point $(x,y)$ is the proportion of the calibration points with score above $s(x,y)$. It is used to determine which labels are included in the prediction set: labels with a p-value over $\alpha$ are included and the rest excluded. This is equivalent to checking if $s(x,y)$ is above the 1-alpha quantile of the calibration score (as explained in the background section of our paper).
>
> A low (high) p-value for $(x,y)$ provides strong evidence that $y$ is not (is) the true label for $x$. Hence, for our (un)certainty metric, we compute the average difference between the conformal p-value for the target class and the max of conformal p-values for all other classes:
>
> $$
> \mathrm{cert}(x') = p_{y^+}(x') - \max_{y \neq y^+} p_y(x')
> $$
>
> If the p-value of the target class is high and the max p-value of the other classes are low—indicating a certain prediction with strong evidence in favour of the target class and against others—then our metric will be high. If the prediction is uncertain, then the p-values of all classes will be similar, leading to a lower value of our metric.
>
> Note that we use our CONFEX-Tree procedure to compute p-values for this metric because of its (quasi-)conditional/local guarantees. We don't use vanilla/marginal CP, because its resulting p-values would be affected by calibration points well away (not local) to the counterfactual point of interest, i.e., marginal p-values suffer from the same exchangeability issues as we detail in Section 3.1.
>
> Results for certainty are provided in Appendix A (pages 14 and 15), and we demonstrate that our method, over all datasets, consistently provides more certain counterfactuals than all competing generators.

---

> > ### Comment · Reviewer_55xF · 2025-11-26
> >
> > I thank the authors for the edits. Since my concerns about the claims are now sufficiently addressed, I am raising my score.
> >
> > Nonetheless, I remain unconvinced about the metrics. If I understand correctly, the proposed certainty metric is optimized by CONFEX, making its dominance on it less informative. Is the difference in p-values (cert) a difference in proportions of calibration points with high enough score from one class compared to the other class (or max of all others) in a leaf that the x' belongs to? Why was LCP with at least a different (e.g., L1) kernel not used here for evaluation?

---

> ### Author Response · Authors · 2025-11-27
> **Official Comment by Authors**
>
> We thank the reviewer for their feedback and for increasing the score.
>
> Yes, the metric proposed in our previous response is the difference between the p-value of the target class and the max p-value of other classes, where p-values are computed as proportions of calibration points in the leaf where $x’$ belongs.
> Our intent was not to optimize the metric for our method but to use our tree-based partitioning to compute local p-values. We do, however, understand this potential concern.
>
> That’s why, following the reviewer’s suggestion, we now compute conformal p-values using LCP (not CONFEX-Tree) with the L1 kernel. This design choice provides better external validity because it is independent of the CONFEX-Tree approach. Results are provided below.
>
>
> | Method        | GermanCredit | CaliforniaHousing | GiveMeSomeCredit | AdultIncome |
> |---------------|--------------------|---------------|-------------------|--------------|
> | MinDist       |   0.140 ± 1.23e-02      |  2.44e-04 ± 2.62e-04   |   6.00e-03 ± 5.64e-03      |  2.11e-02 ± 1.33e-02   |
> | ECCCo         |   0.173 ± 4.51e-02      |   0.00e+00 ± 0.00e+00  |   0.00e+00 ± 0.00e+00      |     9.42e-03 ± 9.68e-03   |
> | Greedy        |   0.101 ± 3.58e-02      |   0.00e+00 ± 0.00e+00  |   6.90e-03 ± 4.06e-03      |  -5.43e-03 ± 2.44e-02  |
> | Wachter       | -8.07e-03 ± 1.25e-02    |   3.58e-02 ± 1.24e-02  |   0.00e+00 ± 0.00e+00      |  -2.91e-03 ± 1.03e-02  |
> | FACE          |    0.313 ± 3.71e-02     |   5.40e-03 ± 4.46e-03  |   1.92e-02 ± 1.52e-02      |  0.144 ± 1.71e-02  |
> | C-CHVAE       |    8.17e-02 ± 5.95e-02  |   0.00e+00 ± 0.00e+00  |   6.67e-04 ± 1.33e-03      |  2.26e-02 ± 1.22e-01   |
> | CNaive        |    0.213 ± 3.03e-02     |   3.26e-02 ± 9.71e-03  |   4.86e-03 ± 4.56e-03      |  6.28e-02 ± 2.71e-02   |
> | CTree         |    **0.483 ± 4.42e-02** |  **0.101 ± 3.57e-02**  |   **7.75e-02 ± 5.88e-02**  |   **0.210 ± 5.06e-02** |
>
>
> We see that over all datasets, CONFEX-Tree provides significantly higher certainty in the counterfactuals. The paper will be updated with these results and the corresponding results for random forest models.
> Note that to tune the bandwidth of the L1 kernel, we performed a grid search and picked the bandwidth that provided the best coverage (i.e., smallest coverage gap) on a held-out set.
>
> We hope this resolves your concerns; we’re happy to clarify any remaining points.

---

> ### Author Response · Authors · 2025-12-03
> **Summary Comment by Authors**
>
> We sincerely thank reviewer 55xF for their review and for their responsiveness, which allowed for a discussion to improve our paper.
>
> This review was the most critical with a score of 2, which was raised to 4 during the rebuttal period. The reviewer could not reply to our previous response, which addressed remaining concerns.
>
> Since our previous response, we have:
> - Updated the certainty metric to use LCP instead of CONFEX-Tree. Results can be found in Appendix A, and an explanation of the metric can be found in Appendix D.2.
> - Included Appendix C, which contains the scalability results requested by this reviewer and Reviewer wspM.
>
> We have responded to every point raised by this reviewer. For the benefit of readers, we have outlined the main changes made in order to improve our paper.
> - We detail the full MIP formulation of our method in Appendix B.
> - Within Appendix B, we included clarification of the linear nature of our method to prevent the confusion that affected this reviewer regarding the multiplication of binary variables.
> - To strengthen comparison with state-of-the-art, we included two plausibility-focused generators, FACE and C-CHVAE, to our experiments. The results discussion section in the paper was updated to describe the results. We made the claims in the abstract and introduction clearer following this as well.
> - Included Appendix C, which contains the scalability results requested by this reviewer and Reviewer wspM.  Appendix C considers the effect that the dimensionality of the data, size of the calibration set, and model complexity has on the time taken to generate CFXs. Additionally we consider how the size of the calibration set affects distance and plausibility.
> - To clearly show the effect of our method and its design, we include a Certainty metric. Following a discussion with the reviewer, we made this metric independent from our CONFEX-Tree method and utilise LCP instead.
> - To evaluate the significance of our results, we provide the p-values for a two-tailed t-test to check whether the result of the best performing generator is significantly different from other generators. We also increased the number of runs over which our results are computed to 5.
> - To improve the presentation of the results deferred to the appendix, we improved Appendix D by providing plots and a short commentary for each dataset, and reformatted tables to make them more readable.

---

### Official Review · Reviewer_bi8T · 2025-10-31

**Soundness:** 3
**Presentation:** 3
**Contribution:** 3
**Rating:** 6
**Confidence:** 4

**Summary:**

Counterfactual explanations are a popular explainability technique that provides the minimum input perturbation/change required to flip a model's decision. This paper proposes a method for generating uncertainty-aware counterfactual explanations, bringing together techniques from Conformal Prediction and Mixed-Integer Linear Programming. They propose new optimization tricks to obtain these counterfactuals. The counterfactuals obtained are designed to have low predictive uncertainty. Experiments compare SOTA methods across diverse benchmarks and metrics.

**Strengths:**

-- I enjoyed reading this paper. The idea of bringing together ideas from conformal prediction and MILP for counterfactuals is interesting. Conformal prediction techniques give prediction sets that are guaranteed to contain the true (unknown) outcome with a given probability. MILP provides a framework for deriving CFXs as a constraint-solving problem. The technique is called CONFEX.

-- They also introduce CONFEX-Tree which is a more efficient way of computing the uncertainty-aware counterfactuals, drawing inspiration from KD trees.

-- Experiments include several baselines and standard tabular datasets. Several metrics have been considered.

**Weaknesses:**

-- The tabular datasets used are also used for gradient-based counterfactual generation techniques. Could you highlight the differences and benefits of this class of technique from gradient based methods?
There's this line in limitations that would be great to elaborate and clarify: gradient-based methods like Wachter and ECCCo are less prone to this problem, but they sacrifice guarantees on CFX validity.

-- How do you compute plausibility and validity in the experiments?

-- The experiments seem to show tradeoffs. Cost increases to do better in other metrics. Some tradeoff plots can also be good.

--  It would be great to write a complete problem statement to clearly have all the desirables that the algorithm is seeking in one place. This will also help to contrast with what other techniques are doing, why CP is necessary, what are the performance metrics of interest, etc.

-- The abstract uses the word "guarantees" but the guarantee is only on validity with a certain probability, right? Would some of the other properties provably hold or they just hold empirically, e.g., LOF? It would be nice to clarify. Again, going back to the problem statement with the desirable properties, then it will be nice to clarify which of those desirable properties provably hold, and which ones hold intuitively (empirically observed benefit).

-- It would be nice to consider including a theorem statement if there is any kind of provable guarantee on the counterfactuals meeting any of these criteria.

**Questions:**

Some of my questions are in the weakness section.

1. What is the advantage/disadvantage over gradient-based methods like Wachter?

2. More clarification on the problem statement, all the properties desired, and which of them are provably achieved (guaranteed), vs which properties are found to hold empirically? This makes the presentation more streamlined.

---

> ### Author Response · Authors · 2025-11-20
>
> ## MILP vs Gradient-Based Methods
> **Difference & Benefits.**
> MILP provides complete search with global optimality guarantee: if a valid CFX exists, MILP will find it and guarantee its optimality (e.g., closest one). Gradient-based methods may get stuck in local minima or fail to find valid CFXs even when one exists.
>
> **On Valid Guarantees.**
> The statement about gradient-based methods sacrificing validity refers to their incompleteness. Watcher and ECCCo use gradient descent, which may not only fail to return optimal solutions but also fail to return valid CFXs (i.e. points that the model predicts to be the target class). You can see this in the experimental results as well: Watcher achieves only 84% validity on GermanCredit, while MILP-based methods guarantee 100% validity if a CFX exist, and if no CFX exists the problem will be infeasible.
>
> **Comparison against Watcher.**
> The advantages of MILP-based methods over Watcher (or other gradient based methods) is that they guarantee validity and optimality and they can solve for addiitional constraints (e.g., actionability, or conformal prediction constraints) even if these are not differentiable (as long as they can be encoded in MILP). Also, they support non-differentiable models (e.g., decision trees).
> On the other hand, they tend to have a higher computational cost than gradient-based approaches and  require piece-wise linearly-representable models. Nonetheless, this covers a wide array of models, including MLPs and CNNs with ReLU and LeakyReLU activations, decision trees, random forests, and gradient-boosted trees.
>
> ## Metrics Computation
> We have provided the complete details on how we compute each metric in Appendix C.2. We also provide a description of the metrics you have mentioned here:
>
> **Validity.**
> Validity refers to the proportion of CFXs which successfully flip the model prediction to that of the target class $P(f(x’)=y^+)$. Invalid CFXs are excluded from other metrics.
>
> **Plausibility.**
> To measure plausibility we use the Local Outlier Factor (LOF). LOF is a method which “computes the local deviation of density of a given sample with respect to its neighbours” [1]. With unseen points, a score of +1 indicates consistency with the observed data, and -1 indicates the opposite. The LOF we use considers the closest 20 points to the counterfactual which belong to the target class as neighbours. We also consider an alternative notion of plausibility, “implausibility”, which was used by [2] and report the results in the appendix.
>
> ## Trade-Offs
> You’re correct about observing a trade-off. Our uncertainty constraints indeed result in better sensitivity/robustness and plausibility. But they also restrict the feasible set of solutions, meaning that they lead to CFXs with higher distance. These are displayed in Fig 1 plots (a) and (b), for CaliforniaHousing, and the same plots for other datasets have now been added to Appendix C.
>
> ## Problem Statement and Guarantees
> We thank the reviewer for the suggestion to include a problem statement and agree that it helps with the flow of the paper. We have now added the problem statement to Lines 192-200 and related on guarantees to lines 255-276 and 377.
>
> To summarise the guarantees that our method provides, our aim is to find the minimal distance counterfactual that yields the target class with probability above a chosen threshold. Importantly, this probability is conditional on the specific CFX and not an average over the data distribution. In the paper revision, we show that this chance constraint can be implemented by requiring that the conformal prediction set is a singleton containing the target class. Additionally, our method guarantees validity, as described on Lines 223-226.
>
> Furthermore, we observe that our method performs empirically well on metrics such plausibility and sensitivity/robustness despite not explicitly optimising for them. As we mention in Section 2, our method can be extended to enforce further desirable properties such as actionability, causality and diversity by adding further constraints, dependent on the domain.
>
> ## References
> [1] The scikit-learn developers, “https://scikit-learn.org/stable/auto_examples/neighbors/plot_lof_outlier_detection.html”
>
> [2] Altmeyer, Patrick, et al. "Faithful model explanations through energy-constrained conformal counterfactuals." Proceedings of the AAAI conference on artificial intelligence. Vol. 38. No. 10. 2024.

---

> ### Author Response · Authors · 2025-12-03
> **Summary Comment by Authors**
>
> We sincerely thank reviewer bi8T for their review and note that the reviewer did not respond to our rebuttal whilst this was possible. Nevertheless, we have used their feedback to improve the paper, and describe the main changes made below:
> - Our main change to the paper was the inclusion of a problem statement, which makes clear the objective of our method.
> - We also provide further statements on guarantees.
> - These details are included on Lines 199-207, 262-280, and 404.

---

### Author Response · Authors · 2025-11-20
**New Revision**

We thank the reviewers for their feedback and following this we have made several improvements to the paper. The new revision is available at https://openreview.net/pdf?id=FE3FVddJaT.

We summarise the changes below:
- Added a formal problem statement and more discussion on the guarantees provided by CONFEX.
- Updated Table 1 and Figure 2 after increasing the number of runs in our experiments to 5 and obtaining additional results for competing generators FACE and C-CHVAE.
- Conducted statistical significance tests of our results in Table 1, showing that our results are significant. Discussion can be found in Appendix A.
- Detailed the full MILP formulation of our method, which can be found in Appendix B.
- Rewritten Appendix C, which details the full results for all datasets, has been improved with plots and a short commentary for each dataset, and tables were  reformatted to be more readable.
- Minor wording changes suggested by reviewer 55xF.

---

### Author Response · Authors · 2025-12-03
**New Revision and Further Comments**

We thank all of the reviewers for their efforts in evaluating our paper. We have released our latest revision here: https://openreview.net/pdf?id=FE3FVddJaT.

During the rebuttal, we addressed every point raised, in particular those of our most critical reviewer 55xF. Unfortunately, reviewers bi8T and wspM did not participate in the discussion; nevertheless, we have made changes following their feedback and we have answered their questions.

The main changes in the latest revision are (a) including the updated certainty metric and (b) including the scalability analysis appendix. For the benefit of readers, we have described all improvements made to our paper following each reviewer’s feedback in a 'Summary Comment' under each review.

---

### Meta-Review · Area_Chair_ToXG · 2026-01-07

**Summary:**

This paper proposes CONFEX, a framework for generating uncertainty-aware counterfactual explanations using conformal prediction combined with  Mixed-Integer Linear Programming (MILP) based optimization. The central contribution is identifying that counterfactual search violates exchangeability, and addressing this via localized conformal prediction (LCP), with a scalable tree-based variant (CONFEX-Tree) that enables local / group-conditional coverage guarantees. Reviewers agreed the problem is important and the technical direction is principled, but initially raised serious concerns about correctness (linearity of the MILP), clarity and completeness of the formulation, scope and wording of claims (especially around plausibility and “state-of-the-art” comparisons), missing baselines, unclear metrics, and lack of scalability analysis.

Through an extensive rebuttal and multiple revisions, the authors substantially strengthened the paper: they provided the full MILP formulation, clarified linearization, added missing baselines (FACE, C-CHVAE), refined claims, added statistical tests, introduced a principled certainty metric independent of CONFEX-Tree, and included a detailed scalability appendix. The most critical reviewer raised their score after these changes.

Overall, it is fair to say that the paper is a careful integration of existing components (localized conformal prediction + MILP counterfactual search) with limited conceptual novelty, and its practical scope is constrained to MILP-encodable models and relatively small tabular settings. Empirical advantages over strong plausibility-/robustness-oriented baselines are mixed and sometimes modest, making it unclear that the added complexity and guarantee machinery consistently translates into meaningful real-world benefit. Thus, this paper is borderline, presenting an incremental contribution just below the ICLR-level acceptance.

**Reviewer Concerns:**

Addressed by the rebuttal / revisions:
* Correctness of the MILP formulation: clarified that all products involve binary variables and are linearizable; full MILP formulations for CONFEX and CONFEX-Tree added in Appendix B.
* Lack of a clear problem statement and guarantees: authors added a formal problem statement and explicitly distinguished which properties are guaranteed (validity, conditional coverage) versus empirical (plausibility, robustness).
* Overstated claims about plausibility and comparison to SOTA: wording was revised to specify comparison against uncertainty-aware generators; abstract and introduction were softened accordingly.
* Missing baselines: FACE and C-CHVAE were added; explanations given for excluding CLUE, PROPLACE, DACE in this setting.
* Metrics ambiguity and statistical validity: detailed metric definitions added; number of runs increased; statistical significance tests reported; validity-selection bias clarified.
* Scalability concerns: an extensive appendix now analyzes runtime versus calibration size, dimensionality, and model class; CONFEX-Tree scalability discussed.
* Uncertainty evaluation: a new certainty metric based on LCP p-values (independent of CONFEX-Tree) was introduced and evaluated across datasets.

Still partially outstanding:
* Novelty is incremental: the method combines existing tools (LCP + MILP) in a principled way; reviewers acknowledge this but still view it as engineering-heavy rather than conceptually new.
* Model scope limitations: applicability restricted to MILP-encodable models; authors justify this for recourse settings, but it remains a limitation.
* Practical impact of guarantees vs heuristics: the motivation for formal coverage guarantees is argued convincingly, but empirical benefit over strong plausibility-based methods is sometimes modest and dataset-dependent.

**Reviewer Scores:**

* bi8T: original 6. Concerns about problem statement, guarantees, and trade-offs were fully addressed. Likely stays 6 or increases to 7.
* 55xF: original 2, later raised to 4 after rebuttal and further to around 5 once claims, formulation, and metrics were clarified.
* wspM: original 4. Many concerns (scalability, motivation for guarantees, scope) were addressed in revisions; likely increases to 5.

---

### Decision · Program_Chairs · 2026-01-26

Reject